# CRISPR screens reveal convergent targeting strategies against evolutionarily distinct chemoresistance in cancer

Chunge Zhong [1,2,3,4,17], Wen-Jie Jiang [5,17], Yingjia Yao[1,2,3], Zexu Li [1,2,3], You Li[1,2,3], Shengnan Wang[1,2,3], Xiaofeng Wang[1,2,3], Wenjuan Zhu[1,2,3], Siqi Wu[1], Jing Wang[1], Shuangshuang Fan[1,2,3], Shixin Ma[1,2,3], Yeshu Liu[1], Han Zhang[1,2,3], Wenchang Zhao[1,2,3], Lu Zhao[1,2,3], Yi Feng[1,2,3], Zihan Li [1,2,3], Ruifang Guo[1], Li Yu[1], Fengyun Pei[6], Jun Hu[7,8,9], Xingzhi Feng[8,9], Zihuan Yang [8,9], Zhengjia Yang[10], Xueying Yang[11], Yue Hou [1], Danni Zhang[12], Dake Xu [12], Ren Sheng [1], Yihao Li[13], Lijun Liu[1], Hua-Jun Wu [14,15,16] ✉, Jun Huang[6,7,8,9] ✉ & Teng Fei [1,2,3,4] ✉

Resistance to chemotherapy has been a major hurdle that limits therapeutic benefits for many types of cancer. Here we systematically identify genetic drivers underlying chemoresistance by performing 30 genome-scale CRISPR knockout screens for seven chemotherapeutic agents in multiple cancer cells. Chemoresistance genes vary between conditions primarily due to distinct genetic background and mechanism of action of drugs, manifesting heterogeneous and multiplexed routes towards chemoresistance. By focusing on oxaliplatin and irinotecan resistance in colorectal cancer, we unravel that evolutionarily distinct chemoresistance can share consensus vulnerabilities identified by 26 second-round CRISPR screens with druggable gene library. We further pinpoint PLK4 as a therapeutic target to overcome oxaliplatin resistance in various models via genetic ablation or pharmacological inhibition, highlighting a single-agent strategy to antagonize evolutionarily distinct chemoresistance. Our study not only provides resources and insights into the molecular basis of chemoresistance, but also proposes potential biomarkers and therapeutic strategies against such resistance.

Despite rapid progress in targeted therapy and immunotherapy, chemotherapy, which employs cytotoxic chemical drugs to destroy rapidly growing cells, continues to serve pivotal roles in cancer treatment[1,2]. Based on the mechanism of action or chemical structure, chemotherapy drugs can be grouped into different categories including alkylating or alkylating-like agents (inducing DNA damage, e.g., cisplatin and oxaliplatin), antimetabolites (substitute of metabolites for DNA/RNA synthesis, e.g., 5-fluorouracil and capecitabine), antitumor antibiotics (DNA intercalator to prevent cell replication, e.g., doxorubicin, also known as adriamycin), topoisomerase inhibitors

(e.g., irinotecan) and mitotic inhibitors (e.g., microtubule inhibitor taxane such as docetaxel and paclitaxel)[3–7]. For many malignancies, chemotherapy remains the first line of treatment. For instance, the conventional first-line therapy for colorectal cancer is the combination of the chemotherapy drugs fluoropyrimidine (e.g., 5-fluorouracil) and either oxaliplatin or irinotecan (FOLFOX or FOLFIRI)[8]. Importantly, chemotherapy can be the only treatment regimen in some cases of unresectable or metastatic cancer[9,10]. Chemotherapy drugs are more frequently used to shrink tumors or eradicate residual cancer cells before or after surgery[11,12]. Furthermore, by applying chemotherapy in

a sequential or combinatorial fashion together with additional treatment modalities such as targeted therapy and immunotherapy, enhanced therapeutic benefits can be achieved[13,14]. However, resistance to chemotherapies remains a primary cause of treatment failure and disease relapse for cancer patients[15,16].

Despite evidence connecting several individual genes to chemoresistance, the understanding of such a phenomenon is still incomplete, and effective therapeutic strategies to overcome chemoresistance are still lacking. It is urgently demanded to take a systematic and comprehensive view of the molecular underpinnings of various chemoresistance. Such efforts will also facilitate the discovery of genetic biomarkers informing the potential outcome of chemotherapy and optimize the treatment regimen accordingly. Recent work using genomic profiling and functional genomics are great endeavors to this goal especially for chemotherapies in hematologic malignancies[17-22], however, a systematic delineation of genetic drivers and vulnerabilities for resistance to widely used chemotherapies in solid tumors is still lacking.

To fill this gap, here we conduct thirty genome-wide CRISPR knockout screens to systematically explore the genetic causes of chemoresistance in multiple cancer cell lines for seven widely used chemotherapy drugs (oxaliplatin, irinotecan, 5-fluorouracil, doxorubicin, cisplatin, docetaxel, and paclitaxel) in solid tumors. Through comprehensive analysis combined with additional experimental exploration, we reveal diversified driving forces and molecular features towards chemoresistance from a general and global perspective and propose actionable targeting strategies to therapeutically overcome evolutionarily distinct chemoresistance.

## Results

### Genome-scale CRISPR screens for chemoresistance genes

To systematically identify the genes whose loss-of-function drives chemoresistance in cancer, we performed multiple genome-scale CRISPR knockout screens for seven commonly used chemotherapy drugs (oxaliplatin, irinotecan, 5-fluorouracil, doxorubicin, cisplatin, docetaxel and paclitaxel) in corresponding colorectal, breast and lung cancer cells (Fig. 1a and Supplementary Data 1). Six representative cell lines (HCT116 and DLD1 for colorectal cancer; T47D and MCF7 for breast cancer; A549 and NCI-H1568 for lung cancer) with different genetic backgrounds and varied responses to indicated drugs were employed for the chemogenomic screens (Supplementary Fig. 1a–c). Pooled lentiviruses encapsulating Cas9 nuclease and 92,817 single guide RNA (sgRNA) targeting 18,436 protein-coding genes in the human genome were transduced into these cells at a low multiplicity of infection (MOI) (-0.3) followed by puromycin selection. After that, cells were cultured for several days in the presence of either DMSO or the corresponding drugs for cell fitness screens. Genomic DNA was then harvested, and sgRNA abundance was quantified by high-throughput sequencing (Fig. 1a). A total of 30 genome-scale screens (24 for drug challenge and 6 for vehicle treatment) were conducted and analyzed by the MAGeCK algorithm[23,24]. Multiple quality control (QC) measurements indicated the high quality of these screens (See Methods). For vehicle (DMSO) treatment screens without any selection pressure, we observed a strong negative selection for core essential genes, and the top negatively selected genes were enriched for essential functional terms such as translation, ribosome biogenesis, and RNA splicing (Supplementary Fig. 1d, e). This demonstrates that our screens can accurately identify functional hits associated with cell fitness. Different drugs imposed differential selection pressure on screened cells as evidenced by the Gini Index informing distribution evenness of sgRNA counts (Supplementary Fig. 1f). The effect of given gene knockout was quantified by an RRA score calculated by MAGeCK with a lower negative value indicating stronger negative selection and a higher positive value denoting stronger positive selection. "Chemoresistance genes" here were defined as those whose loss-of-

function confer resistance to chemotherapy drugs on screened cells and can be pulled out from positively selected hits (score$^{drug}$ - score$^{DMSO}$ > 3 and score$^{drug}$ > 3). Notably, this definition here is only for better understanding and mentioning the resistance phenotype due to functional perturbation of corresponding gene. The gene's function per se is likely to suppress or restrain chemoresistance.

### A systematic view on genetic maps of chemoresistance

We identified tens of chemoresistance genes for each single cell line in response to a given drug (Fig. 1b). Global analysis across conditions showed that chemoresistance genes tended to cluster together primarily by cell-of-origin rather than type of chemotherapy drugs (Fig. 1c), suggesting the importance of genetic background in determining chemoresistance. The cellular specificity was further evidenced by the few overlap of chemoresistance genes across different cell lines (Supplementary Fig. 1g), strengthening the importance of genetic background and adding complexity to the genetic map of chemoresistance. We then took another route by generating drug-specific chemoresistance gene cohorts through a combination of results across cell lines for a given drug in the following analysis. Each chemotherapy drug tested had 81 to 337 chemoresistance genes with a few shared between drugs (Fig. 1d and Supplementary Data 1). Both the genetic background of cells and the drug's mechanism of action contribute to such variance of chemoresistance genes. Interestingly, when focusing on top hits, several of them were shared by different drugs (Fig. 1e). For example, TP53 was a top hit for oxaliplatin resistance in TP53-wild type (WT) HCT116 cells but not in TP53-mutated DLD1 cells, highlighting how genetic background may affect screening results under certain scenarios (Fig. 1e and Supplementary Fig. 1a; Supplementary Data 1). As a well-characterized multi-functional gene[25,26], p53 (encoded by TP53) plays critical roles in DNA damage response, cell cycle regulation, chromosomal instability, and so on. Here, we found that TP53 knockout drove resistance to several chemotherapy drugs on top of its roles in normal cell growth (Fig. 1e). Loss-of-KEAP1, a sensor of oxidative and electrophilic stress, demonstrated irinotecan and cisplatin resistance (Fig. 1e), which agrees with previous findings that these drugs induce oxidative stress[27,28]. Microtubule-related genes such as KIFC1, KATNA1, KIF18B, WDR62, and KATNBL1 were among the top chemoresistance genes for docetaxel and paclitaxel (Fig. 1e), in line with the microtubule-targeting mechanism of these two drugs[29].

Beyond individual gene-based resistance mechanisms, global investigation through functional enrichment and network analysis using chemoresistance gene cohorts allowed us to scrutinize specific or shared functions and/or pathways within cells that might be involved in driving chemoresistance (Fig. 2a, b). For instance, the "Cell cycle" function was strongly implicated in oxaliplatin, irinotecan, or doxorubicin resistance but to a lesser extent for other drugs. "Signal transduction in response to DNA damage" was weakly but broadly enriched among chemoresistance genes for all the tested drugs. Interestingly, "Regulation of fibroblast proliferation" also emerged as a multidrug resistance function, suggesting the pivotal roles of tumor microenvironment in shaping the responses for many chemotherapy drugs (Fig. 2a). In contrast, mitochondria-related terms were specifically associated with irinotecan resistance (Fig. 2a), consistent with the report that targeting mitochondrial oxidative phosphorylation abrogates irinotecan resistance in lung cancer cells[30]. There were also broad protein-protein interactions between chemoresistance genes as exemplified by the case of oxaliplatin (Fig. 2b). These data systematically provide mechanistic insights into how chemoresistance may develop at the gene or pathway levels.

### Clinical relevance of chemoresistance genes

Next, we sought to investigate whether these chemoresistance genes identified by in vitro screens have relevance to human clinics. Given the loss-of-function nature of our screening system, we first examined

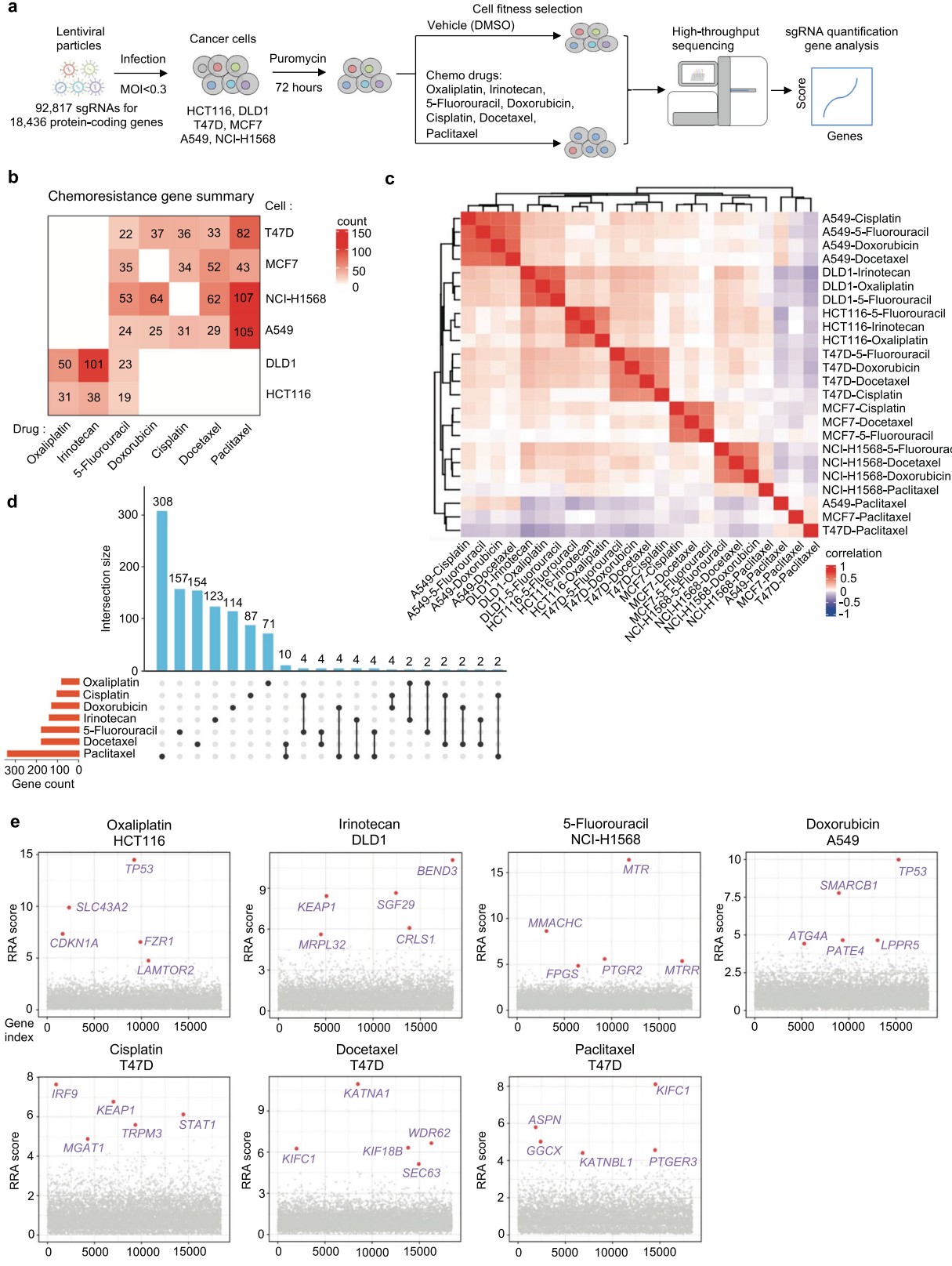

**Fig. 1 | Genome-scale identification of chemoresistance genes. a** Workflow of genome-wide CRISPR knockout screen to identify chemoresistance genes. **b** The number of chemoresistance genes identified from each indicated screen. **c** Pearson correlation of RRA scores for chemoresistance genes among each sample. **d** Overlap of chemoresistance genes for each chemotherapy drug. **e** Top chemoresistance genes from representative screens for each chemotherapy drug.

the relationship between chemoresistance genes with tumor suppressor genes (TSGs) whose loss-of-function contribute to the development of cancer. Using cataloged TSG list from the COSMIC Cancer Gene Census[31], we indeed observed a few but significant overlap

between the two gene groups (Fig. 2c, d). In contrast, oncogenes and pan-essential genes were not enriched among chemoresistance genes (Fig. 2d). Despite scattering reports linking TSGs to drug resistance such as *TP53* which was also repeatedly found here (Fig. 1e)[32], our study

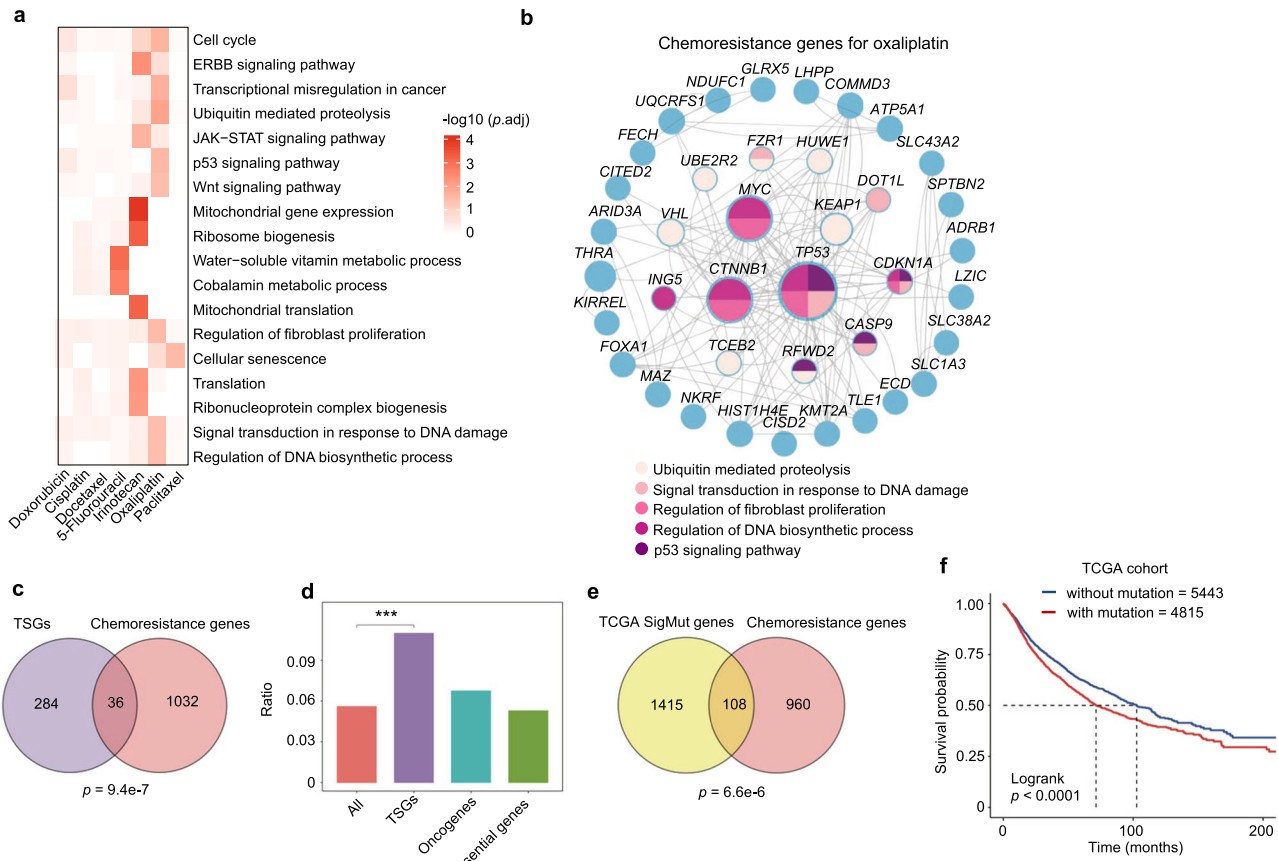

**Fig. 2 | A systematic view on chemoresistance genes. a** Enriched functional terms from Kyoto Encyclopedia of Genes and Genomes (KEGG) and Gene Ontology (GO) for chemoresistance genes of indicated chemotherapy drugs. Unpaired two-sided $t$ test for $p$ value with Benjamini-Hochberg (BH) adjustment. **b** Protein-protein interaction network for chemoresistance genes of oxaliplatin. The genes belonging to indicated functional terms are color-coded. **c** Venn diagram showing the overlap between tumor suppressor genes (TSGs) and chemoresistance genes. The overlap is significant as calculated by the Hypergeometric test (two-sided, $p = 9.4e-7$). **d** Enrichment ratio of chemoresistance genes in all genes, TSGs, oncogenes, and essential genes. Unpaired two-sided t test, $p = 6.3e-4$. **e** Overlap of significantly mutated genes from The Cancer Genome Atlas (TCGA) cohorts (pan-cancer) and chemoresistance genes identified by our screens. The overlap is significant as calculated by the Hypergeometric test (two-sided, $p = 6.6e-6$). **f** Overall survival analysis of all TCGA patients using the mutational status of 108 overlapping genes in **e**. Kaplan–Meier statistical test, $p = 2.3e-9$.

systematically demonstrate the implication of TSGs in mediating chemoresistance. By exploring somatic mutation profile from The Cancer Genome Atlas (TCGA), we found a significant portion of chemoresistance genes were highly mutated in human tumors (Fig. 2e). Furthermore, patients bearing mutations on these chemoresistance genes exhibited significantly poor survival (Fig. 2f), which might be partially due to drug resistance elicited by these altered chemoresistance genes. The frequently mutated chemoresistance genes included known regulators of drug responses such as *TP53* and *MED12* as well as other genes previously unappreciated during chemoresistance[33,34] (Supplementary Fig. 2a). For some of these genes, either mutation or low expression might correlate with poor survival in certain cancer types (Supplementary Fig. 2b–i), further supporting the clinical relevance of our findings. Moreover, the expression levels of several top chemoresistance gene hits in colorectal cancer cells such as *KEAP1*, *SLC43A2*, or *TP53* were significantly down-regulated in recurrent colorectal tumors after oxaliplatin treatment (Supplementary Fig. 2j). Taken together, the mutational profile and/or expression status of chemoresistance genes may serve as potential biomarkers to predict the response to chemotherapy.

**A snapshot of chemosensitizer genes**
We further identified chemosensitizer genes from these genome-wide chemoresistance screens whose knockout accelerate cytotoxic cell death for given chemotherapy drugs (score$^{drug}$−score$^{DMSO}$ < −3 and

score$^{drug}$ < −3) (Supplementary Fig. 3a and Supplementary Data 1). Different drugs had distinct sensitizer gene profiles, except that irinotecan had no sensitizer gene identified due to a strong selection pressure in the screens therefore making it inappropriate for analyzing negatively selected sensitizer genes (Supplementary Figs. 3b and 1f). Interestingly, different drugs shared more chemosensitizer genes than chemoresistance genes (Supplementary Fig. 3b and Fig. 1d). Consistently, some of the top sensitizer genes, such as *NEK7*, *KDM5C*, *KLF5*, and *BCL2L1* were shared between different drug groups (Supplementary Fig. 3c), suggesting consensus vulnerabilities for multidrug resistance. Functional enrichment analysis for these sensitizer genes indicated potential pathways that might be targeted for combination therapy (Supplementary Fig. 3d). For instance, drugs that interfere with DNA repair processes may enhance the efficacy of oxaliplatin, whereas those that target RNA splicing pathways may increase cellular sensitivity to paclitaxel. These data suggest that potential therapeutic vulnerabilities may converge although the routes to chemoresistance are divergent. On the other hand, like chemoresistance genes, chemosensitizer genes also varied between different cell lines (Supplementary Fig. 3e), further underscoring the importance of cell-of-origin. Together, chemogenomic screens revealed multiple genetic players and mechanisms underlying chemoresistance with a vast diversity primarily determined by genetic background and mechanism of action of drugs. Considering such complexity, solely based on targeting specific chemoresistance drivers or sensitizers, it is still probably

difficult to implement a defined and practical therapy against chemoresistant tumors with evolutionarily diversified characters. Instead, exploring multiple lines of chemoresistance simultaneously at an advanced level may be the key to uncover convergent targeting strategy, if any, for practical application in clinics.

## Rapid derivation of chemoresistant cells via evolutionarily distinct routes

Typical approaches to study chemoresistance mainly through either direct comparison between drug-tolerant versus -sensitive cells with different genetic backgrounds, or derivation of matched drug-resistant cells from sensitive parental cells via long-term drug challenge. The difference in genetic backgrounds complicates the data interpretation for the former method, while the latter approach is time-consuming and also constrained by undefined factors behind resistance. Benefited by our identification of chemoresistance genes, we tried to derive multiple chemoresistant cells with clearly defined routes in a time- and resource-efficient manner. To test this possibility, we first focused on several top-selected chemoresistance genes from HCT116-oxaliplatin and DLD1-irinotecan groups. Using two independent sgRNAs targeting each gene (Supplementary Fig. 4a–c), we found that knockout of *SLC43A2*, *TP53*, or *CDKN1A* significantly conferred oxaliplatin resistance to HCT116 cells, compared to Vector or *AAVS1* control groups (Fig. 3a). With continued drug treatment on these knockout cells, we could readily establish oxaliplatin-resistant cell lines with significantly increased response dose within just one week (Fig. 3b and Supplementary Fig. 5a). *TP53* and *CDKN1A* (encoding p21) are well-known tumor suppressor genes that have been shown to be implicated

in multidrug resistance including oxaliplatin[34–36]. SLC43A2, a member of the L-amino acid transporter family, was reported to control T cell function through regulating methionine metabolism and histone methylation[37], but was not known to be implicated in chemoresistance. Similarly, knockout of *BEND3*, *KEAP1,* and *SGF29* also led to quick generation of irinotecan-resistant DLD1 cells within around 10 days (Fig. 3c and Supplementary Fig. 5b). BEND3 is a transcriptional repressor and epigenetic regulator[38,39]. KEAP1 acts as a sensor of oxidative and electrophilic stress. SGF29 (also known as CCDC101) is a subunit of the chromatin-modifying SAGA complex[40,41]. Next, we extended this approach to broader cell lines using corresponding top-ranked chemoresistance genes and successfully established 5-fluorouracil-resistant MCF7 and NCI-H1568 cells in around two weeks by knocking out *MED19* and *MMACHC* gene, respectively (Fig. 3d, e; Supplementary Figs. 4d, e and 5c, d). *MED19* is a subunit of Mediator complex that plays critical role during transcription control[42] while *MMACHC* is postulated to be involved in cobalamin (vitamin B12) trafficking[43]. In parallel, we also derived oxaliplatin-resistant HCT116 cells and irinotecan-resistant DLD1 cells by gradual drug adaptation and "randomly" clonal selection, which took around three months to achieve a comparable resistance level to that of the above specific gene-derived resistant lines (Fig. 3f–i and Supplementary Fig. 5e, f).

The quick establishment of chemoresistant cells via specific gene knockout suggests a sufficient role but not merely a promotive function for the individual gene loss in driving resistance. The short duration of drug treatment on CRISPR-mediated gene knockout cell pools seemed to just eradicate unsuccessfully edited sensitive cells. The resistance was, in essence, established once the specific gene was

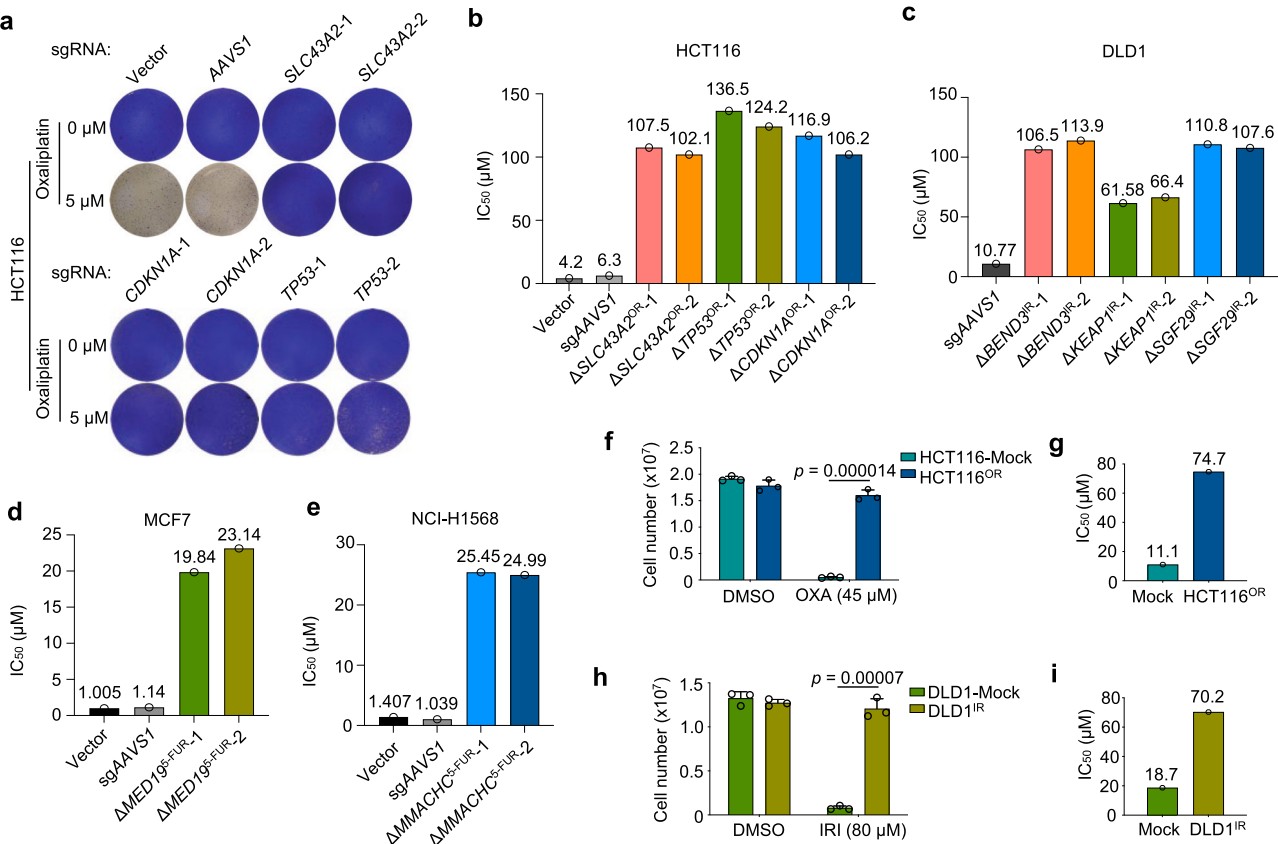

**Fig. 3 | Rapid derivation of chemoresistant cells. a** Crystal violet staining of HCT116 cells with lentiviral infection of indicated sgRNAs and oxaliplatin treatment. **b**–**e** Bar chart showing the value of IC$_{50}$ of indicated cells derived from three experimental replicates shown in Supplementary Fig. 5a–d. **f**–**i** Establishment of acquired resistance to oxaliplatin in HCT116 (**f**) and to irinotecan in DLD1 (**h**) cells

by "randomly" clonal selection with gradual increase of drug treatment. OXA oxaliplatin, IRI irinotecan, Mean ± SD with *n* = 3 biological replicates. Unpaired two-sided *t* test for *p* value. The IC$_{50}$ values are shown in **g** and **i**, respectively, which are derived from three experimental replicates shown in Supplementary Fig. 5e, f.

knocked out without the need for corresponding drug adaptation. To test this and determine whether a single gene's loss-of-function is sufficient to confer chemoresistance, using *TP53* as an example, we selected multiple single-cell clones with complete disruption of p53 expression by any of the two independent sgRNAs (Supplementary Fig. 4f). Compared to control cells, all the nine Δ*TP53* clones immediately exhibited intrinsic resistance to oxaliplatin treatment without prior drug adaptation (Supplementary Fig. 4g, h). Despite previous implications of p53 during oxaliplatin resistance[34,35], our data here directly and unambiguously demonstrate that a single genetic alteration such as *TP53* loss-of-function is sufficient for oxaliplatin resistance in HCT116 cells. We inferred that the other top chemoresistance genes whose loss-of-function directly drove the quick establishment of resistant cells also work in a similar manner. Furthermore, the importance and clinical relevance of several such genes were also supported by the survival analysis in corresponding tumor types (Supplementary Fig. 2f–i). For those mid- or low-ranked chemoresistance genes, knockout of themselves might not be sufficient to acquire immediate resistance but help to accelerate the development of chemoresistance over time. These data not only consolidate the drug response phenotype of the top chemoresistance hits, but also provide valuable resources for quick generation of multiple chemoresistant cell lines.

## Cellular features of chemoresistant cells

The establishment of multiple resistant cells by knocking out specific genes or long-term drug selection allowed us to characterize the molecular mechanisms behind these evolutionarily distinct chemoresistances. Using oxaliplatin resistance as a model system, we firstly confirmed that these resistant cells derived from distinct sources showed no significant morphological changes (Fig. 4a) and survival decrease under oxaliplatin treatment (Fig. 4b). Cell cycle analysis showed that oxaliplatin significantly reduced cell fraction of S phase and arrested sensitive cells at G2/M phase at medium (5 μM) (Fig. 4c) or high dose (10 μM) (Supplementary Fig. 6a), while at low dose of 2.5 μM (Supplementary Fig. 6b) cells were also arrested at G1 phase. In contrast, resistant cells established from different routes all successfully counteracted such effect (Fig. 4c and Supplementary Fig. 6a, b). Consistently, we observed compromised expression of p53 or p21 across multiple resistant lines compared to sensitive controls (Fig. 4d), which are important cell cycle arrest mediators in response to DNA lesions. Oxaliplatin treatment caused significant decrease for cyclin D1, Cyclin E, phosphorylated Rb (p-Pb), and CKD1 in sensitive control cells, while different resistant lines could largely withstand such decrease (Fig. 4d). Furthermore, ERK activation is reported to mediate cell cycle arrest and apoptosis after DNA damage[44–46]. Indeed, we found that ERK signaling was repressed or less activated under oxaliplatin treatment across all the tested oxaliplatin-resistant cells (Supplementary Fig. 6c). Apoptosis was only slightly increased upon oxaliplatin treatment using doses of current study in sensitive control cells but not in resistant cells (Fig. 4e). These data suggest that cell cycle arrest but not cell apoptosis is the major cytostatic effect of oxaliplatin at relatively low doses in colorectal cancer. Resistant cells from different sources usually exhibit impaired p53/p21 signaling through genetic or epigenetic perturbations to counteract such cell cycle arrest to develop oxaliplatin resistance.

We further examined irinotecan resistance models. Various resistant cell lines displayed similar morphology and cell growth advantages under irinotecan treatment (Supplementary Fig. 7a, b). Cell cycle analysis indicated that irinotecan elicited a significant arrest at G2/M phase and decrease in G1 and S phases in sensitive control cells, while multiple resistant cells did not undergo such cell cycle arrest (Supplementary Fig. 7c). Similar to oxaliplatin, apoptosis rate was significantly increased only in sensitive control cells, but the overall percentage of apoptosis was not large under tested irinotecan doses (Supplementary Fig. 7d). Unlike oxaliplatin resistance models, we did

not see decreased ERK or differences of other tested signaling activities between sensitive and resistant cells by immunoblot (Supplementary Fig. 7e). These results indicate that targeted therapy against these cell growth-related kinases might not be attractive options to overcome irinotecan resistance in colorectal cancer.

## Gene expression signatures underlying oxaliplatin or irinotecan resistance

To characterize intrinsic molecular features underlying chemoresistance, we performed transcriptome profiling by RNA sequencing (RNA-seq) for multiple chemoresistant cells in basal state without drug challenge (Supplementary Data 2). Comparison was made firstly between each line of resistant cells and their parental control sensitive cells to retrieve differential gene signatures for each resistance model. After that, these chemoresistance-related gene signatures were compared across different models to see whether shared or model-specific gene expression alterations exist for chemoresistance. We observed varied gene expression patterns between different oxaliplatin-resistant cells in comparison to control sensitive cells (Fig. 4f, g), indicating that different driving forces could dictate divergent gene expression programs towards resistance. Notably, Δ*TP53*[OR]-2 and Δ*CDKN1A*[OR]-1 cells shared a more portion of differentially regulated genes (Fig. 4f, g), which is concordant with that p53 can transcriptionally induce p21 expression in response to DNA damage[47]. Interestingly, despite differential gene expression profiles, cell cycle related genes were repetitively enriched across multiple oxaliplatin-resistant cell lines (Fig. 4h). For example, *CCNA2* (encoding cyclin A2), *CDK1* and *CCNB1* (encoding cyclin B1) were generally up-regulated while negative regulators of cell growth such as *CDKN1C* and *GPER1* were broadly down-regulated in resistant cells compared to sensitive cells (Fig. 4i). These data further suggest that propelling cell cycle progression might be one of the major routes for cells to counteract cytotoxic effect of oxaliplatin.

For differential genes associated with irinotecan resistance, there was a significant overlap of such gene sets between the three gene-specific resistant cells (Δ*BEND3*[IR]-1, Δ*KEAP1*[IR]-2 and Δ*SGF29*[IR]-2) while randomly derived resistant cells (DLD1[IR]) had a distinct gene expression pattern (Supplementary Fig. 8a, b). It is interesting that these three mechanistically distinct genes might work synergistically along similar routes to confer irinotecan resistance. Moreover, genes related to "microtubule cytoskeleton organization", "mitotic cell cycle" and "DNA damage response" were significantly enriched among those up-regulated genes across three gene-specific resistant cells (Supplementary Fig. 8c, d), suggesting the implication of these pathways in shaping chemoresistance of indicated cells. In addition, several drug transporter genes involved in multidrug resistance were also up-regulated in some resistant cells (Supplementary Fig. 8e), further complementing the mechanisms for chemoresistance. When focusing on those commonly regulated genes in the three gene-specific resistant cells, we found that shared up-regulated genes tended to be negatively selected during chemoresistance screen (Supplementary Fig. 8f), and their high expression correlated to bad survival for colorectal cancer patients (Supplementary Fig. 8g).

To exclude the potential bias in gene-specific resistant cell models, we specifically examined the gene expression profiles of traditional drug resistance model derived via gradual drug pulsing and clonal selection. We observed that positively selected hits in chemoresistance screen tended to be down-regulated in corresponding resistant cell lines (HCT116[OR] or DLD1[IR]), supporting the loss-of-function effect of these genes in driving chemoresistance (Supplementary Fig. 9a). On the other hand, up-regulated genes in resistant cells were more negatively selected in corresponding chemoresistance screens (Supplementary Fig. 9b), suggesting that some of the up-regulated genes might functionally drive chemoresistance. Furthermore, the genes relating to prominently enriched functional terms for up- or down-

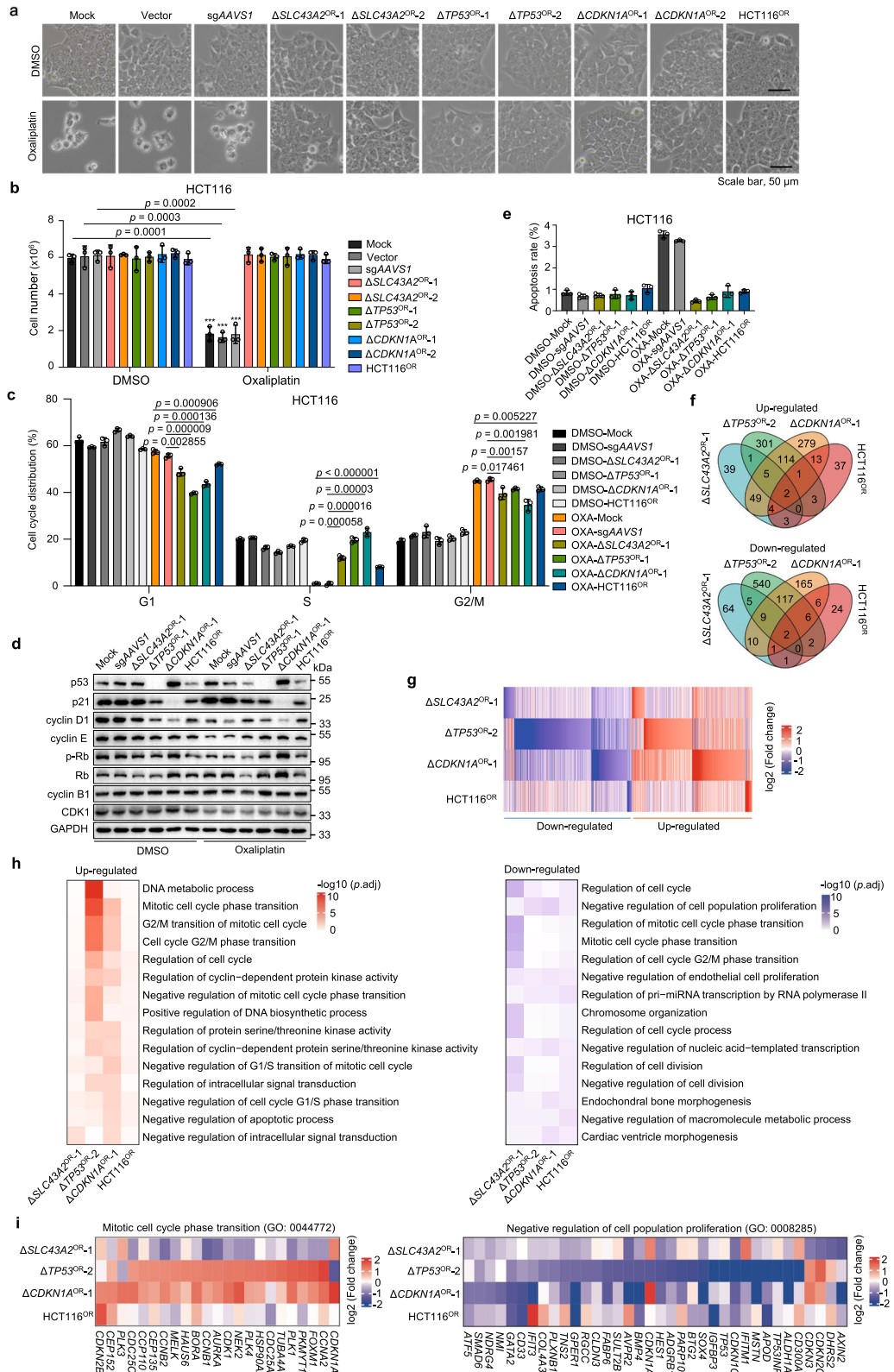

regulated genes in resistant cells tended to be either more negatively selected or positively selected during chemoresistance screen, respectively (Supplementary Fig. 9c, d). These data again established the link between gene expression signatures and the functional effect in driving chemoresistance. In addition, higher expression of the top up-regulated genes in resistant cells were also positively associated with worse colorectal cancer patient survival (Supplementary

Fig. 9e, f), further indicating the clinical relevance of those chemoresistance-related gene expression signatures. Taken together, these results showed that chemoresistance derived from evolutionarily distinct routes could possess diversified molecular features in terms of gene expression profiles but convergent on their functional processes (e.g. cell cycle regulation) pertinent to cellular features of chemoresistant cells.

**Fig. 4 | Cellular and molecular features of oxaliplatin-resistant cells. a** Cell morphology of multiple oxaliplatin-sensitive (Mock, Vector, and sg*AAVS1* groups) and -resistant cell lines in the absence or presence of 5 μM oxaliplatin for 6 days. Scale bar, 50 μm. **b** Cell number quantification of indicated oxaliplatin-sensitive and -resistant cell lines after 7 days of oxaliplatin treatment. Mean ± SD with $n = 3$ biological replicates. Unpaired two-sided $t$ test (oxaliplatin vs. DMSO) for $p$ value. **c** Cell cycle analysis by propidium iodide (PI) staining of indicated cell lines in the absence or presence of 5 μM oxaliplatin for 48 h. OXA, oxaliplatin. Mean ± SD with $n = 3$ biological replicates. Unpaired two-sided $t$ test for $p$ value. **d** Immunoblot analysis of indicated proteins for multiple oxaliplatin-sensitive or -resistant HCT116 cell lines in the absence or presence of 5 μM oxaliplatin for 48 h. **e** Apoptosis analysis by PI and Hoechst staining of indicated cell lines in the absence or presence of 5 μM oxaliplatin for three days. Mean ± SD with $n = 3$ biological replicates. **f** Venn diagram of differentially expressed genes (ΔSLC43A2^OR-1, ΔTP53^OR-1 or ΔCDKN1A^OR-1 vs. sg*AAVS1*; HCT116^OR vs. untreated Mock) in four indicated oxaliplatin-resistant HCT116 cell lines determined by RNA-seq analysis. **g** Heatmap showing differentially expressed genes in four indicated oxaliplatin-resistant HCT116 cell lines. **h** Functional enrichment analysis showing the prominently enriched terms among differentially expressed genes in four oxaliplatin-resistant HCT116 cell lines. Unpaired two-sided $t$ test for $p$ value with Benjamini–Hochberg (BH) adjustment. **i** Highlight of individual genes within indicated functional terms across the four oxaliplatin-resistant cell lines.

## Second-round CRISPR screens for druggable targets against chemoresistance

Although multiple chemoresistant cells displayed diverse molecular features, we still hope to identify convergent vulnerabilities across all the resistant cells towards practical therapeutics. To explore potential druggable targets, we synthesized a new CRISPR knockout library with 12,000 sgRNAs targeting 1,716 druggable genes in the human genome (Supplementary Data 3; See Methods). Each druggable gene corresponds to one or more targeted drugs with inhibitory function to the gene. Using this druggable gene library, we carried out 20 CRISPR knockout screens on four lines of the above established HCT116-based and two additional oxaliplatin-resistant lines of different genetic backgrounds (DLD1^OR and HCT8^OR) together with their parental control sensitive cells (Fig. 5a and Supplementary Fig. 10; Supplementary Data 4). Two biological replicates correlated well for each condition (Supplementary Fig. 11a–j), suggesting a high reproductivity of the screening data. To pinpoint specific drug targets against chemoresistance, we preferentially focused on those druggable genes that were more negatively selected in resistant cells but less negatively or neutrally selected in normal sensitive cells (score^resistant–score^sensitive < −1 & score^resistant < −2 & score^sensitive < 1). Such strategy helps to identify prioritized drugs that are administrated as single-agent regimen to treat recurrent or resistant tumors post chemotherapy. Using such criteria, we identified a cohort of genes across all the six oxaliplatin-resistant cell lines, including a common hit - cell cycle regulator *PLK4* (Fig. 5b, c). Functional enrichment analysis for these druggable hits indicated that targeting "cell cycle" processes might be tentative approaches to preferentially eradicate oxaliplatin-resistant cells (Fig. 5d). Similar six screens and analysis were also performed against irinotecan resistance and tens of druggable genes were shared between at least two irinotecan-resistant cell lines (Supplementary Fig. 12a–c). Functional terms related to "mitochondria" and "cellular respiration" were preferentially enriched for druggable genes antagonizing irinotecan resistance (Supplementary Fig. 12d), which is also in accordance to enriched mitochondria-related genes in first-round irinotecan resistance screen (Fig. 2a). The top consensus druggable genes for irinotecan-resistant cells were highlighted (Supplementary Fig. 12c, e). These results suggest that convergent vulnerabilities indeed exist for evolutionarily distinct chemoresistance, and these druggable targets or pathways identified by such approach provide a tentatively practical choice to apply single-agent regimens to overcome chemoresistance.

## Selective dependency on PLK4 for oxaliplatin-resistant cells

To test the validity of convergent targeting strategy, we preferentially focused on the top hit *PLK4* for its roles and targeting potential against oxaliplatin resistance in colorectal cancer (Fig. 5c). *PLK4* encodes a serine/threonine protein kinase that regulates centriole duplication as well as cytokinesis during the cell cycle[48–51] and is also implicated in resistance to radiotherapy, cisplatin, temozolomide and taxanes[52–55]. Using individual sgRNAs, we specifically knocked out *PLK4* or its homolog gene *PLK1* in HCT116 cells (Fig. 5e). *PLK4* knockout led to more significant cell growth reduction in all the seven oxaliplatin-resistant cell lines compared to three groups of control sensitive cells (Fig. 5f). In contrast, knockout of *PLK1*, recently reported to be associated with chemoresistance[56], only had minor effect in a subset of oxaliplatin-resistant cells (Supplementary Fig. 13a), suggesting that *PLK4* is a more convergent and fundamental vulnerability than *PLK1*. To rule out potential off-target effect and determine whether kinase activity of PLK4 is necessary for this phenotype, we constructed a Cas9-resistant (mutating PAM sequence without changing open reading frame) wild type (WT) and kinase-dead (KD) form (D154A) of PLK4 for rescue experiments. As shown in Fig. 5g, PLK4-WT expression could fully rescue the growth reduction resulting from *PLK4* knockout, whereas PLK4-KD failed to do so, suggesting that the kinase activity of PLK4 is required for its specific cell growth phenotype in resistant cells. Moreover, for three additional oxaliplatin-resistant cell lines using "randomly" clonal selection method based on DLD1, HCT8 and HCT29 parental cells (Supplementary Fig. 10a–c), *PLK4* knockout still significantly decreased cell growth for these resistant cells despite diversified genetic backgrounds (Fig. 5h–j). Moreover, consistent with the results from genetic ablation of *PLK4*, when applying a specific PLK4 inhibitor CFI-400945[57,58], similar cell growth reduction effects were also observed in all the oxaliplatin-resistant cells tested (Fig. 5k–n and Supplementary Fig. 13b). These data demonstrate that catalytically active PLK4 is a convergently vulnerable target for multiple lines of oxaliplatin-resistant cells, despite their varied genetic backgrounds, distinct derivation routes and divergent molecular features.

To explore how PLK4 exerts its function in oxaliplatin-resistant cells, we examined the effect of *PLK4* knockout on cell cycle regulation. A pronounced G2/M phase accumulation was observed upon *PLK4* knockout in resistant cells versus sensitive controls (Fig. 6a). Consistently, CFI-400945 treatment also significantly drove the accumulation of G2/M phase for resistant cells (Fig. 6b). Given that PLK4 can regulate centriole biogenesis and mitosis[48,49], we performed a time-course analysis on mitotic spindle assembly. Interestingly, either *PLK4* knockout or CFI-400945 treatment led to spindle collapse only in resistant cells but not in sensitive control cells (Fig. 6c, d and Supplementary Fig. 14a, b), which was consistent with corresponding cell growth phenotype (Fig. 5f, k). As PLK4 was required for centriole duplication[48,49], centrosomes were significantly depleted in *PLK4* knockout cells for both sensitive and resistant cells (Supplementary Fig. 14c, d). However, CFI-400945 treatment did not reduce centrosome number (Supplementary Fig. 14e), which is consistent with previous reports that relatively low dose range of CFI-400945 can rather increase centriole duplication possibly due to an increase in protein levels of partially active PLK4[58]. These results suggest that centriole biogenesis may not be the key to explain the differential dependency on PLK4 between sensitive and resistant cells. PLK4 could also control spindle assembly in acentrosomal cells through coordination with pericentriolar material (PCM)[59,60]. More importantly, PLK4 was also reported to be essential for cytokinesis[50,51]. Failed spindle assembly and cytokinesis may cause enhanced polyploidy and stalled cell division. Consistent with this hypothesis, we observed increased multi-centrosomes (≥3) in resistant cells than sensitive cells synchronized at G2/M phase upon *PLK4* knockout or CFI-400945 treatment (Fig. 6e, f),

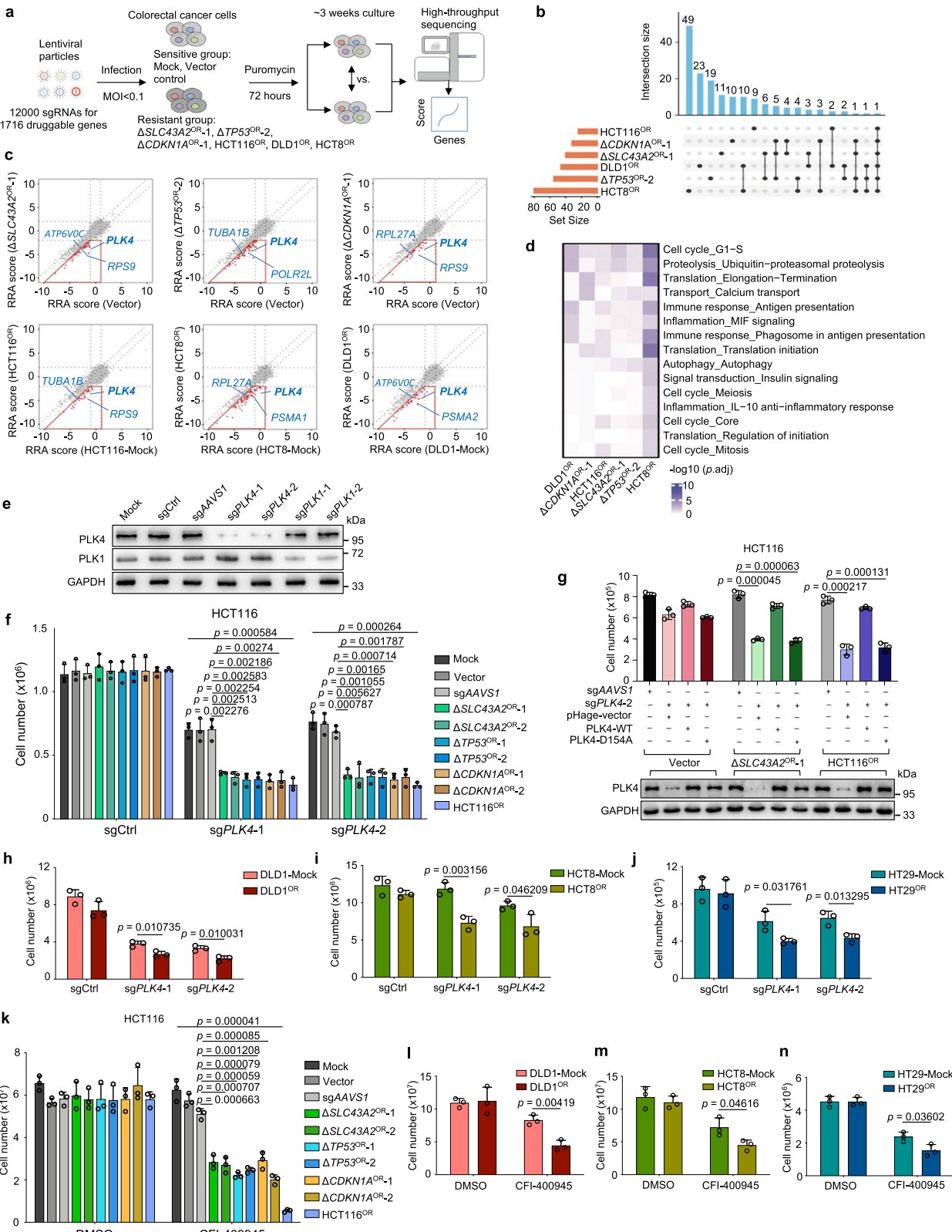

which is possibly due to polyploidy from failed daughter cell separation after DNA duplication. Quantification of >4 N (DNA content) cells provided direct evidence that *PLK4* loss-of-function led to enhanced polyploidy in resistance cells compared to sensitive control cells (Fig. 6g, h). Little effect on apoptosis was observed upon *PLK4* knockout or inhibition in these cells (Supplementary Fig. 14f, g). These data indicate that, compared to control sensitive cells, PLK4-mediated

spindle assembly and cytokinesis control is essentially required specifically for oxaliplatin-resistant cell growth.

We further sought to identify downstream PLK4 target genes by RNA-seq. Upon CRISPR-mediated *PLK4* knockout, differential amount of target genes was identified in both resistant and sensitive control cells (Supplementary Fig. 15a, b and Supplementary Data 5). Interestingly, we did not observe consensus difference for PLK4-regulated

**Fig. 5 | Interrogation of druggable targets against oxaliplatin resistance by second-round CRISPR screens. a** Workflow of identification of potential targets against oxaliplatin resistance using druggable gene CRISPR knockout screens. **b** Venn diagram of preferably druggable gene hits ($\Delta SLC43A2^{OR}$-1, $\Delta TP53^{OR}$-2 or $\Delta CDKN1A^{OR}$-1 vs. Vector; HCT116$^{OR}$, HCT8$^{OR}$, DLD1$^{OR}$ vs. corresponding untreated Mock) in six oxaliplatin-resistant cell lines. **c** The RRA scores of all the tested genes for oxaliplatin-sensitive control or -resistance cells in each comparison. The genes in red box are preferential targets against resistance. **d** The prominently enriched functional terms for druggable gene sets against oxaliplatin in six resistant cell lines. Unpaired two-sided t test for p value with Benjamini-Hochberg (BH) adjustment. **e** Immunoblot analysis validating the knockout effect of PLK4 or PLK1. sgCtrl, empty control sgRNA vector. **f** Cell growth effect of *PLK4* knockout in oxaliplatin-sensitive or -resistant cell lines. Mean ± SD with n = 3 biological replicates. Unpaired two-sided t test ($\Delta SLC43A2^{OR}$, $\Delta TP53^{OR}$ or $\Delta CDKN1A^{OR}$ vs. sg*AAVS1*; HCT116$^{OR}$ vs. Mock) for p value. **g** Expression of PLK4-WT but not kinase-dead mutant PLK4-D154A rescues the growth inhibition caused by *PLK4* knockout in oxaliplatin-resistant cells. Mean ± SD with n = 3 biological replicates. Unpaired two-sided t test for p value. **h–j** Cell growth inhibition by two independent sgRNAs targeting *PLK4* in oxaliplatin-resistant DLD1 (**h**), HCT8 (**i**) and HT29 (**j**) cells. Mean ± SD with n = 3 biological replicates. Unpaired two-sided t test for p value. **k** Cell growth effect of indicated cell lines treated with DMSO or 12 nM CFI-400945 for 7 days. Mean ± SD with n = 3 biological replicates. Unpaired two-sided t test ($\Delta SLC43A2^{OR}$, $\Delta TP53^{OR}$ or $\Delta CDKN1A^{OR}$ vs. sg*AAVS1*; HCT116$^{OR}$ vs. Mock) for p value. **l–n** Cell growth inhibition of DLD1 (**l**), HCT8 (**m**) and HT29 (**n**) cells by 10 nM CFI-400945 for 11 days, 15 nM CFI-400945 for 11 days, and 15 nM CFI-400945 for 10 days, respectively. Mean ± SD with n = 3 biological replicates. Unpaired two-sided t test for p value.

targets between resistant and sensitive groups. Rather, $\Delta SLC43A2^{OR}$-1 sample resembled $\Delta TP53^{OR}$-1 group, and sensitive vector cells were more similar to the resistant $\Delta CDKN1A^{OR}$-1 cells (Supplementary Fig. 15a, b). Functional enrichment analysis showed that RNA processing-related genes were enriched in PLK4 up-regulated targets for all cell types (Supplementary Fig. 15c). When focusing on specific enrichment only in resistant cells, more basic terms such as "ribosome", "proteasome" and "cell cycle" or functions related to "cell adhesion" appeared (Supplementary Fig. 15d, e), consistent with pronounced vulnerability to PLK4 inhibition for resistant cells. Previous studies showed that PLK4 inhibition causes synthetic lethality in cancer cells with genomic amplification or overexpression of the centrosomal ubiquitin ligase TRIM37 which is a negative regulator of PCM[59–62]. Compromised PCM failed to proceed acentrosomal spindle assembly and led to mitotic failure in cells with centrosome depletion caused by PLK4 inhibition. Indeed, we observed that several PCM scaffolding genes (e.g., *TUBG1* and *CEP192*) were de-regulated in *PLK4* ablated oxaliplatin-resistant cells (Supplementary Fig. 15f), which further supports our speculation that PLK4-regulated PCM might be more critical than centriole biogenesis in distinguishing the spindle collapse phenotype among oxaliplatin-resistant and -sensitive cells.

## Mechanistic insights on oxaliplatin resistance and PLK4 dependency

Through a combinatorial effort including CRISPR screen, cellular and molecular characterization of multiple oxaliplatin-resistant models, we gained some preliminary insights on how colorectal cancer develops oxaliplatin resistance (Fig. 6i). In sensitive cells, the formation of oxaliplatin-DNA adduct creates DNA lesions that can be sensed by p53. The activated p53-p21 pathway then induces both G1 and G2/M cell cycle arrest, thereby blocking cell growth of oxaliplatin-sensitive cells. Once the cells undergo certain genetic or epigenetic alterations that block the function of p53-p21 pathway (e.g., knockout of *TP53*, *CDKN1A* or *SLC43A2*), the cells become resistant to oxaliplatin treatment due to unconstrained cell cycle progression. Notably, other mechanisms such as altered drug transport, detoxification, DNA repair and cell death were reported to underlie oxaliplatin resistance in various contexts including colorectal cancer[63]. Our results here favored a cell cycle-centered theory as the primary mechanism to explain oxaliplatin resistance in colorectal cancer. As cells become resistant to oxaliplatin, they may also bear certain special vulnerabilities that are conferred concomitantly by those causal alterations. Indeed, multiple oxaliplatin-resistant cells are accordingly more dependent on PLK4 function to complete cell division in the presence of oxaliplatin-induced DNA lesions. Consistently, p53 was reported to negatively regulate PLK4 expression during stress response[64]. Either genetic ablation or pharmaceutical inhibition of PLK4 can lead to failed spindle assembly and cytokinesis preferentially in resistant cells (Fig. 6i). Such vulnerability

creates an opportunity to apply single-agent PLK4-targeting drug for the treatment of oxaliplatin-resistant colorectal cancers.

## Targeting PLK4 in xenograft model and clinical samples

To further examine the therapeutic potential of targeting PLK4 to antagonize oxaliplatin resistance, we set up a tumor xenograft model in mice with oxaliplatin-responsive (vector) or -resistant (HCT116$^{OR}$) HCT116 cells and monitored in vivo tumor growth upon oxaliplatin or CFI-400945 treatment. As shown in Supplementary Fig. 16a–d, HCT116$^{OR}$ xenograft indeed showed resistance to oxaliplatin in vivo while tumor growth of sensitive group can be significantly repressed by the same oxaliplatin treatment. In contrast, the oxaliplatin-resistant tumors (HCT116$^{OR}$) displayed remarkable growth repression upon CFI-400945 treatment compared to medium response of control sensitive tumors (Fig. 7a–c and Supplementary Fig. 16e).

In human tumors, *PLK4* is frequently mutated in many types of cancer, including colorectal cancer (Supplementary Fig. 17a). Tumors bearing mutant *PLK4* tend to have higher tumor mutational burden and are usually associated with advanced stages of tumor progression in colorectal cancer (Supplementary Fig. 17b, c). Increased RNA expression of *PLK4* was found in colorectal tumor samples compared to normal tissues as evidenced in two independent clinical datasets (Supplementary Fig. 17d, e). Due to the difficulty to obtain defined oxaliplatin-resistant clinical samples, we employed tumor tissues resected from colorectal cancer patients with or without neoadjuvant chemotherapy and successfully established seven patient-derived organoid (PDO) models (Supplementary Fig. 18a, b, see Methods). Consistent with RNA expression pattern shown above, PLK4 protein levels were also significantly higher in these tumor tissues than matched adjacent normal counterparts (Fig. 7d). In contrast, PLK1 did not display apparent tumor-specific expression pattern (Fig. 7d). Next, we tested ex vivo drug responses of these PDOs to oxaliplatin and PLK4 inhibitor CFI-400945. As shown in Fig. 7e, f, C-1 PDO was sensitive to oxaliplatin treatment at 1 μM concentration whereas C-5 PDO was resistant to oxaliplatin even at 20 μM concentration. Conversely, C-5 growth was significantly inhibited by a low concentration of CFI-400945 (5 nM) while C-1 resisted growth inhibition from a high concentration (40 nM) of CFI-400945 treatment. Such inverted response indicated that oxaliplatin-resistant tumor was more sensitive to CFI-400945, and vice versa. To extend this observation, we plotted the growth inhibition status of each data point for all the organoids and observed a clear anti-correlation pattern for drug responses to oxaliplatin versus CFI-400945 (Fig. 7g and Supplementary Fig. 18c). Moreover, we found that PLK4 expression was relatively higher in oxaliplatin-refractory tumors (C-5, C-6 and C-7) than those responsive ones (C-1, C-2 and C-3) (Fig. 7d and Supplementary Fig. 18d), which might partially explain the efficacy of targeting PLK4 in chemoresistant samples. Collectively, these data suggest that single-agent administration with PLK4 inhibitor CFI-400945 may be a promising

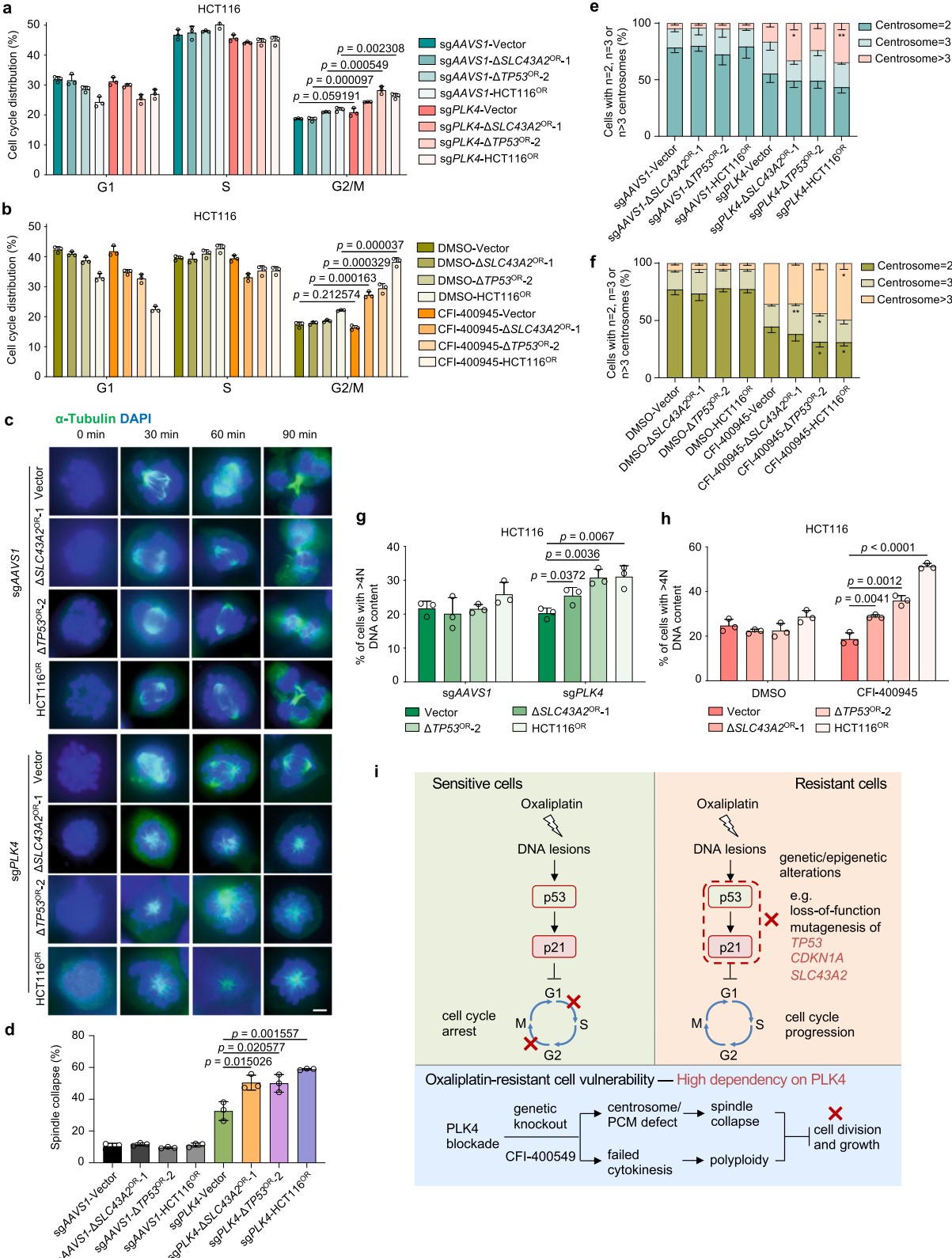

therapeutic strategy to overcome oxaliplatin-involved chemoresistance in colorectal cancer.

## Discussion

How cancer cells resist to chemotherapeutic agents and how to overcome chemoresistance clinically remain largely elusive. Here we systematically reveal the genetic causes of chemoresistance for multiple chemotherapy drugs across different types of cancer, elucidate diversified routes and characteristics of chemoresistance, and propose useful biomarkers and therapeutic strategies against chemoresistant cancers. Our study not only provides valuable resources to better understand the molecular basis behind chemoresistance in cancers, but also demonstrates promising strategies to predict and overcome such resistance in clinics.

**Fig. 6 | Selective dependency on *PLK4* for oxaliplatin-resistant cells. a** Cell cycle analysis upon *PLK4* knockout in oxaliplatin-sensitive and -resistant cells. Mean ± SD with *n* = 3 biological replicates. Unpaired two-sided *t* test for *p* value. **b** Cell cycle analysis by BrdU and PI staining upon CFI-400945 treatment in oxaliplatin-sensitive and -resistant cells. Mean ± SD with *n* = 3 biological replicates. Unpaired two-sided for *p* value. **c** *PLK4* knockout results in spindle collapse, folding, and slippage in oxaliplatin-resistant cells. Scale bar, 5 μm. **d** Quantification of spindle collapse for (**c**). Mean ± SD with *n* = 3 biological replicates, 100–200 cells were analyzed per independent experiment. Unpaired two-sided *t* test for *p* value. **e, f** Distribution of cell populations with indicated centrosome number according to γ-tubulin staining as shown in Supplementary Fig. 14c before or after *PLK4* knockout (**e**) or CFI-400945 treatment (**f**) for oxaliplatin-sensitive and -resistant cells synchronized at G2/M phase. Mean ± SD with *n* = 3 biological replicates, 100–200 cells were analyzed per independent experiment. Unpaired two-sided *t* test, sg*PLK4*-Vector vs. sg*PLK4*-Δ*SLC43A2*$^{OR}$-1, Centrosome >3, \**p* = 0.013517; sg*PLK4*-Vector vs. sg*PLK4*-ΔHCT116$^{OR}$, Centrosome >3, \*\**p* = 0.009327; CFI-400945-Vector vs. CFI-400945-Δ*SLC43A2*$^{OR}$-1, Centrosome = 3, \*\**p* = 0.007507; CFI-400945-Vector vs. CFI-400945-Δ*TP53*$^{OR}$-2, Centrosome = 2, \**p* = 0.026611; Centrosome = 3, \**p* = 0.031089; CFI-400945-Vector vs. CFI-400945-ΔHCT116$^{OR}$, Centrosome >3, \**p* = 0.01103; Centrosome = 2, \**p* = 0.017076. **g, h** Quantification of cells with >4 N DNA content (polyploidy) upon *PLK4* knockout (**g**) or CFI-400945 treatment (**h**) in oxaliplatin-sensitive and -resistant cells. Mean ± SD with *n* = 3 biological replicates. Unpaired two-sided *t* test for *p* value. **i** Schematic model illustrating the molecular mechanisms about oxaliplatin resistance in colorectal cancer and the PLK4 dependency of oxaliplatin-resistant cells.

Chemoresistance has been linked to a number of reported mechanisms, including reduced drug uptake, increased drug efflux, drug-target alteration, drug inactivation, altered DNA repair mechanism and impaired cell death signaling. In addition, intratumor heterogeneity, cell-cell interaction, cell lineage transition, remodeling of tumor microenvironment and epithelial-mesenchymal transition (EMT) also contribute to chemoresistance[21,65–68]. Such resistance is either derived intrinsically from pre-existing resistant clones before treatment or acquired after therapy[17–20]. Both genetic and epigenetic factors such as DNA mutation or gene expression in tumor cells underlie the development of cell-autonomous chemoresistance[16,69]. Our systematic investigation through functional genomics revealed a wealth of information on the known and unknown drivers and mechanisms underlying chemoresistance in solid tumors at an unprecedented scale, which greatly deepens our understanding on cell-autonomous chemoresistance. We found that aberrant changes in cell cycle, DNA damage response and tumor microenvironment remodeling represent the major mechanisms for genetic alterations to drive multidrug resistance, complementing previous theories about multidrug resistance which primarily underline drug transport and metabolism, DNA repair and cell death signaling[16,69]. Some pronounced drug-specific mechanisms also emerged. For instance, we showed that deregulation of vitamin metabolism which connects to drug processing and inactivation is the primary driving force for the resistance of antimetabolite drug 5-fluorouracil. Several top resistance genes (e.g., *MTR*, *MTRR*, and *MMACHC*) in our screens control the activity of methionine synthase and thereby affect folate metabolism which in turn modulates the metabolism and activity of 5-fluorouracil[70,71]. We also unraveled that mitochondria-related process, among other mechanisms, is strongly associated with irinotecan resistance. The importance of mitochondria during chemoresistance has been linked to apoptotic pathways[72] while other functions such as energy control and metabolic rewiring may also be involved. Interestingly, for drugs sharing the same mechanism of action such as docetaxel and paclitaxel that both target microtubule dynamics, we observed significant difference between their general mechanisms towards resistance although the top genetic drivers and functions are still shared. Paclitaxel resistance is primarily associated with cellular senescence, while docetaxel resistance is more relevant to broader processes involved in multidrug resistance. Such disparity could be explained by their elaborate difference of pharmacological profiles and the incomplete cross-resistance between the two taxanes which are also confirmed in several tumor types clinically[73]. These systematic insights retrieved from our chemogenomic screens further substantiate the ways by which chemoresistance may develop.

The findings that chemoresistance may arise from divergent routes of genetic alterations with varied cellular and molecular features pose great challenge for predicting and treating such resistance. We also found that multidrug resistance tends to occur by the same sets of chemoresistance genes, which makes chemoresistant tumors quite difficult to be eradicated or controlled. More importantly, genetic backgrounds greatly diverge the evolutionary routes towards resistance, which means different patients may develop chemoresistance via distinct mechanisms, further complicating the choice of clinical intervention. Despite tremendous efforts on characterizing specific resistance mechanisms and developing targeting strategies accordingly[16,21,65–69,74,75], the lack of effective biomarkers and prior knowledge of resistance mechanisms on the individual tumors still impedes the application of patient-specific precision medicine at current stage. Therefore, seeking convergent vulnerabilities, if any, for evolutionarily distinct chemoresistance represents another important and innovative approach to develop practical therapeutics against chemoresistance. The identification of significant chemoresistance genes from our screens allowed us to establish multiple lines of resistant cells far more efficiently than traditional ways, which greatly accelerates characterization of chemoresistance. With second-round of druggable gene screens on the established resistant cells, we could identify potentially vulnerable targets preferentially against chemoresistance. It is interesting to see that consensus drug targets indeed exist for different resistant cells derived from evolutionarily distinct paths. These findings suggest that it might be a promising strategy to use a convergent regimen with single-agent, rather than complicated combination therapy, to conquer chemoresistance resulting from diversified genetic alterations or molecular pathways.

As an example, we validated PLK4 as an actionable target against oxaliplatin resistance in colorectal cancer and explored the potential of using PLK4 inhibitor CFI-400945 to treat oxaliplatin-resistant tumors. Multiple clinical trials (NCT01954316, NCT03187288, NCT03624543, NCT04176848 and NCT04730258) were initiated to evaluate CFI-400945 for treating breast cancer, myeloid leukemia and myelodysplastic syndrome. Results from a phase I trial have indicated good safety profile of this candidate drug in patients with advanced solid tumors[76]. Therefore, CFI-400945 represents one of the PLK4 inhibitors that is readily translated into real clinics and will be thoroughly evaluated in the follow-up clinical trials to treat oxaliplatin-resistant cancers.

Taken together, our study not only provides a wealth of valuable information on the biomarkers, genetic drivers, mechanisms and intervention targets for chemoresistant cancers, but also reveals convergent targeting strategies against evolutionarily distinct chemoresistance. Such efforts systematically enhance our knowledge on chemoresistance and suggest practical strategies towards clinical management of chemoresistant cancers. Moreover, our approach employing tandem CRISPR screens can also be applied to broader scenarios related to drug resistance.

There are several limitations of our current study. First, our study design focuses on chemoresistance mainly driven by cell-autonomous effects, while other types of chemoresistance involving tumor microenvironment and cell-cell interaction are not covered. Second, the chemogenomic screens were solely based on in vitro culture of specific cell lines as 2D monolayers. Further screens in 3D spheroid and in vivo mouse models may produce additional discoveries and insights. Third,

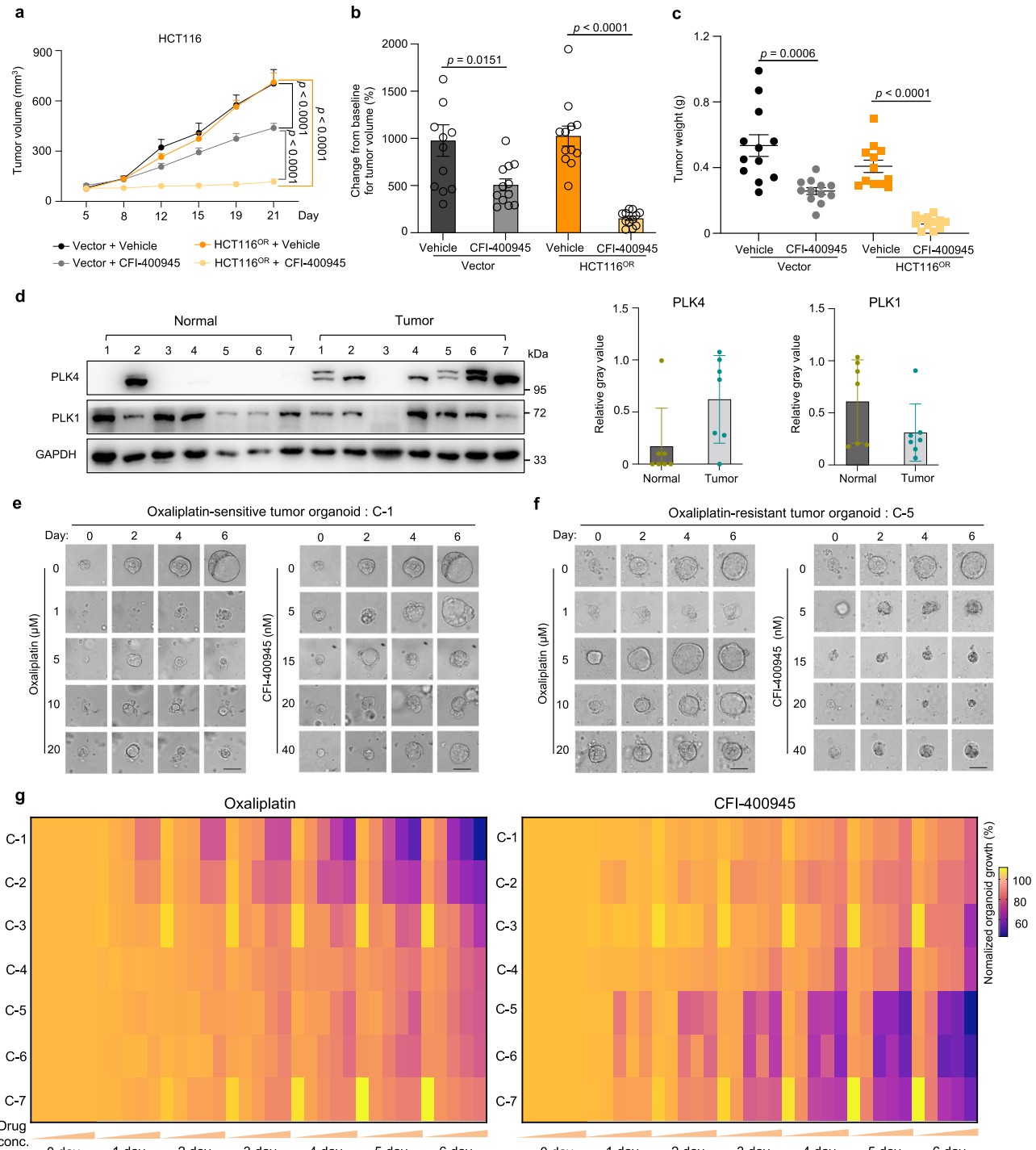

**Fig. 7 | Validation of efficacy of the PLK4 inhibitor CFI-400945 against oxaliplatin resistance in mice and at clinical levels. a** Tumor volume measured at indicated time points after xenograft implantation for oxaliplatin-sensitive (Vector) or -resistant (HCT116^OR) xenograft treated with vehicle or CFI-400945. (*n* = 12, number of tumor). Mean ± SEM. Two-way ANOVA for *p* value. **b–c** Relative tumor volume (compared to day 0) (**b**) or tumor weight (**c**) of mouse xenografts for indicated HCT116 cells measured at Day 21 post implantation. (*n* = 12, number of tumor). Mean ± SEM. Unpaired two-sided t test for *p* value. **d** Immunoblot of PLK4

and PLK1 in tumor or adjacent normal tissues from colorectal cancer patients (#C-1 to #C-7) (left). Quantification of relative protein expression from the immunoblot image (right), Mean ± SD with *n* = 7. **e, f** Representative organoid growth in response to oxaliplatin or CFI-400945. C-1 (**e**) is sensitive to oxaliplatin, while C-5 (**f**) is resistant to oxaliplatin. Scale bar, 50 μm. **g** Anti-correlation pattern of relative viability of all the seven tumor-derived organoids (C-1 to C-7) in response to oxaliplatin (left: 0, 1, 5, 10, and 20 μM oxaliplatin) or CFI-400945 (right: 0, 5, 15, 20, and 40 nM CFI-400945).

we just applied loss-of-function CRISPR screens in this study. Next phase of the study should move to base-editing screen to better interrogate cancer gene mutation during chemoresistance. Forth, as a start point, we preferentially examined chemoresistance effects to each single drug, while drug combinations are often used in clinical

context and dosing effects may also affect the drug responses. More advanced models and detailed experimental design which better mimic clinical scenarios will help to resolve these questions. Last, as a balance between scope coverage and individual points, the detailed mechanisms behind each chemoresistance gene cannot be fully

explored in the current study. Additional efforts are needed to fill these gaps. Thorough evaluation of these potential biomarkers and therapeutic interventions holds promise to benefit patients suffering chemoresistance.

# Methods

## Ethical statement

All animal experimental procedures were performed according to the Guidelines for the Care and Use of Laboratory Animals and were approved by the Biological and Medical Ethics Committee of Northeastern University (#NEU-EC-2021A020S). Ethical approval regarding to human tissue collection was obtained from the ethical committee of The Sixth Affiliated Hospital of Sun Yat-sen University (2021ZSLYEC-466).

## Cells

HEK293FT (Cat# CRL-1573), HCT116 (Cat# CCL-247), DLD1 (Cat# CCL-221), T47D (Cat# HTB-133), MCF7 (Cat# HTB-22), A549 (Cat# CRM-CCL-185), NCI-H1568 (Cat# CRL-5876), HCT8 (Cat# CCL-244) and HT29 (Cat# HTB-38) cells were obtained from the American Type Culture Collection (ATCC). All cells were regularly tested negative for mycoplasma contamination and maintained in DMEM (for HEK293FT, HCT116, T47D, MCF7, A549 and HT29 cells) or RPMI 1640 (for DLD1, NCI-H1568 and HCT8 cells) medium with 10% fetal bovine serum plus 1% penicillin/streptomycin at 37 °C with 5% $CO_2$.

## Mice

Around 6-week-old SFP-BALB/cA-nu male mice were purchased from Beijing HFK Bioscience Co.,Ltd. (Beijing, China). All mice were fed in standard individual ventilated cages, and maintained with 12 h: 12 h light cycle, 24–26 °C room temperature and 40–60% relative humidity.

## Human tissues

Tumor tissues were obtained from colorectal cancer patients during surgery with informed consent. Adjacent normal tissues were obtained from resected colorectal segments with tumors. The isolation of tumor epithelium was performed essentially as described previously[77]. Tumor epithelium was washed with cold PBS until the supernatant was clear. Tumor tissue was cut into small pieces and incubated with collagenase IV for 30 min at 37 °C. Filtrate the mixture with a 70 μm cell strainer to remove large tissue debris. Cells were resuspended in Matrigel and seeded on well plates for later use.

## Patient-derived organoids

Tumor tissue-derived organoids were derived as described previously[78]. Place fresh tumor tissue from colorectal patient into cold PBS (PBS+1×Primocin™), remove non-cancerous tissue such as fat, and wash 3 times. Tumor tissue was cut into small pieces (approximately 2 mm) and incubated with collagenase IV (4 mg/mL) and Y27632 (10 μM) for 30 min at 37 °C while shaking (less than 20 × g). After incubation, complete medium (Advanced DMEM/F-12 with 1× penicillin/streptomycin, 1× HEPES, 1× GlutaMAX, 1× B-27 Supplement, 1× N-2 Supplement, 1 mM N-acetylcysteine, 50 ng/ml EGF, 10 mM Nicotinamide, 500 nM A-83-01, 10 μM SB202190, 10 nM Gastrin1, and RWN CM) was added (3-4 times the volume of collagenase to terminate digestion) and the filtrate mixture was placed on a 70 μm cell strainer to remove large fragments. Cells were subsequently spun at 180 × g at 4 °C for 3 min. The pellet was resuspended in complete medium and centrifuged again at 180 × g, 4 °C for 3 min. This procedure was repeated twice to remove debris and collagenase. Cells were resuspended in Matrigel and seeded on 96-well or 24-well plates. After the Matrigel becomes solidified, add pre-warmed commercial totipotent medium (STEMCELL # 06010) and incubate the cells at 37 °C. The organoids were cultured with 1% penicillin/streptomycin and 10 μM Y27632 at 37 °C with 5% $CO_2$. The self-made medium was also used. The

composition of self-made medium is as follows: basal culture medium (Advanced DMEM/F-12) with 1× penicillin/streptomycin, 1× HEPES, 1× GlutaMAX, 1× B-27 Supplement, 1× N-2 Supplement, 1 mM N-acetylcysteine, 50 ng/ml EGF, 10 mM Nicotinamide, 500 nM A-83-01, 10 μM SB202190, 10 nM Gastrin1, and RWN CM (12 mL / 20 mL). RWN CM represents DMEM medium containing R-Spondin3, Wnt3A and Noggin. RWN was secreted by L-RWN cells which was kindly provided by Prof. Ren Sheng at Northeastern University. To visualize the morphology and architecture, patient-derived organoids cultured in 96-well plate for around one week post isolation were washed twice with PBS. Aspirate PBS, and add 100 μL/well paraformaldehyde (Servicebio #G1101) for 30 min. Aspirate paraformaldehyde and washed three times with PBS for 5 min each time. Then, 100 μL of the nuclear dye Hoechst 33342 (Biosharp # BL803A) was added for 5 min at room temperature in the dark. Aspirate Hoechst 33342 and wash three times with PBS for 5 min each. The 96-well plate was placed in the Opertta CLS High Content Analysis System (PerkinElmer) to acquire and analyze 2D and 3D images of organoids. H&E staining of parental tumor tissues for each organoid was performed by pathologist in clinics. To quantify organoid growth in response to oxaliplatin or CFI-400945 treatment, organoid number were counted from the photographs taken by the Opertta CLS High Content Analysis System (PerkinElmer) at indicated time point in quadruplicate.

## Lentivirus production

Lentiviruses were generated by co-transfecting lentivector plasmids of interest together with packaging vectors pCMVR8.74 and pMD2.G using Opti-MEM (Gibco) and Lipofectamine 2000 reagent (Invitrogen) in HEK293FT cells. Viral supernatants were collected at 48 hour (h) or 72 h post transfection and centrifuged at 1580 × g for 5 min. The viral supernatants collected were subsequently aliquoted and stored at −80 °C before use.

## First-round genome-wide CRISPR knockout screen

The pooled genome-wide CRISPR knockout library (Addgene, #1000000132) was kindly provided by Prof. X. Shirley Liu and Prof. Myles Brown at Dana-Farber Cancer Institute. We used the h1 part of the half library which contains 92,817 gRNAs targeting 18,436 genes (5 gRNAs per gene) in the human genome and is constructed under lentiCRISPRv2-puro backbone. Pooled lentiviruses encapsulating Cas9 and conjugated gRNAs in the library were produced in HEK293FT cells. HCT116, DLD1, T47D, MCF7, A549 and NCI-H1568 cells were infected with differential amount of lentivirus to maintain a low MOI (< 0.3) during screening. After 48 h, infected cells were selected with puromycin (1, 3.5, 3.5, 2, 1, and 2 μg/ml, respectively) for 3 days followed by recovery for additional two days. Seven days post infection (Day 0, start point of screen), cells were treated with DMSO or indicated chemotherapy drugs (oxaliplatin, irinotecan, 5-fluorouracil, doxorubicin, cisplatin, docetaxel and paclitaxel) for corresponding cell lines. The concentration of drugs for screening was empirically determined by integrative consideration of $IC_{50}$ values and cell survival status in the duration timeline of the screens. The details are as follows: HCT116-oxaliplatin (5 μM), HCT116-irinotecan (10 μM), HCT116-5-fluorouracil (10 μM), DLD1-oxaliplatin (20 μM), DLD1-irinotecan (10 μM), DLD1-5-fluorouracil (10 μM), MCF7-paclitaxel (0.01 μM), MCF7-docetaxel (0.01 μM), MCF7-cisplatin (60 μM), MCF7-5-fluorouracil (20 μM), T47D-paclitaxel (0.01 μM), T47D-docetaxel (0.005 μM), T47D-cisplatin (60 μM), T47D-5-fluorouracil (5 μM), T47D-doxorubicin (0.03 μM), A549-paclitaxel (0.02 μM), A549-docetaxel (0.005 μM), A549-cisplatin (20 μM), A549-5-fluorouracil (5 μM), A549-doxorubicin (0.05 μM), NCI-H1568-paclitaxel (0.01 μM), NCI-H1568-docetaxel (0.005 μM), NCI-H1568-5-fluorouracil (5 μM) and NCI-H1568-doxorubicin (0.02 μM). As cells had varied response to drugs, cell fitness selection by drug challenge lasted for around 9 - 21 days, depending on selection strength, amount of surviving cells and cell passages, before

harvesting samples at the end of screen. The details of the drug treated days are as follows: HCT116-oxaliplatin (21 days), HCT116-irinotecan (21 days), HCT116-5-fluorouracil (21 days), DLD1-oxaliplatin (14 days), DLD1-irinotecan (14 days), DLD1-5-fluorouracil (21 days), MCF7-paclitaxel (12 days), MCF7-docetaxel (12 days), MCF7-cisplatin (12 days), MCF7-5-fluorouracil (15 days), T47D-paclitaxel (20 days), T47D-docetaxel (9 days), T47D-cisplatin (17 days), T47D-5-fluorouracil (9 days), T47D-doxorubicin (12 days), A549-paclitaxel (21 days), A549-docetaxel (21 days), A549-cisplatin (21 days), A549-5-fluorouracil (21 days), A549-doxorubicin (21 days), NCI-H1568-paclitaxel (21 days), NCI-H1568-docetaxel (21 days), NCI-H1568-5-fluorouracil (21 days) and NCI-H1568-doxorubicin (21 days). At least 300X coverage of cells were collected for samples at Day 0 and the end of screen. Genomic DNA was then extracted from these samples and the sgRNA fragment was PCR-amplified. High-throughput sequencing (PE150) was performed (Novogene) to determine the abundance of the sgRNAs. The MAGeCK algorithm was employed to analyze the data.

### Genomic DNA extraction and sequencing
For large number of cells in the screens, cells were collected in 15 mL conical tubes. Add 4 mL lysis buffer (300 mM NaCl, 0.2% SDS, 2 mM EDTA, 10 mM Tris-HCl pH 8.0) in a 15 mL conical tube as well as 40 μL RNase A (10 mg/ml). Incubate tubes at 65 °C for 1 h with rotation. Add 40 μL proteinase K (10 mg/ml) and incubate tubes at 55 °C overnight (or 6 h). Add 4 mL phenol/chloroform/isoamyl alcohol solution (25:24:1), mix the samples and centrifuge at $3560 \times g$ for 15 min. Take the supernatant and mix with 1 volume of isopropanol to precipitate genomic DNA (gDNA). Spin and wash 2 times by 75% ethanol. Add 1 mL clear water (nuclease-free) to dissolve gDNA and measure the concentration using Nanodrop. For low or medium number of cells during indel assays, use 1.5 mL centrifuge tubes during gRNA extraction with reduced quantity of reagents and similar procedures as mentioned above.

To construct sequencing libraries for quantification of sgRNA abundance, two rounds of PCR were performed to amplify sgRNA region from gDNA and ligate adaptor sequences for Illumina platform sequencing. For the first-round PCR, perform 25–30 separate 100 μL reactions with 6–8 μg genomic DNA in each reaction using Q5 High-Fidelity DNA Polymerase (New England Biolabs) for ~18–20 cycles and then combine the resulting amplicons. The primers (Supplementary Data 6) used for first-round PCR are as follows: lentiCRIPR_1st_Forward: 5′-AATGGACTATCATATGCTTACCGTAACTTGAAAGTATTTCG-3′. lenti CRISPR_1st_Reverse: 5′-GGAGTTCAGACGTGTGCTCTTCCGATCTCCAG TACACGACATCACTTTCCCAGTTTAC-3′. The second-round PCR is to attach Illumina adaptors and barcode samples. Perform the second-round PCR in a 100 μL reaction volume using 1 μL of the product from the first-round PCR for around 10–12 cycles. Primers for the second-round PCR are as follows: lentiCRIPR_2nd_Forward: 5′-AATGAT ACGGCGACCACCGAGATCTACACTCTTTCCCTACACGACGCTCTTCC GATCTATCTTGTGGAAAGGACGAAACACC-3′; lentiCRIPR_2nd_Index_ Reverse: 5′-CAAGCAGAAGACGGCATACGAGATNNNNNNNNGTGACTG GAGTTCAGACGTGTGCTCTTCCGATCT-3′ (N(8) are the specific index sequences). Purify PCR product using 1.2% agarose gel and Gel Purification Kit before proceeding to high-throughput sequencing on Illumina PE150 platform (Novogene).

### Design and construction of druggable gene CRISPR knockout library
To design human druggable gene CRISPR knockout library, we selected 1716 druggable genes with well-defined drug-target interactions (6236 associated drugs) included in DGIdb3.0[79]. In addition, we also included 106 known "core essential genes" from published resources, whose perturbations are known to have strong effects on cell proliferation or viability[80,81]. We then extracted the corresponding sgRNA sequences and annotations for these genes from the Human CRISPR Knockout Library H3 (contributed by Profs. X. Shirley Liu and Myles

Brown, Addgene pooled library #133914). On average, 6 guides are designed per gene. In addition, 496 AAVS1- or ROSA26-targeting sgRNAs and 495 non-targeting sgRNAs were selected from the H3 library, which were not considered to have functional impact. In total, 12,000 sgRNAs were included for this druggable gene CRISPR knockout library. The oligos with flanking sequences were synthesized in a pooled format (Synbio Technologies).

Pooled synthetic oligonucleotides were PCR-amplified and cloned into the lentiCRISPRv2-puro vector (expressing Cas9 and sgRNA cassette simultaneously) by Gibson Assembly via the BsmB I site. The ligation mix was transformed into electrocompetent stable E. coli cells by electroporation to reach the efficiency with at least 100X coverage. Transformed bacteria were grown in liquid LB medium for 16–20 h at 30 °C to minimize recombination events in E. coli. The library plasmids were then extracted with the EndoFree Maxi Plasmid Kit (Tiangen, Cat# 4992194).

### Second-round CRISPR screen with druggable gene library
Pooled lentiviruses encapsulating Cas9 and conjugated gRNAs in the druggable gene library were produced in HEK293FT cells. For oxaliplatin-resistant models, the screens were performed on four lines of control sensitive cells (untreated HCT116-Mock and Vector; DLD1-Mock and HCT8-Mock), and four lines of oxaliplatin-resistant HCT116 cells (ΔSLC43A2^OR-1, ΔTP53^OR-2, ΔCDKN1A^OR-1 and HCT116^OR) as well as two additional oxaliplatin-resistant lines (DLD1^OR, and HCT8^OR). Two biological replicates were performed for second-round CRISPR screen using oxaliplatin resistance models. For irinotecan resistance models, the screens were performed on two lines of control sensitive cells (untreated DLD1-Mock and Vector) and four lines of irinotecan-resistant DLD1 cells (ΔBEND3^IR-1, ΔKEAP1^IR-2, ΔSGF29^IR-2 and DLD1^IR). These base cells were created by stably expressing indicated gRNAs with blasticidin resistance. For screening, these cells were infected with pooled lentiviruses of druggable gene library at a low MOI (<0.1). After 48 h, infected cells were selected with puromycin (2 and 3.5 μg/ml for HCT116 and DLD1 cells, respectively) for 3 days followed by recovery for additional two days. The resulting cells (Day 0, start of screen) were continually cultured in normal media for about three weeks (end of screen) to allow fitness selection. At least 300X coverage of cells were collected for samples at Day 0 and the end of screen. Genomic DNA was then extracted for these samples and sgRNA fragment was PCR-amplified. High-throughput sequencing (PE150) was performed (Novogene) to determine the abundance the sgRNAs.

### Individual gene knockout by CRISPR-Cas9
Two independent sgRNAs (Supplementary Data 6) were designed for each target gene and cloned into lentiCRISPRv2-blast vector (Addgene #83480) or lentiCRISPRv2-puro vector (Addgene #52961) which expresses Cas9 and sgRNA simultaneously once transduced into target cells. Lentiviruses were produced in HEK293FT cells and then used to infect target cells. After 48 hours, cells were selected by blasticidin for 3-5 days or puromycin for 3 days and the resulting cells were amplified for downstream analysis. To determine the knockout efficiency, indel assay was performed by T7 Endonuclease I-based Mutation Detection with the EnGen® Mutation Detection Kit (NEB #E3321) to detect how efficiently a given sgRNA produces indels at targeted genomic DNA locus. T7 Endonuclease I recognizes and cleaves imperfectly matched DNA. In the first step, PCR products were obtained from the genomic DNA of target cells spanning the Cas9-sgRNA targeted region. In the second step, the PCR products were annealed and digested with T7 Endonuclease I. Fragments were analyzed by agarose gel to determine the efficiency of genome editing. Calculate the efficiency of estimated genome editing using the following formula: % indel = $100 \times (1 - \sqrt{1 - \text{fraction of cleaved bands}})$. For target gene with antibodies available, immunoblot analysis was performed to examine the level of protein reduction as a measurement of knockout efficiency.

## Establishment of chemoresistant cell lines

For chemoresistant cells derived from specific gene knockout, two independent sgRNAs (numbered as -1 and -2) targeting corresponding gene were designed and cloned into lentiCRISPRv2-blast vector (with blasticidin resistance cassette) or into lentiCRISPRv2-puro vector (with puromycin resistance cassette). Empty vector expressing null sgRNA (Vector or sgCtrl) or construct expressing sgRNA targeting *AAVS1* locus (sgAAVS1) served as control for gene-specific knockout. Lentiviruses were produced in HEK293T cells and used to infect HCT116, DLD1, MCF7 or NCI-H1568 cells. After 48 h, cells were selected with blasticidin for 7 days (for HCT116 and DLD1 cells) or with puromycin for 3 days (for MCF7 and NCI-H1568 cells) followed by recovery for additional two days. Successfully infected cells were further selected with oxaliplatin for 6 days ($\Delta SLC43A2^{OR}$-1, $\Delta SLC43A2^{OR}$-2, $\Delta TP53^{OR}$-1, $\Delta TP53^{OR}$-2, $\Delta CDKN1A^{OR}$-1 and $\Delta CDKN1A^{OR}$-2), irinotecan for 11 days ($\Delta BEND3^{IR}$-1, $\Delta BEND3^{IR}$-2, $\Delta KEAP1^{IR}$-1, $\Delta KEAP1^{IR}$-2, $\Delta SGF29^{IR}$-1 and $\Delta SGF29^{IR}$-2), 5-Fluorouracil for 15 days ($\Delta MED19^{5\text{-}FUR}$-1 and $\Delta MED19^{5\text{-}FUR}$-2) or 5-Fluorouracil for 19 days ($\Delta MMACHC^{5\text{-}FUR}$-1 and $\Delta MMACHC^{5\text{-}FUR}$-2) to establish corresponding chemoresistant cell lines. The additional drug adaptation after single-gene KO during establishment of chemoresistant cells helped to get rid of unedited cells and to make sure the resistance phenotype. For acquired resistance by "randomly" clonal selection with gradual drug challenge such as HCT116$^{OR}$ and DLD1$^{IR}$, wild type HCT116, DLD1, HCT8 and HT29 cells were treated with a stepwise increase of oxaliplatin or irinotecan for about 2-3 months to continually keep the survived cells until significant resistance appeared. Untreated wild type cells (Mock) served as a control in experiments for these "randomly" selected resistant cells. The drug response of control sensitive cells or resistant cells was determined by cell viability assay to confirm the resistance phenotype.

## Cell viability assay

Cell viability assays for IC$_{50}$ determination were based on MTT tetrazolium salt colorimetry. Succinate dehydrogenase in the mitochondria of living cells can reduce exogenous MTT to water-insoluble blue-purple crystalline formazan and deposit in cells, while dead cells have no such function. The amount of MTT crystals formed is proportional to the number of cells. Viable cell number was then measured according to the absorbance value. Cells (4 or $5 \times 10^3$ per well) were plated in 96-well plates and treated with chemotherapy drugs for 3 or 4 days followed by MTT addition. Cell viability was determined by the absorbance value at 490 nm using a BioTek Microplate Spectrophotometer (Gene Company Limited). For other experiments to evaluate cell growth, cell number was directly counted using a hemocytometer. Cell survival status was also visualized by crystal violet staining. Tested cells were washed twice with cold PBS and fixed with 100% methanol for 15 minutes at −20 °C. Subsequently, cells were stained with crystal violet and incubated for 15 minutes at room temperature. Wash away crystal violet with tap water and use a camera to acquire images of cell staining.

## RNA-seq sample preparation

At least $10^7$ cells were harvested for each RNA-seq sample. RNA was extracted by TRIzol reagent. Strand-specific libraries were prepared by Novogene and sequenced on Illumina PE150 platform. For RNA-seq to profile transcriptome in oxaliplatin- or irinotecan-resistant HCT116 or DLD1 cells, each condition was performed in duplicate. For all the other RNA-seq, three biological replicates were performed for each condition. Data analysis details are elaborated in the following analysis section.

## Immunoblot assay

Cells were lysed in RIPA buffer (Beyotime) containing phosphatase inhibitors (Meilunbio # MB12707) and protease inhibitors (Meilunbio #

MB26780) for 15 minutes at 4 °C. Centrifuge the lysis at $18472 \times g$ for 10 minutes at 4 °C and take the supernatant. Mix the protein supernatant with sample loading buffer and then separate proteins by running on a 10% bis-tris polyacrylamide gel. Proteins in gel were transferred to nitrocellulose (NC) membrane (Pall Corporation #27574625) and blocked with 5% skimmed milk (Difco™ Skim Milk #4296916) in TBST buffer. After sequential incubation with primary and secondary antibody, the signal of target protein was detected using a chemiluminescent imaging system (Tanon-5200). The following antibodies were used: p53 (1:1000, Santa Cruz Biotechnology, Cat# sc-126), GAPDH (1:3000, Santa Cruz Biotechnology, Cat# sc-25778), PLK4 (1:1000, Cell Signaling Technology, Cat# 71033), PLK1 (1:1000, Proteintech, Cat# 10305-1-AP), p-AKT (Thr308) (1:1000, Cell Signaling Technology, Cat# C31E5E), p-AKT (Ser473) (1:1000, Cell Signaling Technology, Cat# D9E), AKT (1:1000, Proteintech, Cat# 10176-2-AP), p-GSK3β (Ser 9) (1:1000, Beyotime, Cat# AF1531), GSK3β (1:1000, Proteintech, Cat# 22104-1-AP), p-ERK1/2 (1:1000, Cell Signaling Technology, Cat# 4370 S), ERK1/2 (1:1000, Cell Signaling Technology, Cat# 137F5), p21 (1:1000, Proteintech, Cat# 10355-1-AP), Cyclin D1 (1:1000, Santa Cruz Biotechnology, Cat# sc-718), Cyclin E (1:1000, Santa Cruz Biotechnology, Cat# sc-481), p-Rb (1:1000, Cell Signaling Technology, Cat# 9308), Rb (1:2000, Cell Signaling Technology, Cat# 9309), Cyclin B1 (1:1000, Proteintech, Cat# 55004-1-AP), CDK1 (1:2000, Proteintech, Cat# 19532-1-AP), Goat anti-Rabbit IgG (1:5000, Thermo Fisher Scientific, Cat# 31460) and Rabbit anti-Mouse IgG (1:5000, Thermo Fisher Scientific, Cat# 31450).

## Immunofluorescence

Cells were seeded in 24-well plates with small round glass slides. To visualize the process of spindle assembly, cells were synchronized after adherence by 2 mM thymidine for 16 h followed by recovery in normal media for 8 h. Repeat this treatment and recovery cycle again and treat cells with 100 ng/mL nocodazole for 5 h. After that, cells were fixed at five time points: 0 min, 30 min, 45 min, 60 min and 90 min. Before fixation, wash the cells with PBS for 3 times with 3 min each time. The slides then were fixed with 3.7% paraformaldehyde or methanol for 10 min followed by PBS washing for 3 times with 3 min each time. Use 0.1% Triton X-100 (prepared with PBS) to permeabilize the cell membrane at room temperature for 2 minutes followed by 3 times of PBS washing cycle. Use bovine serum albumin (BSA) to block the glass slides at room temperature for 30 min. Add diluted primary antibody (α-tubulin 1:100, Proteintech # 66031-1-Ig or γ-tubulin 1:200, Proteintech # 15176-1-AP) to each slide, put it in a humid chamber and incubate at 4 °C overnight. Immerse slides 3 times in PBS for 3 min each, add fluorescent secondary antibody (Goat anti-Mouse IgG, 1:500, Thermo Fisher Scientific # A-11001 or Goat anti-Rabbit IgG, 1:500, Thermo Fisher Scientific # A-11008) and incubate for 1 h at room temperature in a wet box. Immerse slides in PBS for 3 times with 3 min each time. Add DAPI for 5 min followed by PBS washing for 3 times with 5 min each time. Dry the liquid on the slide with absorbent paper, and cover the slide with anti-fluorescence quencher liquid seal. Images were collected under a confocal microscope (ZEISS LSM 900).

## Cell cycle analysis

For cell cycle assays, two staining methods (BrdU + PI or PI only) were used. For BrdU plus propidium iodide (PI) staining, 75 μM BrdU was added to the medium one hour before harvesting the treated cells. Cells were fully digested and washed with PBS. Add 90% ethanol dropwise while vortexing to fix the cells. Protect from light at 4 °C overnight. Resuspend and wash samples with PBS. Add 0.5 mL of 2 M HCl (0.5% Triton X-100) dropwise while vortexing and incubate for 30 min at room temperature followed by washing with PBS. Add 1 mL of 100 mM sodium borate solution (pH = 8.5), resuspend the cells, and centrifuge at $700 \times g$ for 5 min. Discard the supernatant, add 1 mL of 3%

BSA blocking solution prepared in PBST, and incubate at room temperature for 30 min followed by centrifugation to remove the supernatant. Add 100 μL BrdU primary antibody (Cell Signaling Technology # 5292; 1:200 dilution in PBST with 1% BSA) and incubate for one hour at room temperature. After washing with PBST, add 100 μL mouse fluorescence secondary antibody (Thermo Fisher Scientific # A-11001; 1:500 dilution in PBST with 1% BSA). Protect from light and incubate at room temperature for 30 min followed by washing with PBST. Add 350 μL PI/RNase Staining Buffer (BD # 550825) and incubate at room temperature for 30 min while protecting from light. Samples were then filtered through 200-mesh nylon membrane and cell cycle status was analyzed by running on a flow cytometer (BD LSRFortessa). For PI staining, cells were collected and washed once with PBS. Fix cells with 70% pre-cooled ethanol at -20 °C overnight. After washing with PBS, add 500 μL PI/RNase Staining Buffer and incubate at room temperature for 30 min in the dark. Use 200-mesh nylon membrane to filter the samples and proceed to flow cytometry for cell cycle analysis. The gating strategy for flow cytometry analysis was shown in Supplementary Fig. 19.

## Apoptosis assay

For apoptosis assays, cells were stained by two methods (PI + Annexin V or PI + Hoechst) and analyzed by flow cytometry. For PI plus Annexin V (FITC) staining, cells were collected and washed once with PBS. Take $1 \times 10^5$ resuspended cells, add 195 μL Annexin V-FITC binding solution, 5 μL Annexin V-FITC (Beyotime # C1062) and 10 μL of PI staining solution (Beyotime # C1062). Mix gently and incubate in the dark for 10-20 min at room temperature. Cells were passed through 200-mesh nylon membrane and analyzed on a flow cytometer (BD LSRFortessa). For PI plus Hoechst staining, about 1 million cells were collected for each cell sample in a 1.5 mL centrifuge tube. The cell pellet was resuspended in 0.8 mL of cell staining buffer. Add 5 μL Hoechst staining solution (Beyotime # C1056) and 5 μL PI staining solution. Mix well and incubate on ice or 4 °C for 20-30 min. Pass cells through 200-mesh nylon membrane and proceed to flow cytometry analysis. The gating strategy for flow cytometry analysis was shown in Supplementary Fig. 19.

## Xenograft assay

For testing CFI-400945 efficacy, two lines of HCT116 cells (sensitive control: Vector; resistant line: HCT116[OR]) were employed for xenograft assay. The mice were randomly allocated to each experimental group. A total of $3 \times 10^6$ cells were engrafted into both flanks by bilateral subcutaneous injection. When the tumor volume reached 100 mm$^3$ (6 days after cell inoculation), the mice were given intragastric administration of vehicle control (DMSO) or small molecule CFI-400945 at 10 mg/kg/day, 5 days a week. At 21 days post inoculation, the tumor was excised, measured and weighed. For validating oxaliplatin resistance, the mice were given intraperitoneal injection of vehicle control (H$_2$O) or oxaliplatin at 7 mg/kg/day, 2 days a week. At 20 days post inoculation, the tumor was excised, measured and weighed. Tumor volume was measured with vernier calipers and calculated using the following formula: volume = (length*width*width)/2. All the tumors were within the maximal size (< 1.5 cm) and volume (< 2000 mm$^3$) limit allowed by the related guidelines of the Biological and Medical Ethics Committee of Northeastern University.

## Software and algorithms

The software and algorithms used in this study are as follows: Base 4.0.4, Circlize 0.4.13, ComplexHeatmap 2.7.11, clusterProfiler 3.18.1, dbplyr 2.1.1, dplyr 1.0.8, DESeq2 1.30.1, ggplot2 3.3.5, ggtext 0.1.1, ggvenn 0.1.9, GSVA 1.38.2, MAGeCK 0.5.9.5, MAGeCKFlute 1.10.0, UpSetR 1.4.0, tidyverse 1.3.1, enrichplot 1.10.2, and Cytoscape 3.9.1.

## CRISPR screen data analysis

CRISPR screen data were analyzed by MAGeCK, MAGeCK-VISPR and MAGeCKFlute essentially as described[23,24,80]. Briefly, raw sequencing data were pre-processed by using MAGeCK to obtain the read counts for each sgRNA with default parameters. Quality control measurements by MAGeCK-VISPR showed good performance of the screening samples at the sequencing level, read count level, sample level and gene level. Control sgRNAs were used to normalize the data. The MAGeCK TEST algorithm was used to compare treatment with control samples to obtain the significantly enriched and depleted sgRNAs and genes. MAGeCK RRA uses the Robust Rank Aggregation (RRA) score to indicate gene performance during the screening by comparing sgRNA count of screen end point versus that of Day 0 (start of screen). RRA score was returned using ReadRRA function using MAGeCKFlute pipeline. For positive selection, a positive RRA score was assigned for each gene and higher RRA score indicates stronger positive selection for the corresponding gene. For negative selection, a negative value of RRA score was given for each gene and lower RRA score indicates stronger negative selection for the indicated gene. During first-round genome-scale chemogenomic screens, chemoresistance genes were defined as those whose loss-of-function confers resistance to corresponding drugs on screened cells and can be pulled out from positively selected genes with the cutoff as follows: score$^{drug}$ - score$^{DMSO}$ > 3 & score$^{drug}$ > 3. For chemosensitizer genes whose knockout accelerates cytotoxic cell death for given chemotherapy drugs, only samples with modest selection (Gini Index ≤ 0.3) without too much loss-of sgRNA or count distortion were analyzed as chemosensitizer genes should be pulled out from negatively selected hits. The cutoff for chemosensitizer genes was as follows: score$^{drug}$ - score$^{DMSO}$ < -3 & score$^{drug}$ < -3. For second-round druggable library CRISPR knockout screens, we preferentially focused on those druggable genes that were more negatively selected in resistant cells but less negatively or neutrally selected in normal sensitive cells. The cutoff for preferably druggable gene hits was as follows: score$^{resistant}$ – score$^{sensitive}$ < -1 & score$^{resistant}$ < -2 & score$^{sensitive}$ < 1.

## RNA-seq analysis

Raw reads were aligned to the hg38 human reference genome with the UCSC known gene transcript annotation using HISAT2 with the default parameters. Gene counts were quantified by HTSeq. Differentially expressed genes were identified by DESeq2 with the cutoff of |log2(fold change)| > 1 & $p$.adjust < 0.05.

## Functional enrichment analysis

The Gene Ontology (GO) and Kyoto Encyclopedia of Genes and Genomes (KEGG) functional enrichment analysis were performed by using the functions of "enricher" and "GSEA" in clusterProfiler R package. For "enricher" analysis, the input is gene sets, and the libraries are composed by KEGG and GOBP. For "GSEA" analysis, the input is the gene expression matrix, and the libraries are the same as used in "enricher". Enriched terms with false discovery rate (FDR) < 0.05 are regarded as significant.

## Network analysis

Chemoresistance genes for oxaliplatin or irinotecan were submitted to the STRING online database (Version 11.5, https://cn.string-db.org). We kept the links with interaction score > 0.4 and removed the disconnected nodes in the network. The interactions were imported into Cytoscape software (Version 3.9.1, https://cytoscape.org) to compose the series of proteins and their connecting lines as the network. To evaluate and visualize the biological networks, we used the CytoNCA plug-in betweenness (BC) as centrality analysis, which provides multiple centrality calculations for both weighted and unweighted networks. The proteins enriched in multiple functions and pathways were labeled and highlighted in the network.

## Data interrogation from public sources

**Gene catalogs and Mutational profiles.** Tumor suppressor genes (TSGs) and oncogenes were downloaded from the COSMIC Cancer Gene Census[31] (https://cancer.sanger.ac.uk/census/). Essential genes were obtained from previous study[80]. The mutational profiles of the cell lines used in this study were extracted from the Cancer Cell Line Encyclopedia (CCLE) (https://sites.broadinstitute.org/ccle/).

**TCGA significantly mutated genes.** The MutSig v2.0 analysis results were obtained from GDAC data portal (https://gdac.broadinstitute.org/), and genes with *FDR* < 0.01 were identified as significantly mutated genes. The mutational profiles of overlapping genes between these significantly mutated genes and our chemoresistance genes identified from screens were visualized by OncoPrint in cBioPortal (https://www.cbioportal.org/). The mutational profile of *PLK4* was also obtained from cBioPortal.

**Public colorectal cancer RNA-seq data.** The RNA-seq data of 54 samples (normal colon, primary CRC, and liver metastasis) from 18 CRC patients were downloaded from NCBI Gene Expression Omnibus (GEO) database under accession number GSE50760. The RNA-seq data of 60 samples (30 normal colon, 30 colon cancer) were obtained from NCBI GEO database under accession number GSE74602.

## Survival analysis

Clinical data of TCGA were obtained from GDAC data portal. Patients were divided into two groups based on either mutation status or expression levels for the tested gene or gene groups. Kaplan–Meier survival analysis and the Cox proportional hazard (log-rank) test were used to compare the overall or disease-free survival time between the two groups. The GEPIA (Gene Expression Profiling Interactive Analysis) (http://gepia.cancer-pku.cn/) web server was also employed to perform survival analysis for individual genes according to their expression levels.

## Statistics and reproducibility

No statistical methods were used to predetermine the sample size. To proceed with statistical analysis, more than three samples or repeats were performed. No data were excluded from the analyses. At least two or three independent replicates were performed in general to ensure reproducibility, and all attempts at reproducibility were successful. The number of replicates for corresponding experiments was also indicated in Methods and figure legends. Biological replicates here all refer to biologically distinct sources, i.e., independently cultured, treated or processed samples. The experiments were not randomized. The Investigators were not blinded to allocation during experiments and outcome assessment. GraphPad Prism 9 software and the R package were employed for statistical analysis. The replicate data were shown as Mean ± SD or Mean ± SEM. The unpaired two-sided *t* test was used to compare between two groups. Two-way ANOVA was used to estimate the difference for more than two groups. The Wilcoxon rank sum test was used when the data are not normally distributed. Asterisks indicate $*p < 0.05$, $**p < 0.01$, and $***p < 0.001$.

## Reporting summary

Further information on research design is available in the Nature Portfolio Reporting Summary linked to this article.

# Data availability

The raw sequencing data generated in this study have been deposited and are publicly available in the Genome Sequence Archive in the National Genomics Data Center, China National Center for Bioinformation/Beijing Institute of Genomics, Chinese Academy of Sciences, under accession code (GSA-Human: HRA002646) at https://ngdc.cncb.ac.cn/gsa-human. Tumor suppressor genes (TSGs)

and oncogenes were sourced from the COSMIC Cancer Gene Census (https://cancer.sanger.ac.uk/census/), while essential genes were derived from a previous study, and mutational profiles of the cell lines were obtained from the CCLE (https://sites.broadinstitute.org/ccle/). Significantly mutated genes from TCGA were identified using MutSig v2.0 analysis from the GDAC data portal, focusing on genes with FDR < 0.01. The overlapping genes between these significantly mutated genes and chemoresistance genes were visualized using OncoPrint in cBioPortal (https://www.cbioportal.org/), which also provided the mutational profile of PLK4. Additionally, RNA-seq data for colorectal cancer were collected from the NCBI GEO database, including 54 samples from 18 CRC patients (normal colon, primary CRC, liver metastasis) under accession number GSE50760, and 60 samples (30 normal colon, 30 colon cancer) under accession number GSE74602. Source data are provided in this paper.

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

## Acknowledgements

This work was supported by the National Natural Science Foundation of China (31871344; 32071441), Guangdong Basic and Applied Basic Research Foundation (2023A1515140084), the Fundamental Research Funds for the Central Universities (N182005005; N2020001), the 111 Project (B16009), the Construction Project of Liaoning Provincial Key Laboratory, China (2022JH13/10200026), and LiaoNing Revitalization Talents Program (XLYC1807212) to T.F.; the National Key Clinical Discipline, the National Natural Science Foundation of China (81972885) and the 1010 project of the Sixth Affiliated Hospital of Sun Yat-sen University (1010CG (2020)-20) to J.H.; the National Natural Science Foundation of China (32270683), the National Key R&D Program of China (2021YFC1712805), the Fundamental Research Funds for the Central Universities (BMU2021YJ064), and the High-performance Computing Platform of Peking University to H.-J.W.

## Author contributions

T.F. conceived the study. T.F., J.H., and H.-J.W. designed the research. C.Z. performed most of the experiments. W.-J.J. and Z.L. conducted the bioinformatics analysis. All the authors analyzed the data. T.F. wrote the manuscript with the input from C.Z., W.-J.J., H.-J.W., and J.H. as well as the help from all the other authors. T.F., J.H., and H.-J.W. supervised the study.

## Competing interests

A patent application has been filed through Northeastern University with T.F., C.Z., and Z.L. listed as co-inventors regarding targeting PLK4 in oxaliplatin-resistant cancers. Y.L. is an employee of BeiGene. All other authors declare no competing interests.

## Additional information

¹Key Laboratory of Bioresource Research and Development of Liaoning Province, College of Life and Health Sciences, Northeastern University, Shenyang 110819, China. ²National Frontiers Science Center for Industrial Intelligence and Systems Optimization, Northeastern University, Shenyang 110819, China. ³Key Laboratory of Data Analytics and Optimization for Smart Industry (Northeastern University), Ministry of Education, Shenyang 110819, China. ⁴Foshan Graduate School of Innovation, Northeastern University, Foshan 528311, China. ⁵Peking University Third Hospital, Beijing 100191, China. ⁶Department of Colorectal Surgery, the Sixth Affiliated Hospital, Sun Yat-sen University, Guangzhou, China. ⁷Clinical Research Center, the Sixth Affiliated Hospital, Sun Yat-sen

University, Guangzhou, China. [8]Guangdong Provincial Key Laboratory of Colorectal and Pelvic Floor Diseases, the Sixth Affiliated Hospital, Sun Yat-sen University, Guangzhou, China. [9]Guangdong Institute of Gastroenterology, Guangzhou, China. [10]Department of Cardiothoracic Surgery, Jinqiu Hospital of Liaoning Province, Shenyang, China. [11]Department of Thoracic Surgery, The Fourth Affiliated Hospital of China Medical University, Shenyang, China. [12]Key Laboratory for Anisotropy and Texture of Materials (Ministry of Education), School of Materials Science and Engineering, Northeastern University, Shenyang, China. [13]BeiGene Institute, BeiGene (Shanghai) Research & Development Co., Ltd, 200131 Shanghai, China. [14]Key laboratory of Carcinogenesis and Translational Research (Ministry of Education/Beijing), Peking University Cancer Hospital & Institute, Beijing 100142, China. [15]Department of Biomedical Informatics, School of Basic Medical Sciences, Peking University Health Science Center, Beijing 100191, China. [16]Center for Precision Medicine Multi-Omics Research, Institute of Advanced Clinical Medicine, Peking University, Beijing 100191, China. [17]These authors contributed equally: Chunge Zhong, Wen-Jie Jiang. ✉e-mail: hjwu@pku.edu.cn; huangj97@mail.sysu.edu.cn; feiteng@mail.neu.edu.cn

