## [Peer Review File · Nature Communications]

Reviewers' Comments:

Reviewer #1:

Remarks to the Author:

Drug resistance remains the major impediment to effective treatment of cancer and a search for mechanisms of resistance that are actionable holds the promise of improving the efficacy of chemotherapy. In the manuscript submitted by Zhong et al., the authors sought to identify genes associated with chemoresistance in cancer using genome-wide CRISPR-KO screens. The authors observed that the gene perturbations associated with drug resistance were largely cell-line dependent. However, there were some functional overlaps across the screens, such as genes involved in the cell cycle and tumor suppressor genes. The authors check some of their findings by generating single-gene KO lines and drug-resistant models. However, the utility of the findings is limited. The rationale underlying some of the experimental approaches needs to be improved, and the interpretation of some results needs to be clarified or contradicts some of the data. In addition, the study lacks mechanistic exploration, which is a failed opportunity to utilize the CRISPR datasets effectively.

Major Criticisms and Questions

1. Limiting this extensive survey to knocking out genes biases the results enormously, since many clinically significant resistance or survival mechanisms may require activation or mutation of genes. The unfortunate reality is that, without clinical correlation, these results might reflect both artificial selection conditions (cultured cells) and a very limited view of the set of genes that can contribute to resistance/survival of cancer cells in tissue culture.

2. The method section states that screens were carried out for 9-21 days; however, the authors should be explicit in stating the duration of the screen for each of the drugs and model cell lines. The methods section and descriptions in the results need to be revised and clarified with more detail throughout (discussed further below). The duration of drug exposure would be expected to make a big difference in the sort of genes associated with chemoresistance: short exposure would favor genes whose loss helps cancer cells survive chemotherapy without necessarily making them resistant, whereas longer exposure might favor genes whose knockout actually renders the cancer cells resistant to the agent in question.

3. Throughout the manuscript, the authors refer to the hits that were positively selected as "chemoresistance genes." This is conceptually confusing as the gene KO was associated with increased survival/resistance. It would make more sense to consider these genes as potential "resistance suppressors," analogous to tumor suppressor genes, as many of the hits were, in fact, tumor suppressors. In addition, since many of the hits were in genes that propel cell division but might result in defective mitoses, this reflects more of a transient survival mechanism and might not reflect long term resistance which requires that the cells be able to form colonies.

4. To validate the findings of the CRISPR screen, single-gene KO lines were generated. The authors state that the single KOs developed resistance to oxaliplatin or irinotecan within one-two weeks. The methodology here needs to be clarified. How did the single-gene KO alone affect drug response? Did the single-gene KOs initially show changes in drug sensitivity? How much did the drug response change once the cells were subject to a 1-2 week adaptation? At what dose was this selection carried out, and how did it compare to the dosing scheme for the WT cell models? What were the criteria for selecting these time points? Were the matched vector controls also exposed to the drug for this time? Did the adaptation time allow the activation or inhibition of other genes essential for the cells to survive the selection? The authors state, "The quick establishment of chemoresistant cells via specific gene knockout suggests a sufficient role but not merely a promotive function for the individual gene loss in driving resistance." (Lines 277- 279). However, if the single gene KO did not significantly alter the drug response without additional adaptation, then the gene KO is insufficient to drive resistance, and additional changes are required for the resistance phenotype.

5. The rationale for gene expression profiling is unclear, and the methodology is confusing. In lines 317-321, the researchers describe the differentially expressed genes derived by comparing the different oxaliplatin-resistant cells to control sensitive cells. Does this mean that all the drug-

adapted resistant lines (WT and single-gene KO) were compared to the parental line? Or was the analysis a pairwise comparison between drug-adapted single-gene KO to the matched drug-naïve single-gene KO? The latter would suggest that the authors sought to identify the additional alterations required for a fully resistant phenotype and support that these single gene alterations were insufficient for drug resistance, as stated above. If the drug-adapted KO cells are compared to the drug naïve KO cells, then the effects of the gene KO would not be detected. Also, one would not expect the KO of single genes from different functional pathways to identify many common changes in gene expression. The authors should more carefully examine this data set with a mechanistic rationale.

6. How did the drug-naïve single gene KOs affect the gene expression profiles compared to the WT-resistant cells? It is unclear why some single-gene targets were selected for expression profiling. The epigenetic regulators could have been of particular interest and may demonstrate the ability to regulate a set of genes differentially expressed in cells that acquire resistance natively. However, it is hard to know whether these results would be meaningful without knowing the phenotype of the drug naïve KO cells.

7. How did the WT-resistant expression profile compare to the results of the CRISPR screen? Were positively selected KO hits also significantly downregulated in the cells that naturally acquired drug resistance? Are the top upregulated genes in the resistant model less likely to survive when knocked out?

8. Expression profiles from the single KO models bias the functional overlap between the various models. What were the top functions that were dysregulated in the WT-resistant models? How do these functions relate to the genes identified in the CRISPR screen? How does the gene expression program of the resistant cell line compare to what is observed in patients/associated with survival?

9. The authors identified that KEAP1, BEND3, and CCDC KO cells have overlap in expression profiles and functional alterations. It may be worthwhile to examine this gene set more closely. How does this gene set compare to the results from the CRISPR screen? Is there any association between these genes with clinical outcomes?

10. TP53, CDKN1A, MED19, MDM2 and KEAP1 are essential genes for human cells. It is expected that complete knockout of these genes might affect cell growth, morphology and genomic stability. Studies show that TP53 deletion affects DNA structure and nuclear architecture which causes chromosomal instability (Rangel-Pozzo et al, J Clin Med. 2020). Such phenomena are not mentioned in the manuscript.

11. TP53 is the most well-studied tumour suppressor gene. It is well-established that deletion of TP53 and its downstream-regulated gene (P21) alters chemotherapeutic response. p53 transcriptionally controls Slc43a2 (Panatta et al; Cell Reports 2022). p53 cooperatively regulates PLK4 activity and centrosome integrity under stress (Nakamura et al; Nat Commun 2013). PLK4 overexpression accelerates tumorigenesis in the p53-deficient epidermis (Serçin et al; Nat Cell Biol 2016) and inhibition of PLK4 kinase activity generates polyploidy (Rajendran et al; J Physiol 2020). All this pre-reported information explains the possible mechanisms of PLK4 sgRNA or CFI-400945 mediated HCT116 (OR) cell growth inhibition (Fig 5). Therefore, the manuscript doesn't add new information to the literature.

12. The rationale for the second round screen needs to be described in a more compelling manner. The utilization of both the resistant and sensitive lines is intriguing, as it could identify synthetic vulnerabilities that arise as a consequence of drug resistance. However, there is no clear explanation or investigation into why the resistant cells, or cells with a particular perturbation, are more vulnerable to particular hits in the screen. It appears the screen predominantly yielded common essential genes that are not specific. A subset of genes were negatively selected in the resistant cells but positively selected in the sensitive cells. What were the functions of this gene set? What about genes that were negatively selected in the resistant cells but were not significant in the sensitive cells? It is unclear what was learned about the various paths to drug resistance from performing this screen and no mechanistic insights were provided.

Figure specific comments

- Paclitaxel group clusters together rather than with the matched cell line samples Figure 1C. The authors state that this indicates that the cells share a conserved resistance mechanism line (142-146). However, this could also be due to a suboptimal drug dose that did not exert effective selective pressure for the duration of the screen. The heat map also indicates that the Paclitaxel-treated samples are not highly correlated or have a slight negative correlation, and therefore, the data do not support the authors' conclusions.
- Figure 2 findings are expected due to the screening approach and the identification of tumor suppressor genes. Perhaps select panels in Figures 1 and 2 could be combined into one figure summarizing the approach of the screen. The clinical data is not needed, as they describe well-known tumor suppressor genes.
- In Figure 4C, an accumulation of cells in G2/M arrest and fewer cells in G1 in the control groups (Mock, sgAAVS1) would be expected with 5 μ M oxaliplatin. This assay should be replicated across a range of doses. How do the proliferation rates of the KO models compare to controls?
- In Figure 4E, why is p-ERK diminished in the oxaliplatin-resistant models? Would an increase in ERK-AKT signaling in the resistant cells be expected? The authors should clarify these findings and further explain their relevance.
- Figure 5F, sgCtrl alone significantly impacted cell proliferation in the HCT116 resistant model.
- Figures 5 and 6 have multiple control groups, and it is unclear what statistical comparisons are being reported. For example, vs. HCT116-OR-PLK4 KO vs. HCT116-Mock-PLK4 KO or HCT116-OR-PLK4 KO vs. HCT116-OR-sgCtrl. Perhaps it is appropriate to conduct all comparisons with Bonferroni correction, as they provide different information.

Reviewer #2:

Remarks to the Author:

Key results of this study are that via CRISPR whole genome screens the author reveal a potential role of polo-like kinase 4 (PLK4) in oxaliplatin resistance in cancer (particularly colorectal cancer).

The authors are to be commended on the significant amount of work performed here, and in general, the screening data coming out of this study could serve as an additional resource in parallel with the large-scale analyses available on <https://depmap.org/portal>, and build on findings from projects like DRIVE, Achilles, AVANA and SCORE.

The key findings, which centre around the role of PLK4 in driving oxaliplatin resistance, are less clear. The models used, termed rapidly derived models of resistance (following sgRNA targeting), require more characterisation to show that the mechanisms of resistance are indeed stable (and robust), as compared to their longer-term clonal selection models (which the authors ran in parallel for the control arm of the study). The resistance itself generated in many of these models, is not striking (e.g. Fig. 3b,d,f,g - this is upto 2-fold increase in sensitivity maximum, but for most drugs and models is less), which is likely to have a direct effect on the target identification downstream.

Ultimately, this is reflected in the key findings where a PLK4 inhibitor is used to reverse resistance in e.g. HCT-116 in vivo models, and we can only see a slow-down of tumour growth (tumours are just growing through at a slower pace; Fig.7a; Extended Fig. 7a). If this is a bona fide mechanism of resistance, tumour growth over such a short period of time (30 day study) should be blocked effectively, with subsequent further acquired resistance developing over a much longer period of time. In vivo data on oxaliplatin response in this model is also lacking (as a relevant control; these are meant to be oxaliplatin resistant tumours). Detailed mechanistic understanding of the PLK4 blockade in this context of resistance reversal is also lacking and would increase the fundamental significance of the study.

The patient-derived organoids, both normal and tumour, require associated basic molecular characterisation (to confirm their status) as they're used significantly for drug response experiments.

Clinical relevance of PLK4, as noted on page 12 is unclear. PLK4 is stated as frequently mutated, but this is maximally just under 8% (graph in extended fig. b may be misleading as the scale is only goes to 8%). Moreover, the mutations and other aberrations such as deep deletions have usually a different biological role (usually loss-of-function) compared with increased mRNA expression, so the 3rd paragraph on page 12 requires restructuring.

Reviewer #3:

Remarks to the Author:

In this manuscript the authors describe two sets of CRISPR knock-out screens performed in HCT116, DLD1, T47D, MCF7, A549 and NCI-H-1568 cell lines looking for genes whose loss influences the response to oxaliplatin, irinotecan, 5-FU, doxorubicin, cisplatin, docetaxel, and paclitaxel, or genes that arise as secondary sensitizers in cells that have developed resistance. The authors generate a large amount of data that will be generally useful to the broader scientific community. Some of the results are likely limited by the number of replicate experiments, but nonetheless are broadly useful. Their identification of Plk4 as a secondary target whose importance increases after the development of oxaliplatin resistance in colon cancer cell lines is a potentially clinically important finding.

Major comments:

1- The empiric dose-response curves from which the IC50s were determined for each cell line and drug should be shown as Supplemental Information.

2- Details of the methods need to be made clearer. The IC50 values must reflect some choice about how many days the cells were treated for - what were these for each drug. Of the 30 screens that were performed, how many were straight replicas? That is how many times were the same cell type treated with the same set of lentiviral constructs cells and then treated with a particular drug? It is hard to know how to evaluate the hits from the screens, if each screen was, in essence, performed only once. In addition, lack of replica data makes claims about the lack of commonality of resistance genes between the different cell types somewhat limited, because we do not know how reproducible a particular hit from the screen was in replicas within the same cell type. I suspect this may be, in part, why only very robust genes, like p53 and KEAP1 emerged. Nonetheless, the data are extremely valuable, but likely need to be interpreted in the light of limited experimental replication.

3- Lines 141-146 "Interestingly, only for paclitaxel resistance, the driver genes clustered together across different cell lines (Fig. 1c). These results indicated that a conserved genetic mechanism exist regardless of cellular background underlying paclitaxel resistance, while other tested chemotherapy drugs may develop cell-autonomous resistance involving a cell type-specific multidrug resistance mechanism."

I believe this is a total mis-representation and mis-understanding of the data. The set of genes conferring paclitaxel resistance here shows positive correlation among all the cells treated with paclitaxel, but differs from all the cell types treated with the other agents because paclitaxel is the only drug that targets microtubules. The other drugs damage DNA. The claims in the manuscript text above are incorrect.

4- Why is there no DLD1-Paclitaxel data in Figure 1C?

5- Lines 205-209 - "For some of these genes, either mutation or low expression might correlate with poor survival in certain cancer types (Fig. 2h-k), further supporting the clinical relevance of our findings. The mutational profile and/or expression status of chemoresistance genes may serve as potential biomarkers to predict the response to chemotherapy " How many of the resistance genes that they identified, for which patient data is available, correlated with patient survival, compared to a similar set of genes chosen at random? I am not sure what the authors are claiming here is even reasonable, since we do not know any of the treatment histories of the patients that

are shown. It might be more useful to select a patient cohort who were treated with a particular drug for a particular tumor type, and ask how many of the resistance genes the authors identified correlated with progression-free or absolute survival. This would be best done using patients with a tumor type that is represented in the cell lines that were screened.

6- Figure 4A - the cells are very difficult to see. Please use better images.

7- The same question about replication of the screen applied to the second round CRISPR screen. Please clarify if each cell lines was only screened once? How many of the 'hits' from this screen overlapped with the same gene hit in the original screen, if we only look at the sensitive 'control' cells? That is, were any of these hits from the druggable genome CRISPR screen came up as 'hits' in the control vehicle-treated cells from the original screen?

8- Figure 6A and b - the increased G2/M population following PLK4 knock-down or inhibition with CFI-400945 seems very modest at best, and in panel b, there is a similar change in G2/M among the resistant cells following DMSO treatment, that only fails to reach statistical significance because of a single data point in the DMSO-vector sample.

9- The data about spindle collapse in the Plk4-inhibited resistant cells needs to be quantified. Similarly, the number of centrosomes in the sensitive and resistant cells needs to be quantified before and after treatment.

10- Figure 7 a and b - it looks like there is a small difference even in the oxaliplatin-responsive tumors. The improved response of the HCT116OR tumors is not that much better than the oxaliplatin-responsive controls.

11- The Discussion should clarify a bit more what the authors are thinking in terms of Plk4 function. What the data say, to my understanding, is that a subset of cell lines that develop resistance to oxaliplatin, now have become sensitive, somehow, the Plk4 inhibition. The authors should speculate a little at least, on why this might be.

Minor comments:

12- Line 65 - I do not consider cisplatin or oxaliplatin to be alkylating agents, like the classic molecules such as chlorambucil or iphosphamide, since the platinum drugs do not attach an alkyl group to DNA. These agents are better described as DNA cross-linkers (or perhaps "alkylating-like").

Point-by-Point Responses to Reviewers

We would like to thank all the reviewers for their insightful and constructive comments. We have now performed a series of new experiments and analyses to address the concerns of the reviewers and carefully revised the manuscript accordingly. The changes are marked red in the manuscript, and point-by-point responses to the reviewers' comments are provided below.

Summary of Revision:

- Perform deeper analysis to support clinical relevance of screening hits.
- Provide more mechanistic insights on chemoresistance and PLK4 vulnerability using new experiments and analysis.
- Repeat and perform new *in vivo* xenograft experiments using optimized drug doses and additional treatment groups.
- Repeat and perform new CRISPR screening experiments with biological replicates for several representative samples.
- Other minor experiments, analyses and text modulation suggested by the reviewers.
- Newly added figure panels include: **Figs. S2j, 4d, S6a-b, S8f-g, S9a-f, 5b-f, S11a-j, 6d-i, S14b-e, 7a-c, S16a-e, S17b-c, and S18a.**
- Revised figure panels include: **Figs. 4c, 5a-d, 5f, S12b-e, 6a-c, and S14a.**

Reviewers' comments:

Reviewer #1 - Chemoresistance, CRISPR screens, genomics (Remarks to the Author):

Drug resistance remains the major impediment to effective treatment of cancer and a search for mechanisms of resistance that are actionable holds the promise of improving the efficacy of chemotherapy. In the manuscript submitted by Zhong et al., the authors sought to identify genes associated with chemoresistance in cancer using genome-wide CRISPR-KO screens. The authors observed that the gene perturbations associated with drug resistance were largely cell-line dependent. However, there were some functional overlaps across the screens, such as genes involved in the cell cycle and tumor suppressor genes. The authors check some of their findings by generating single-gene KO lines and drug-resistant models. However, the utility of the findings is limited. The rationale underlying some of the experimental approaches needs to be improved, and the interpretation of some results needs to be clarified or contradicts some of the data. In addition, the study lacks mechanistic exploration, which is a failed opportunity to utilize the CRISPR datasets effectively.

Response: We thank the reviewer for appreciating the importance of our work and meanwhile making several constructive comments to improve the manuscript.

Major Criticisms and Questions

1. Limiting this extensive survey to knocking out genes biases the results enormously, since many clinically significant resistance or survival mechanisms may require activation or mutation of genes. The unfortunate reality is that, without clinical correlation, these results might reflect both artificial selection conditions (cultured cells) and a very limited view of the set of genes that can contribute to resistance/survival of cancer cells in tissue culture.

Response: Gene knockout is a classic loss-of-function (LOF) approach to study gene function. According to the data from PCAWG (The Pan-Cancer Analysis of Whole Genomes) study ^{1,2}, for annotated cancer driver genes, the number of LOF genes (282) is slightly higher than activation genes (256) (https://dcc.icgc.org/releases/PCAWG/driver_mutations; the statistical data are extracted from *TableS1_compendium_mutational_drivers.xlsx*). Currently, several major large-scale consortiums or projects such as DepMap, DRIVE and Achilles are built on loss-of-function data as a starting point to decode cancer. Therefore, as an initial effort, we believe that CRISPR-mediated gene knockout still represents a necessary approach in our current study to identify functional genes during chemoresistance. For mechanisms requiring gene activation or specific mutations, CRISPR-mediated gene activation screens or base editing screens are actually being considered in our next phase studies. We now discussed this point in the revised manuscript (the last part of **Discussion** about “Limitations of this study”). Moreover, in the revised manuscript, we provided more clinical correlation (**Figs. 2c-f, S2, S8g, S9e,f, 7d-g, S17, and S18**) to support the clinical significance of our findings (newly added figures for this point are shown below).

New Supplementary Fig. 2j

j, Expression levels of *KEAP1*, *TP53*, and *SLC43A2* in recurrent or non-recurrent tumors from RNA-seq data of a colorectal cancer patient cohort treated with oxaliplatin. 92 out of 203 patients in this cohort received oxaliplatin treatment and 20 patients showed recurrence after the treatment.

New Supplementary Fig. 8g

g, Survival analysis of commonly up-regulated genes in $\Delta BEND3^{IR}$, $\Delta KEAP1^{IR}$ and $\Delta CCDC101^{IR}$ resistance models in a colorectal cancer patient cohort (GSE17538).

New Supplementary Fig. 9e,f

e-f, Survival analysis of top 100 up-regulated genes in HCT116^{OR} (**e**) or DLD1^{IR} (**f**) resistance models in a colorectal cancer patient cohort (GSE17538).

New Supplementary Fig. 17b,c

b, Colorectal cancers with mutated *PLK4* tend to have significantly higher tumor mutational burden according to TCGA data from cBioPortal. **c**, *PLK4* mutation is positively associated with advanced progression of colorectal cancers.

a

Organoid number	MLH1	MSH2	MSH6	PMS2	Ki-67	CDX2	HER2	CK7	CK20	BRAF
C-1	+	+	+	+	unknown	unknown	unknown	unknown	unknown	unknown
C-2	+	+	+	+	80%+	+	-	-	+	-
C-3	+	+	+	+	70%+	+	-	-	+	-
C-4	+	+	+	+	40%+	+	-	-	+	-
C-5	+	+	+	+	80%+	+	-	-	+	-
C-6	+	+	+	+	60%+	-	-	-	+	+
C-7	+	+	+	+	80%+	+	-	-	+	-

New Supplementary Fig. 18a

a, Molecular characterization for several clinical biomarkers in tumors for deriving organoids.

2.The method section states that screens were carried out for 9-21 days; however, the authors should be explicit in stating the duration of the screen for each of the drugs and

model cell lines. The methods section and descriptions in the results need to be revised and clarified with more detail throughout (discussed further below). The duration of drug exposure would be expected to make a big difference in the sort of genes associated with chemoresistance: short exposure would favor genes whose loss helps cancer cells survive chemotherapy without necessarily making them resistant, whereas longer exposure might favor genes whose knockout actually renders the cancer cells resistant to the agent in question.

Response: We have provided more details of methods in the revision to clarify this point (see **Methods** section: "First-round genome-wide CRISPR knockout screen"). During CRISPR screening, different CRISPR-edited cell pools exhibited differential but significantly accelerated pace to develop drug resistance. We usually stopped the screening until the edited resistant cell pools can steadily propagate for several passages in the presence of drug exposure and meanwhile the unedited control cell pools were all dead or clearly ceased to grow using the same drug treatment regimen.

3. Throughout the manuscript, the authors refer to the hits that were positively selected as "chemoresistance genes." This is conceptually confusing as the gene KO was associated with increased survival/resistance. It would make more sense to consider these genes as potential "resistance suppressors," analogous to tumor suppressor genes, as many of the hits were, in fact, tumor suppressors. In addition, since many of the hits were in genes that propel cell division but might result in defective mitoses, this reflects more of a transient survival mechanism and might not reflect long term resistance which requires that the cells be able to form colonies.

Response: We thank the reviewer for this insightful comment. We named "chemoresistance genes" to just emphasize their phenotypic relationship to chemoresistance. As these hits were identified via a loss-of-function manner and we felt it may be still premature to claim that these genes on the other side SUPPRESS chemoresistance without clear gain-of-function evidence. Similar logic applies to "chemosensitizer genes". To avoid conceptual confusion, we clearly defined "chemoresistance genes" in the main text: "*Chemoresistance genes were defined as those whose loss-of-function confer resistance to chemotherapy drugs on screened cells*" and further added a sentence for clarification: "*Notably, this definition here is only for better understanding and mentioning the resistance phenotype due to functional perturbation of corresponding gene. The gene's function per se is likely to suppress or restrain chemoresistance.*"

We agree that many hits were selected via a cell cycle and/or survival mechanism, but these gene effects were only evident in drug treatment conditions rather than normal growth condition, suggesting a drug-responsive but not a general growth-promoting effect. These findings were quite consistent with the cytotoxic mechanisms of chemotherapy drugs. Moreover, for the top hits with very strong selection effect, we have showed that knockout of the single gene can directly confer long-term resistance by various assays including single colony experiments (**Figs. 3, S4 and S5**).

4. To validate the findings of the CRISPR screen, single-gene KO lines were generated. The authors state that the single KOs developed resistance to oxaliplatin or irinotecan within one-two weeks. The methodology here needs to be clarified. How did the single-gene KO alone affect drug response? Did the single-gene KOs initially show changes in drug sensitivity? How much did the drug response change once the cells were subject to a 1-2 week adaptation? At what dose was this selection carried out, and how did it compare to the dosing scheme for the WT cell models? What were the criteria for selecting these time points? Were the matched vector controls also exposed to the drug for this time? Did the adaptation time allow the activation or inhibition of other genes essential for the cells to survive the selection? The authors state, “The quick establishment of chemoresistant cells via specific gene knockout suggests a sufficient role but not merely a promotive function for the individual gene loss in driving resistance.” (Lines 277- 279). However, if the single gene KO did not significantly alter the drug response without additional adaptation, then the gene KO is insufficient to drive resistance, and additional changes are required for the resistance phenotype.

Response: We apologize for not clearly describing the method details here and have provided this information in the **Methods** section of the revision. Actually, for the individual hits shown in **Fig. 3**, single-gene KO was sufficient to confer corresponding drug resistance. After CRISPR knockout, the majority of single-gene KO cell pools immediately resisted to corresponding drug treatment. The additional drug adaptation after single-gene KO during the establishment of chemoresistant cells was only to get rid of unedited cells as CRISPR knockout efficiency is not 100%, and to make sure the resistance phenotype. The short duration of drug adaption only killed a small percentage of single-gene KO cell pools which were likely those unsuccessful knockout and drug-sensitive cells. Furthermore, using *TP53* as an example, we have shown that complete knockout in multiple single colonies immediately causes resistance to oxaliplatin without any additional drug adaptation (**Fig. S4**). Compared to traditional drug pulsing and “randomly” clonal selection drug resistance models, these single-gene KO lines represent a different but more defined and quicker route to establish drug resistance. Using multiple and diverse resistance models are necessary to obtain more general and complete insights on chemoresistance. For better understanding and clarifications, we also discussed this point in the revised main text (in the second paragraph of the section: “*Rapid derivation of chemoresistant cells via evolutionarily distinct routes*”).

5. The rationale for gene expression profiling is unclear, and the methodology is confusing. In lines 317-321, the researchers describe the differentially expressed genes derived by comparing the different oxaliplatin-resistant cells to control sensitive cells. Does this mean that all the drug-adapted resistant lines (WT and single-gene KO) were compared to the parental line? Or was the analysis a pairwise comparison between drug-adapted single-gene KO to the matched drug-naïve single-gene KO? The latter would suggest that the authors sought to identify the additional alterations required for a fully resistant phenotype and support that these single gene alterations were insufficient for drug resistance, as stated above. If the drug-adapted KO cells are compared to the drug naïve KO cells, then the effects of the gene KO would not be detected. Also, one would not expect the KO of single genes from different functional pathways to identify many common changes in

gene expression. The authors should more carefully examine this data set with a mechanistic rationale.

Response: To understand the gene expression alternations underlying drug resistance, comparison was made firstly between each line of resistant cells and their parental control sensitive cells (WT resistance vs WT; single-gene KO resistance vs KO vector control) to retrieve differential gene signatures for each resistance model. After that, these chemoresistance-related gene signatures were compared across different models to see whether shared or model-specific gene expression alterations exist for chemoresistance. No matter the resistance models were established via “randomly” clonal selection under drug adaptation (so called here as “WT”) without knowing exact genetic/epigenetic alterations or defined single-gene KO-based routes (with or without drug adaptation), we regarded them as evolutionarily divergent resistance models for chemotherapy. In the revised main text, this rationale was now mentioned in the first paragraph of the section: “Gene expression signatures underlying oxaliplatin or irinotecan resistance”.

6. How did the drug-naïve single gene KOs affect the gene expression profiles compared to the WT-resistant cells? It is unclear why some single-gene targets were selected for expression profiling. The epigenetic regulators could have been of particular interest and may demonstrate the ability to regulate a set of genes differentially expressed in cells that acquire resistance natively. However, it is hard to know whether these results would be meaningful without knowing the phenotype of the drug naïve KO cells.

Response: We apologize for not clearly introducing the rationale and property of single-gene KO resistance models. As explained in above Point 4, drug adaption of single-gene KO cells is only a technical step to eradicate incomplete knockout cells. Complete single-gene KO is sufficient to confer corresponding drug resistance phenotype and this status is independent of drug adaption. Single-gene KO illustrates more diverse routes leading to chemoresistance. As different resistance models were established via divergent ways with differential genetic/epigenetic alterations or drug adaptation, it is hard to make direct comparison of gene expression profiles between different resistance cells. In contrast, we made comparison between each resistant line versus matched sensitive parental control cells to obtain model-specific resistance-associated gene expression signatures, and then compared these signatures between different single-gene KO and WT-resistant models to determine whether multiple models share consensus gene expression features underlying chemoresistance (**Figs. 4f-i and S8**).

7. How did the WT-resistant expression profile compare to the results of the CRISPR screen? Were positively selected KO hits also significantly downregulated in the cells that naturally acquired drug resistance? Are the top upregulated genes in the resistant model less likely to survive when knocked out?

Response: We performed this analysis as the reviewer suggested. Indeed, positively selected KO hits were significantly down-regulated in WT-resistant HCT116^{OR} or DLD1^{IR} models (**Fig. S9a**), supporting the loss-of-function effect of these genes in driving chemoresistance. Those up-regulated genes in WT-resistant models tended to be

negatively selected (less likely to survive) during chemoresistance gene knockout screens (**Fig. S9b**), suggesting that some of the up-regulated genes might functionally drive chemoresistance.

New Supplementary Fig. 9a,b

a, The expression status of positively selected gene hits during corresponding chemoresistance CRISPR screen in RNA-seq data of HCT116^{OR} and DLD1^{IR} resistant models. **b**, The RRA score of significantly up-regulated genes in HCT116^{OR} and DLD1^{IR} resistant models during HCT116-oxaliplatin or DLD1-irinotecan chemoresistant screen.

8.Expression profiles from the single KO models bias the functional overlap between the various models. What were the top functions that were dysregulated in the WT-resistant models? How do these functions relate to the genes identified in the CRISPR screen? How does the gene expression program of the resistant cell line compare to what is observed in patients/associated with survival?

Response: As the reviewer suggested, we specifically focused on the gene expression profiles in WT-resistant models. We displayed top enriched functional terms for up- or down-regulated genes in HCT116^{OR} or DLD1^{IR} WT-resistant models (**Fig. S9c,d**). Consistently, the genes comprising these enriched functional terms for up- or down-regulated genes in resistant cells tended to be either more negatively selected or positively selected during chemoresistance screen, respectively (**Fig. S9c,d**). In addition, high expression of the top up-regulated genes in WT-resistant models were significantly associated with bad survival in colorectal cancer patients (**Fig. S9e,f**).

New Supplementary Fig. 9c-f

c, The top enriched functional terms for up-regulated or down-regulated genes in HCT116^{OR} resistant model and their mean RRA scores for the genes in corresponding terms during HCT116-oxaliplatin chemoresistance screen. **d**, The top enriched functional terms for up-regulated or down-regulated genes in DLD1^{IR} resistant model and their mean RRA scores for the genes in corresponding terms during DLD1-irinotecan chemoresistance screen. **e-f**, Survival analysis of top 100 up-regulated genes in HCT116^{OR} (**e**) or DLD1^{IR} (**f**) resistant models in a colorectal cancer patient cohort (GSE17538).

9. The authors identified that KEAP1, BEND3, and CCDC KO cells have overlap in expression profiles and functional alterations. It may be worthwhile to examine this gene set more closely. How does this gene set compare to the results from the CRISPR screen? Is there any association between these genes with clinical outcomes?

Response: Following the reviewer's suggestion, we found that those overlapped up-regulated genes of the three single-gene KO resistant models ($\Delta BEND3^{IR}$, $\Delta KEAP1^{IR}$ and $\Delta CCDC101^{IR}$) exhibited more negative selection in chemoresistance CRISPR screen (Fig. S8f). These data suggest that targeting these up-regulated genes in resistant cells might sensitize cells to irinotecan treatment. Moreover, high expression of these commonly up-regulated genes across the three single-gene KO resistant models correlated to bad survival for colorectal cancer patients (Fig. S8g). These results not only link the gene expression profiles to chemoresistance function and clinical outcomes, but also corroborate the validity and usefulness of various single-gene KO resistant models.

New Supplementary Fig. 8f,g

f, RRA Score for commonly up- or down-regulated genes in $\Delta BEND3^{IR}$, $\Delta KEAP1^{IR}$ and $\Delta CCDC101^{IR}$ resistant models during DLD1-Irinotecan chemoresistance screen. **g**, Survival analysis of commonly up-regulated genes in $\Delta BEND3^{IR}$, $\Delta KEAP1^{IR}$ and $\Delta CCDC101^{IR}$ resistant models in a colorectal cancer patient cohort (GSE17538).

10. TP53, CDKN1A, MED19, MMACHC and KEAP1 are essential genes for human cells. It is expected that complete knockout of these genes might affect cell growth, morphology and genomic stability. Studies show that TP53 deletion affects DNA structure and nuclear architecture which causes chromosomal instability (Rangel-Pozzo et al, J Clin Med. 2020). Such phenomena are not mentioned in the manuscript.

Response: We agree that these genes play important and multi-faceted roles in cancer. Here we revealed novel functions of these genes in the context of chemoresistance on top of normal cell growth condition. We have now mentioned this point and cited the reference (ref:26) in the main text "As a well characterized multi-functional gene^{25,26}, p53 (encoded by TP53) play critical roles in DNA damage response, cell cycle regulation, chromosomal instability and so on. Here we found that TP53 knockout drove resistance to several chemotherapy drugs on top of its roles in normal cell growth (Fig. 1e)".

11. TP53 is the most well-studied tumour suppressor gene. It is well-established that deletion of TP53 and its downstream-regulated gene (P21) alters chemotherapeutic response. p53 transcriptionally controls Slc43a2 (Panatta et al; Cell Reports 2022). p53

cooperatively regulates PLK4 activity and centrosome integrity under stress (Nakamura et al; Nat Commun 2013). PLK4 overexpression accelerates tumorigenesis in the p53-deficient epidermis (Serçin et al; Nat Cell Biol 2016) and inhibition of PLK4 kinase activity generates polyploidy (Rajendran et al; J Physiol 2020). All this pre-reported information explains the possible mechanisms of PLK4 sgRNA or CFI-400945 mediated HCT116 (OR) cell growth inhibition (Fig 5). Therefore, the manuscript doesn't add new information to the literature.

Response: We respectfully disagree that our manuscript doesn't add new information to the literature. In addition to individually validated genes, our study provides a wealth of resource by comprehensively cataloging genes in relevance to chemotherapeutic response, as appreciated by Reviewer #2 and #3. Important cancer genes are usually studied from various angles in different contexts, however, their exact functions or working mechanisms in certain defined biological process may still vary and are not readily proved by just connecting scattered pieces of evidence in different contexts. Although the genes such as p53 and PLK4 have been deeply explored in cancer biology, our key findings that PLK4 can serve as an actionable therapeutic vulnerability for oxaliplatin-resistant colorectal cancer are not reported before.

12. The rationale for the second round screen needs to be described in a more compelling manner. The utilization of both the resistant and sensitive lines is intriguing, as it could identify synthetic vulnerabilities that arise as a consequence of drug resistance. However, there is no clear explanation or investigation into why the resistant cells, or cells with a particular perturbation, are more vulnerable to particular hits in the screen. It appears the screen predominantly yielded common essential genes that are not specific. A subset of genes were negatively selected in the resistant cells but positively selected in the sensitive cells. What were the functions of this gene set? What about genes that were negatively selected in the resistant cells but were not significant in the sensitive cells? It is unclear what was learned about the various paths to drug resistance from performing this screen and no mechanistic insights were provided.

Response: By comparing druggable vulnerabilities between resistant and sensitive lines, we preferred to focus on those targets that would be more effective in hard-to-kill chemoresistant cells. Such strategy helps to identify single-agent drug to use after the chemoresistance has been already established. Applying sequential drug treatment regimen to antagonize chemoresistance may alleviate the toxicity from multi-drug administration based on synthetic lethal strategy.

We updated the second-round CRISPR screening data by re-performing the screening for oxaliplatin-resistant models with two biological replicates and adding another two WT-resistant models (DLD1^{OR}, HCT8^{OR}) (Figs. 5a-d, S10 and S11). Also, we re-analyzed second-round CRISPR screen data for irinotecan resistance models and applied the same analytic parameters and cutoff for druggable gene hits to that in oxaliplatin-resistant models (Fig. S12). We specifically took a look at those subsets of genes as the reviewer suggested. Actually, there were few genes fallen in the category that were negatively selected in the resistant cells but positively selected in the sensitive cells. For those genes

in oxaliplatin-resistant models that were negatively selected in the resistant cells but were not significant in the sensitive cells, we found that the major functions enriched include cell cycle, proteolysis and immune responses (**Reviewer-only Fig. 1a,b**). For those genes fallen in that category in irinotecan-resistance models, similar functional terms were enriched as those in oxaliplatin group (**Reviewer-only Fig. 1c,d**).

As for the mechanistic insights, we dug into this question during the revision with a series of new experiments and new data (**Figs. 4d, S6b, 6a-i, and S14a-e**) on top of previous results (e.g., **Figs. 4, S6, and S15**). We summarized these findings into a working model in **Fig. 6i**. Meanwhile, we added a section entitled “*Mechanistic insights on oxaliplatin resistance and PLK4 dependency*” in **Results** part to explicitly discuss this point.

New Fig. 5b-d

b, Venn diagram of preferably druggable gene hits (ΔSLC43A2^{OR-1}, ΔTP53^{OR-2} or ΔCDKN1A^{OR-1} vs. Vector; HCT116^{OR} vs. untreated Mock; HCT8^{OR} vs. untreated HCT8 Mock; DLD1^{OR} vs. untreated DLD1 Mock) in six oxaliplatin-resistant colorectal cancer cell lines by second-round CRISPR screens.

c, The RRA scores of all the tested genes for oxaliplatin-sensitive control or -resistance cells in each comparison. The genes in red box are preferential targets against resistance. Several top genes are highlighted. **d**, The top enriched functional terms for druggable gene sets against oxaliplatin identified from second-round CRISPR screens in six resistant cell lines.

New Supplementary Fig. 11. Replicate correlation for indicated second-round CRISPR screen samples

a-j, Correlation of RRA scores between two biological replicates (R1 and R2) across all the indicated conditions during second-round druggable CRISPR screens against oxaliplatin resistance.

Reviewer-only Fig. 1. Gene number and enriched functional terms for special gene category a-b, Gene number overlap (a) and enriched terms among shared genes (b) for those negatively selected in the resistant cells but not significant in the sensitive cells across six oxaliplatin resistance models in 2nd round CRISPR screen. **c-d**, Gene number overlap (c) and enriched terms among shared genes (d) for those negatively selected in the resistant cells but not significant in the sensitive cells across four irinotecan resistance models in 2nd round CRISPR screen.

New Fig. 4d

d, Immunoblot analysis of indicated proteins for multiple oxaliplatin-sensitive or -resistant HCT116 cell lines in the absence or presence of 5 μ M oxaliplatin for 48 h.

New Supplementary Fig. S6b

b, Cell cycle analysis by propidium iodide (PI) staining of indicated cell lines in the absence or presence of 2.5 μ M oxaliplatin for 48 h. OXA, oxaliplatin. Mean \pm SD with $n = 3$. Unpaired t test, * $p < 0.05$ and *** $p < 0.001$.

New Fig. 6a-i

a, Cell cycle analysis upon *PLK4* knockout in oxaliplatin-sensitive and -resistant cells. Mean \pm SD with $n = 3$. Unpaired t test, $*p < 0.05$, $**p < 0.01$ and $***p < 0.001$. **b**, Cell cycle analysis by BrdU and PI staining upon CFI-400945 treatment in oxaliplatin-sensitive and -resistant cells. Mean \pm SD with $n = 3$. Unpaired t test, $**p < 0.01$ and $***p < 0.001$. **c**, *PLK4* knockout results in spindle collapse, folding and slippage in oxaliplatin-resistant cells. Scale bar, 5 μm . **d**, Quantification of spindle collapse for (c). Mean \pm SD with $n = 3$, 100-200 cells were analyzed per independent experiment. Unpaired t test, $*p < 0.05$ and $**p < 0.01$. **e-f**, Distribution of cell populations with indicated centrosome number according to γ -tubulin staining as shown in **Fig. S14c** before or after *PLK4* knockout (**e**) or CFI-400945 treatment (**f**) for oxaliplatin-sensitive and -resistant cells synchronized at G2/M phase. Mean \pm SD with $n = 3$, 100-200 cells were analyzed per independent experiment. Unpaired t test, $*p < 0.05$ and $**p < 0.01$. **g-h**, Quantification of cells with $>4N$ DNA content (polyploidy) upon *PLK4* knockout (**g**) or CFI-400945 treatment (**h**) in oxaliplatin-sensitive and -resistant cells. Mean \pm SD with $n = 3$. Unpaired t test, $*p < 0.05$ and $**p < 0.01$. **i**, Schematic model illustrating the molecular mechanisms about oxaliplatin resistance in colorectal cancer and the *PLK4* dependency of oxaliplatin-resistant cells.

New Supplementary Fig. 14a-e

a, Treatment with CFI-400945 results in spindle collapse, folding and slippage in oxaliplatin-resistant cells. Scale bar, 5 μ m. **b**, Quantification of spindle collapse for **(a)**. Mean \pm SD with n = 3, 100-200 cells were analyzed per independent experiment. Unpaired t test, * p < 0.05 and ** p < 0.01. **c**, Representative images for centrosome quantification by γ -tubulin staining. **d-e**, Distribution of cell populations with zero centrosome number before or after *PLK4* knockout **(d)** or CFI- 400945

treatment (e) for oxaliplatin-sensitive and -resistant cells. Mean \pm SD with n = 3, 100-200 cells were analyzed per independent experiment. Unpaired t test, * $p < 0.05$ and ** $p < 0.01$.

Figure specific comments

- Paclitaxel group clusters together rather than with the matched cell line samples Figure 1C. The authors state that this indicates that the cells share a conserved resistance mechanism line (142-146). However, this could also be due to a suboptimal drug dose that did not exert effective selective pressure for the duration of the screen. The heat map also indicates that the Paclitaxel-treated samples are not highly correlated or have a slight negative correlation, and therefore, the data do not support the authors' conclusions.

Response: We thank the reviewer for correcting this point and we have removed the statement in the revised manuscript.

- Figure 2 findings are expected due to the screening approach and the identification of tumor suppressor genes. Perhaps select panels in Figures 1 and 2 could be combined into one figure summarizing the approach of the screen. The clinical data is not needed, as they describe well-known tumor suppressor genes.

Response: Although there are scattered reports showing the implication of individual tumor suppressor genes (TSGs) in drug response, here we displayed a comprehensive view on TSGs for their functional relevance to specific chemo drug response, which to our opinion is still very important for understanding the molecular basis of chemoresistance. To highlight the main findings, we indeed followed the reviewer's suggestion to re-organize **Figs.1 and 2** by removing some clinical data into supplementary **Fig. S2**.

- In Figure 4C, an accumulation of cells in G2/M arrest and fewer cells in G1 in the control groups (Mock, sgAAVS1) would be expected with 5 μ M oxaliplatin. This assay should be replicated across a range of doses. How do the proliferation rates of the KO models compare to controls?

Response: We have now repeated the assay using different doses of oxaliplatin. Oxaliplatin significantly reduced cell fraction of S phase and arrested sensitive cells at G2/M phase at medium (5 μ M) (**Fig. 4c**) or high dose (10 μ M) (**Fig. S6a**), while at low dose of 2.5 μ M (**Fig. S6b**) cells were also arrested at G1 phase. In contrast, resistant cells established from different routes all successfully counteracted such effect (**Figs. 4c and S6a,b**).

The proliferation rates of these resistant lines (including single-gene KO models) were similar to their parental control cells under normal growth condition (**Reviewer-only Fig. 2a**). In accordance, the single-gene KO did not show strong cell fitness selection in vehicle condition but only positively selected upon oxaliplatin treatment during chemoresistance CRISPR screen (**Reviewer-only Fig. 2b**).

New Supplementary Fig. 6a,b

a-b, Cell cycle analysis by propidium iodide (PI) staining of indicated cell lines in the absence or presence of 10 μ M (**a**) or 2.5 μ M (**b**) oxaliplatin for 48 h. OXA, oxaliplatin. Mean \pm SD with $n = 3$. Unpaired t test, * $p < 0.05$ and *** $p < 0.001$.

Reviewer-only Fig. 2. Cell growth rate of oxaliplatin-resistant or -sensitive control cells (a, Cell number quantification of indicated oxaliplatin-sensitive and -resistant cell lines under normal cell culture condition. b, RRA scores for the indicated genes during chemoresistance screen.

•In Figure 4E, why is p-ERK diminished in the oxaliplatin-resistant models? Would an increase in ERK-AKT signaling in the resistant cells be expected? The authors should clarify these findings and further explain their relevance.

Response: It was reported that ERK activation mediates cell cycle arrest and apoptosis after DNA damage^{3, 4}. We surmise that diminished p-ERK in oxaliplatin-resistant models may help to antagonize cell cycle arrest and apoptosis elicited by oxaliplatin. We mentioned this point in the revised text “Furthermore, ERK activation is reported to mediate cell cycle arrest and apoptosis after DNA damage^{44, 45}. Indeed, we found that ERK signaling was repressed or less activated under oxaliplatin treatment across all the tested oxaliplatin-resistant cells (Fig. S6c).”.

•Figure 5F, sgCtrl alone significantly impacted cell proliferation in the HCT116 resistant model.

Response: The comparison was made between sgCtrl vs sgPLK4 in the same resistance model. In this assay, each resistance model was independently tested and the initial cell plating number for each resistant model has not been strictly controlled. We repeated this experiment by carefully controlling the initial cell plating number and new data led to the same conclusion as before (Fig. 5f).

New Fig. 5f

f, Cell growth effect of *PLK4* knockout by two independent sgRNAs in oxaliplatin-sensitive or -resistant cell lines. Mean \pm SD with $n = 3$. Unpaired t test (Δ *SLC43A2*^{OR-1}, Δ *TP53*^{OR-1} or Δ *CDKN1A*^{OR} vs. *sgAAVS1*; HCT116^{OR} vs. Mock), ** $p < 0.01$ and *** $p < 0.001$.

•Figures 5 and 6 have multiple control groups, and it is unclear what statistical comparisons are being reported. For example, vs. HCT116-OR-PLK4 KO vs. HCT116-Mock-PLK4 KO or HCT116-OR-PLK4 KO vs. HCT116-OR-sgCtrl. Perhaps it is appropriate to conduct all comparisons with Bonferroni correction, as they provide different information.

Response: Following the reviewer's suggestion, we conducted all comparisons with Bonferroni correction and indicated comparison group in figure panels or figure legends.

Reviewer #2 - Resistance, CRISPR screens, functional genomics - (Remarks to the Author):

Key results of this study are that via CRISPR whole genome screens the author reveal a potential role of polo-like kinase 4 (PLK4) in oxaliplatin resistance in cancer (particularly colorectal cancer).

The authors are to be commended on the significant amount of work performed here, and in general, the screening data coming out of this study could serve as an additional resource in parallel with the large-scale analyses available on <https://depmap.org/portal>, and build on findings from projects like DRIVE, Achilles, AVANA and SCORE.

Response: We thank the reviewer for appreciating the values and efforts of our work.

The key findings, which centre around the role of PLK4 in driving oxaliplatin resistance, are less clear. The models used, termed rapidly derived models of resistance (following sgRNA targeting), require more characterisation to show that the mechanisms of resistance are indeed stable (and robust), as compared to their longer-term clonal selection models (which the authors ran in parallel for the control arm of the study). The resistance itself generated in many of these models, is not striking (e.g. Fig. 3b,d,f,g - this is upto 2-fold increase in sensitivity maximum, but for most drugs and models is less), which is likely to have a direct effect on the target identification downstream.

Response: Using a wide range of drug doses, we have determined IC₅₀ values via cell viability assay as a standard measurement of drug responses for different resistance models (Fig. 3) which clearly indicate that both gene-knockout derived models and long-term clonal selection models have similar and strong resistance effects to corresponding drugs (usually tens of fold increase of drug sensitivity). The growth curve in original Fig. 3b,d,f,h (new Fig. S5a-d) were used to calculate IC₅₀ values and it was not precise to directly define sensitivity gap from single data points at short drug treatment duration. Moreover, the consistency of resistance phenotypes for these different models were also supported by many other assays such as cell growth, cell morphology and cell cycle analysis (e.g., Figs. 3 and 4a-c). Altogether, we believe that these evolutionarily distinct resistance models are valid and effective to identify consensus vulnerabilities for resistant cancers.

Ultimately, this is reflected in the key findings where a PLK4 inhibitor is used to reverse resistance in e.g. HCT-116 in vivo models, and we can only see a slow-down of tumour growth (tumours are just growing through at a slower pace; Fig.7a; Extended Fig. 7a). If this is a bona fide mechanism of resistance, tumour growth over such a short period of time (30 day study) should be blocked effectively, with subsequent further acquired resistance developing over a much longer period of time. In vivo data on oxaliplatin response in this model is also lacking (as a relevant control; these are meant to be oxaliplatin resistant tumours). Detailed mechanistic understanding of the PLK4 blockade

in this context of resistance reversal is also lacking and would increase the fundamental significance of the study.

Response: We have repeated these *in vivo* experiments using optimized drug dosing regimen (increasing CFI-400549 dose from 7.5 mg/kg/day to 10 mg/kg/day) and found that the growth of resistant HCT116^{OR} xenograft was significantly blocked by PLK4 inhibitor CFI-400945 than that of sensitive group (**Figs. 7a-c and S16e**). Moreover, we performed additional xenograft experiments and proved that HCT116^{OR} xenograft was indeed oxaliplatin-resistant while sensitive control cells were not (**Fig. S16a-d**).

As for the mechanistic insights, we dug into this question during the revision with a series of new experiments and new data (**Figs. 4d, S6b, 6a-i, and S14a-e**) on top of previous results (e.g., **Figs. 4, S6, and S15**). We summarized these findings into a working model in **Fig. 6i**. Meanwhile, we added a section entitled “Mechanistic insights on oxaliplatin resistance and PLK4 dependency” in **Results** part to explicitly discuss this point.

New Fig. 7a-c

a, Tumor volume measured at indicated time points after xenograft implantation for oxaliplatin-sensitive (Vector) or -resistant (HCT116^{OR}) xenograft treated with vehicle or CFI-400945. (n=12, number of tumor). Mean ± SEM. Two-way ANOVA, ****p* < 0.001. **b-c**, Relative tumor volume (compared to day 0) (**b**) or tumor weight (**c**) of mouse xenografts for indicated HCT116 cells measured at Day 21 post implantation. (n=12, number of tumor). Mean ± SEM. Unpaired t test, **p* < 0.05 and ****p* < 0.001.

New Supplementary Fig. 16e

e, Photos of tumors collected from oxaliplatin-sensitive or -resistance xenograft bearing mice with vehicle or CFI-400549 treatment at the experimental endpoint.

New Supplementary Fig. 16a-d

a, Tumor volume measured at indicated time points after xenograft implantation for oxaliplatin-sensitive (Vector) or -resistant (HCT116^{OR}) xenograft treated with vehicle or oxaliplatin. (n=12, number of tumor). Mean ± SEM. Two-way ANOVA, ***p < 0.001. **b**, Photos of tumors collected from oxaliplatin-sensitive or -resistance xenograft bearing mice with vehicle or oxaliplatin treatment at the experimental endpoint. **c-d**, Relative tumor volume (compared to day 0) (**c**) or tumor weight (**d**) of mouse xenografts for indicated HCT116 cells measured at Day 20 post implantation. (n=12, number of tumor). Mean ± SEM. Unpaired t test, *p < 0.05 and **p < 0.01. ns, non-significant.

New Fig. 4d

d, Immunoblot analysis of indicated proteins for multiple oxaliplatin-sensitive or -resistant HCT116 cell lines in the absence or presence of 5 μ M oxaliplatin for 48 h.

New Supplementary Fig. S6b

b, Cell cycle analysis by propidium iodide (PI) staining of indicated cell lines in the absence or presence of 2.5 μ M oxaliplatin for 48 h. OXA, oxaliplatin. Mean \pm SD with $n = 3$. Unpaired t test, * $p < 0.05$ and *** $p < 0.001$.

New Fig. 6a-i

a, Cell cycle analysis upon *PLK4* knockout in oxaliplatin-sensitive and -resistant cells. Mean \pm SD with $n = 3$. Unpaired t test, $*p < 0.05$, $**p < 0.01$ and $***p < 0.001$. **b**, Cell cycle analysis by BrdU and PI staining upon CFI-400945 treatment in oxaliplatin-sensitive and -resistant cells. Mean \pm SD with $n = 3$. Unpaired t test, $**p < 0.01$ and $***p < 0.001$. **c**, *PLK4* knockout results in spindle collapse, folding and slippage in oxaliplatin-resistant cells. Scale bar, 5 μm . **d**, Quantification of spindle collapse for (c). Mean \pm SD with $n = 3$, 100-200 cells were analyzed per independent experiment. Unpaired t test, $*p < 0.05$ and $**p < 0.01$. **e-f**, Distribution of cell populations with indicated centrosome number according to γ -tubulin staining as shown in **Fig. S14c** before or after *PLK4* knockout (**e**) or CFI-400945 treatment (**f**) for oxaliplatin-sensitive and -resistant cells synchronized at G2/M phase. Mean \pm SD with $n = 3$, 100-200 cells were analyzed per independent experiment. Unpaired t test, $*p < 0.05$ and $**p < 0.01$. **g-h**, Quantification of cells with $>4N$ DNA content (polyploidy) upon *PLK4* knockout (**g**) or CFI-400945 treatment (**h**) in oxaliplatin-sensitive and -resistant cells. Mean \pm SD with $n = 3$. Unpaired t test, $*p < 0.05$ and $**p < 0.01$. **i**, Schematic model illustrating the molecular mechanisms about oxaliplatin resistance in colorectal cancer and the *PLK4* dependency of oxaliplatin-resistant cells.

New Supplementary Fig. 14a-e

a, Treatment with CFI-400945 results in spindle collapse, folding and slippage in oxaliplatin-resistant cells. Scale bar, 5 μ m. **b**, Quantification of spindle collapse for (a). Mean \pm SD with $n = 3$, 100-200 cells were analyzed per independent experiment. Unpaired t test, $*p < 0.05$ and $**p < 0.01$. **c**, Representative images for centrosome quantification by γ -tubulin staining. **d-e**, Distribution of cell populations with zero centrosome number before or after *PLK4* knockout (d) or CFI-400945

treatment (e) for oxaliplatin-sensitive and -resistant cells. Mean \pm SD with n = 3, 100-200 cells were analyzed per independent experiment. Unpaired t test, * $p < 0.05$ and ** $p < 0.01$.

The patient-derived organoids, both normal and tumour, require associated basic molecular characterisation (to confirm their status) as they're used significantly for drug response experiments.

Response: We have provided more information on molecular characterization for these clinical samples in the revised manuscript (Fig. S18a).

a

Organoid number	MLH1	MSH2	MSH6	PMS2	Ki-67	CDX2	HER2	CK7	CK20	BRAF
C-1	+	+	+	+	unknown	unknown	unknown	unknown	unknown	unknown
C-2	+	+	+	+	80%+	+	-	-	+	-
C-3	+	+	+	+	70%+	+	-	-	+	-
C-4	+	+	+	+	40%+	+	-	-	+	-
C-5	+	+	+	+	80%+	+	-	-	+	-
C-6	+	+	+	+	60%+	-	-	-	+	+
C-7	+	+	+	+	80%+	+	-	-	+	-

New Supplementary Fig. 18a

a, Molecular characterization for several clinical biomarkers in tumors for deriving organoids.

Clinical relevance of PLK4, as noted on page 12 is unclear. PLK4 is stated as frequently mutated, but this is maximally just under 8% (graph in extended fig. b may be misleading as the scale is only goes to 8%). Moreover, the mutations and other aberrations such as deep deletions have usually a different biological role (usually loss-of-function) compared with increased mRNA expression, so the 3rd paragraph on page 12 requires restructuring.

Response: In the revised manuscript, we provided more clinical relevance of PLK4. We found that mutant PLK4 tumors tended to have higher tumor mutational burden and were usually associated with advanced stages of tumor progression in colorectal cancer (Fig. S17b,c), further suggesting an oncogenic role of altered PLK4 in cancers. Following the reviewer's suggestion, we restructured the statements regarding to this part in the text as follows: "In human tumors, PLK4 is frequently mutated in many types of cancer, including colorectal cancer (Fig. S17a). Tumors bearing mutant PLK4 tend to have higher tumor mutational burden and are usually associated with advanced stages of tumor progression in colorectal cancer (Fig. S17b,c). Increased RNA expression of PLK4 was found in colorectal tumor samples compared to normal tissues as evidenced in two independent clinical datasets (Fig. S17d,e).".

New Supplementary Fig. 17b,c

b, Colorectal cancers with mutated *PLK4* tend to have significantly higher tumor mutational burden according to TCGA data from cBioPortal. **c**, *PLK4* mutation is positively associated with advanced progression of colorectal cancers.

Reviewer #3 - Polo-like kinases, synergy (Remarks to the Author):

In this manuscript the authors describe two sets of CRISPR knock-out screens performed in HCT116, DLD1, T47D, MCF7, A549 and NCI-H-1568 cell lines looking for genes whose loss influences the response to oxaliplatin, irinotecan, 5-FU, doxorubicin, cisplatin, docetaxel, and paclitaxel, or genes that arise as secondary sensitizers in cells that have developed resistance. The authors generate a large amount of data that will be generally useful to the broader scientific community. Some of the results are likely limited by the number of replicate experiments, but nonetheless are broadly useful. Their identification of Plk4 as a secondary target whose importance increases after the development of oxaliplatin resistance in colon cancer cell lines is a potentially clinically important finding.

Response: We thank the reviewer for positively commenting on the values and significance of our work.

Major comments:

1- The empiric dose-response curves from which the IC₅₀s were determined for each cell line and drug should be shown as Supplemental Information.

Response: We have moved these data to Supplemental Information as the reviewer suggested (**Figs. S1b and S5**).

2- Details of the methods need to be made clearer. The IC₅₀ values must reflect some choice about how many days the cells were treated for - what were these for each drug. Of the 30 screens that were performed, how many were straight replicas? That is how many times were the same cell type treated with the same set of lentiviral constructs and then treated with a particular drug? It is hard to know how to evaluate the hits from the screens, if each screen was, in essence, performed only once. In addition, lack of replica data makes claims about the lack of commonality of resistance genes between the different cell types somewhat limited, because we do not know how reproducible a particular hit from the screen was in replicas within the same cell type. I suspect this may be, in part, why only very robust genes, like p53 and KEAP1 emerged. Nonetheless, the data are extremely valuable, but likely need to be interpreted in the light of limited experimental replication.

Response: We apologize for not clearly describing the mentioned method details. For IC₅₀ determination, the cells were treated by gradient doses of drugs for three or four days and cell viability was measured by MTT method (see **Methods** section "Cell viability assay").

As for the screening replication, we agree with the reviewer that inclusion of replica data will be more helpful to consolidate the conclusion. However, we did not perform screening replication in the original manuscript because that 1) a huge increase of labor and resources if having screening replicates due to the large sample size in this study; 2) we applied a series of quality control processes to justify the validity of our screening data;

and 3) our team has accumulated ample experiences in CRISPR screening to ensure the success of such experiments (e.g. *PNAS* 2017, *PNAS* 2019; *Mol Cell* 2020; *Nat Commun* 2023; *Nat Biomed Eng.* 2023) ⁵⁻⁹.

To address the reviewer’s concern here, during the revision, we re-performed a new batch of first-round CRISPR screening only for HCT116-oxaliplatin group of samples with two biological replicates (named R2 and R3; the original batch of sample named R1). We found a high concordance for the top hits across the three replicates despite different screening time and batches (**Reviewer-only Fig. 3a**). Notably, *TP53*, *CDKN1A* and *SLC43A2* remains among the very top hits across all the replicates (**Reviewer-only Fig. 3b**). These results support a high quality of the screening data.

Reviewer-only Fig. 3. Reproducibility of HCT116-oxaliplatin chemoresistance screen
(a, RRA scores for the top positively selected hits among the three replicates of screen. b, Top positively selected hits for each screen replicate were highlighted.

Due to a huge demand of efforts, we felt quite difficult to replicate all the screening samples in this study during the revision. However, to further address the replica concern, we did re-performed all the second-round CRISPR screening experiments with two biological replicates for the original HCT116-based four oxaliplatin-resistant models as well as corresponding sensitive controls. Also, we further included two “randomly” clonal selection resistance models (DLD1^{OR}, HCT8^{OR}) and their parental controls in the druggable CRISPR screening section. These new batches of second-round CRISPR screening experiments (20 sets of screen samples) consolidated key findings of previous screens (e.g., *PLK4*) with high concordance between replicates. The details were elaborated later in the “Response to the following Question #7”.

3- Lines 141-146 “Interestingly, only for paclitaxel resistance, the driver genes clustered together across different cell lines (Fig. 1c). These results indicated that a conserved genetic mechanism exist regardless of cellular background⁵ underlying paclitaxel resistance, while other tested chemotherapy drugs may develop cell-autonomous resistance involving a cell type-specific multidrug resistance mechanism.” I believe this is a total mis-representation and mis-understanding of the data. The set of genes conferring paclitaxel resistance here shows positive correlation among all the cells treated with paclitaxel, but differs from all the cell types treated with the other agents because

paclitaxel is the only drug that targets microtubules. The other drugs damage DNA. The claims in the manuscript text above are incorrect.

Response: We thank the reviewer for pointing out our mis-interpretation here and have removed the statement in the revised manuscript.

4- Why is there no DLD1-Paclitaxel data in Figure 1C?

Response: We did not apply Paclitaxel to DLD1 cells for screening experiments as this drug is not a widely used treatment option for colorectal cancer in clinics compared to oxaliplatin and irinotecan.

5- Lines 205-209 - “For some of these genes, either mutation or low expression might correlate with poor survival in certain cancer types (Fig. 2h-k), further supporting the clinical relevance of our findings. The mutational profile and/or expression status of chemoresistance genes may serve as potential biomarkers to predict the response to chemotherapy “ How many of the resistance genes that they identified, for which patient data is available, correlated with patient survival, compared to a similar set of genes chosen at random? I am not sure what the authors are claiming here is even reasonable, since we do not know any of the treatment histories of the patients that are shown. It might be more useful to select a patient cohort who were treated with a particular drug for a particular tumor type, and ask how many of the resistance genes the authors identified correlated with progression-free or absolute survival. This would be best done using patients with a tumor type that is represented in the cell lines that were screened.

Response: Following the reviewer’s suggestion, we tried to explore this question using a more specialized patient cohort. Nevertheless, it was quite difficult to identify such cohorts with survival data treated with oxaliplatin or irinotecan. Instead, we managed to find a colorectal cancer patient cohort with RNA-seq data (GSE87211). This dataset comprises a total of 203 patients with 92 receiving oxaliplatin treatment, and 20 of them experiencing recurrence (drug resistance) post treatment. We found that the expression of chemoresistance gene hits such as *TP53*, *SLC43A2* or *KEAP1* were significantly down-regulated in drug-resistant recurrence tumors (**Fig. S2j**), supporting the functional and clinical relevance of those screening hits.

New Supplementary Fig. 2j

j, Expression levels of *KEAP1*, *TP53*, and *SLC43A2* in recurrent or non-recurrent tumors from RNA-seq data of a colorectal cancer patient cohort treated with oxaliplatin. 92 out of 203 patients in this cohort received oxaliplatin treatment and 20 patients showed recurrence after the treatment.

6- Figure 4A - the cells are very difficult to see. Please use better images.

Response: We replaced the indicated images with new pictures of high resolution (**Fig. 4a**).

New Fig. 4a

a, Cell morphology of multiple oxaliplatin-sensitive (Mock, Vector and sgAAVS1 groups) and -resistant cell lines in the absence or presence of 5 μ M oxaliplatin for six days. Scale bar, 50 μ m.

7- The same question about replication of the screen applied to the second round CRISPR screen. Please clarify if each cell lines was only screened once? How many of the 'hits' from this screen overlapped with the same gene hit in the original screen, if we only look at the sensitive 'control' cells? That is, were any of these hits from the druggable genome CRISPR screen came up as 'hits' in the control vehicle-treated cells from the original screen?

Response: To better address the screening replication concern, we fully repeated the second-round CRISPR screening for all the oxaliplatin-resistant models in original manuscript. Furthermore, we also included two additional "randomly" clonal selection resistance models (DLD1^{OR}, HCT8^{OR}) and their parental controls for the druggable CRISPR screening. All these new CRISPR screening experiments were performed with two biological replicates and a total of 20 new CRISPR screen datasets were generated. As shown in **Fig. S11**, all the replicates correlated very well, suggesting a very high degree of reproducibility. Using similar analytic approach in original manuscript, we now updated druggable hits across different oxaliplatin resistance models. Notably, previous hits such as PLK4 were still recapitulated in new CRISPR screen data. Thus, we used these new data to replace previous ones in **Fig. 5a-d**.

As for the comparison between 2nd round Mock sample (druggable gene screen) and 1st round control DMSO sample (whole genome screen), we examined the performance of 1716 druggable genes covered by both samples and found a significant overlap of significantly negative selection hits (**Reviewer-only Fig. 4a**). In addition, despite different library design and sgRNA efficiency model for both rounds of screens, the RRA scores for all the shared genes in both samples still correlated quite well ($r=0.65$) (**Reviewer-only Fig. 4b**). These results further support a high quality for our CRISPR screen data.

New Supplementary Fig. 11. Replicate correlation for indicated second-round CRISPR screen samples

a-j, Correlation of RRA scores between two biological replicates (R1 and R2) across all the indicated conditions during second-round druggable CRISPR screens against oxaliplatin resistance.

New Fig. 5b-d

b, Venn diagram of preferably druggable gene hits (Δ SLC43A2^{OR-1}, Δ TP53^{OR-2} or Δ CDKN1A^{OR-1} vs. Vector; HCT116^{OR} vs. untreated Mock; HCT8^{OR} vs. untreated HCT8 Mock; DLD1^{OR} vs. untreated DLD1 Mock) in six oxaliplatin-resistant colorectal cancer cell lines by second-round CRISPR screens. **c**, The RRA scores of all the tested genes for oxaliplatin-sensitive control or -resistance cells in each comparison. The genes in red box are preferential targets against resistance. Several top genes are highlighted. **d**, The top enriched functional terms for druggable gene sets against oxaliplatin identified from second-round CRISPR screens in six resistant cell lines.

Reviewer-only Fig. 4. Comparison of 1st round and 2nd round CRISPR screen samples
a, Venn diagram showing the overlap of negatively selected genes (RRA score <-1) for the indicated samples. **b**, Correlation of RRA scores corresponding to commonly covered genes in both screens between the two indicated samples.

8- Figure 6A and b - the increased G2/M population following PLK4 knock-down or inhibition with CFI-400945 seems very modest at best, and in panel b, there is a similar change in G2/M among the resistant cells following DMSO treatment, that only fails to reach statistical significance because of a single data point in the DMSO-vector sample.

Response: We re-analyzed those FACS data by more precisely gating the cells and the results in new **Fig. 6a,b** showed clearer trends than before.

New Fig. 6a,b

a, Cell cycle analysis upon *PLK4* knockout in oxaliplatin-sensitive and -resistant cells. Mean \pm SD with $n = 3$. Unpaired t test, * $p < 0.05$, ** $p < 0.01$ and *** $p < 0.001$. **b**, Cell cycle analysis by BrdU and PI staining upon CFI-400945 treatment in oxaliplatin-sensitive and -resistant cells. Mean \pm SD with $n = 3$. Unpaired t test, ** $p < 0.01$ and *** $p < 0.001$.

9- The data about spindle collapse in the Plk4-inhibited resistant cells needs to be quantified. Similarly, the number of centrosomes in the sensitive and resistant cells needs to be quantified before and after treatment.

Response: Following the reviewer's suggestion, we provided quantified data on spindle collapse using new batches of immunofluorescence data (**Figs. 6c,d and S14a,b**), and got similar conclusions as before.

We also quantified the number of centrosomes in these cells. Centrosomes were significantly depleted in *PLK4* knockout cells for both sensitive and resistant cells (**Fig. S14c,d**). However, CFI-400945 treatment did not reduce centrosome number (**Fig. S14e**), which is consistent with previous reports that relatively low dose range of CFI-400945 can rather increase centriole duplication possibly due to an increase in protein levels of partially active PLK4¹⁰. These results suggest that centriole biogenesis may not be the key to explain the differential dependency on PLK4 between sensitive and resistant cells. PLK4 could also control spindle assembly in acentrosomal cells through coordination with pericentriolar material (PCM)^{11, 12}. Indeed, we observed that several PCM scaffolding genes (e.g., *TUBG1* and *CEP192*) were de-regulated in *PLK4* ablated oxaliplatin-resistant cells (**Fig. S15f**). More importantly, PLK4 was also reported to be essential for cytokinesis^{13, 14}. Failed spindle assembly and cytokinesis may cause enhanced polyploidy and stalled cell division. Consistent with this hypothesis, we observed increased multi-centrosomes (≥ 3) in resistant cells than sensitive cells at synchronized at G2/M phase upon *PLK4* knockout or CFI-400945 treatment (**Fig. 6e,f**), which is possibly due to polyploidy. Quantification of $>4N$ (DNA content) provided direct evidence that PLK4 loss-of-function led to enhanced polyploidy in resistance cells compared to sensitive control cells (**Fig. 6g,h**). These data indicate that, compared to control sensitive cells, PLK4-mediated spindle assembly and cytokinesis control is essentially required specifically for oxaliplatin-resistant cell growth.

New Fig. 6c,d

c, *PLK4* knockout results in spindle collapse, folding and slippage in oxaliplatin-resistant cells. Scale bar, 5 μ m. **d**, Quantification of spindle collapse for (c). Mean \pm SD with $n = 3$, 100-200 cells were analyzed per independent experiment. Unpaired t test, * $p < 0.05$ and ** $p < 0.01$.

New Supplementary Fig. 14a,b

a, Treatment with CFI-400945 results in spindle collapse, folding and slippage in oxaliplatin-resistant cells. Scale bar, 5 μ m. **b**, Quantification of spindle collapse for (a). Mean \pm SD with $n = 3$, 100-200 cells were analyzed per independent experiment. Unpaired t test, * $p < 0.05$ and ** $p < 0.01$

New Supplementary Fig. 14c-e

c, Representative images for centrosome quantification by γ -tubulin staining. **d-e**, Distribution of cell populations with zero centrosome number before or after *PLK4* knockout (d) or CFI- 400945

treatment (e) for oxaliplatin-sensitive and -resistant cells. Mean \pm SD with $n = 3$, 100-200 cells were analyzed per independent experiment. Unpaired t test, $*p < 0.05$ and $**p < 0.01$.

New Fig. 6e-h

e-f, Distribution of cell populations with indicated centrosome number according to γ -tubulin staining as shown in **Fig. S14c** before or after *PLK4* knockout (**e**) or CFI-400945 treatment (**f**) for oxaliplatin-sensitive and -resistant cells synchronized at G2/M phase. Mean \pm SD with $n = 3$, 100-200 cells were analyzed per independent experiment. Unpaired t test, $*p < 0.05$ and $**p < 0.01$. **g-h**, Quantification of cells with >4N DNA content (polyploidy) upon *PLK4* knockout (**g**) or CFI-400945 treatment (**h**) in oxaliplatin-sensitive and -resistant cells. Mean \pm SD with $n = 3$. Unpaired t test, $*p < 0.05$ and $**p < 0.01$.

10- Figure 7 a and b - it looks like there is a small difference even in the oxaliplatin-responsive tumors. The improved response of the HCT116^{OR} tumors is not that much better than the oxaliplatin-responsive controls.

Response: We have repeated these *in vivo* experiments using optimized drug dosing regimen (increasing CFI-400549 dose from 7.5 mg/kg/day to 10 mg/kg/day) and found that the growth of resistant HCT116^{OR} xenograft was significantly blocked by *PLK4* inhibitor CFI-400945 than that of sensitive group (**Figs. 7a-c and S16e**). Moreover, we performed additional xenograft experiments and proved that HCT116^{OR} xenograft was indeed oxaliplatin-resistant while sensitive control cells were not (**Fig. S16a-d**).

New Fig. 7a-c

a, Tumor volume measured at indicated time points after xenograft implantation for oxaliplatin-sensitive (Vector) or -resistant (HCT116^{OR}) xenograft treated with vehicle or CFI-400945. (n=12, number of tumor). Mean ± SEM. Two-way ANOVA, ****p* < 0.001. **b-c**, Relative tumor volume (compared to day 0) (**b**) or tumor weight (**c**) of mouse xenografts for indicated HCT116 cells measured at Day 21 post implantation. (n=12, number of tumor). Mean ± SEM. Unpaired t test, **p* < 0.05 and ****p* < 0.001.

New Supplementary Fig. 16e

e, Photos of tumors collected from oxaliplatin-sensitive or -resistance xenograft bearing mice with vehicle or CFI-400549 treatment at the experimental endpoint.

New Supplementary Fig. 16a-d

a, Tumor volume measured at indicated time points after xenograft implantation for oxaliplatin-sensitive (Vector) or -resistant (HCT116^{OR}) xenograft treated with vehicle or oxaliplatin. (n=12, number of tumor). Mean ± SEM. Two-way ANOVA, *** $p < 0.001$. **b**, Photos of tumors collected from oxaliplatin-sensitive or -resistance xenograft bearing mice with vehicle or oxaliplatin treatment at the experimental endpoint. **c-d**, Relative tumor volume (compared to day 0) (**c**) or tumor weight (**d**) of mouse xenografts for indicated HCT116 cells measured at Day 20 post implantation. (n=12, number of tumor). Mean ± SEM. Unpaired t test, * $p < 0.05$ and ** $p < 0.01$. ns, non-significant.

11- The Discussion should clarify a bit more what the authors are thinking in terms of Plk4 function. What the data say, to my understanding, is that a subset of cell lines that develop resistance to oxaliplatin, now have become sensitive, somehow, the Plk4 inhibition. The authors should speculate a little at least, on why this might be.

Response: We thank the reviewer for the suggestion. To strengthen this part, we dug into this question during the revision with a series of new experiments and new data (**Figs. 4d, S6b, 6a-i, and S14a-e**) on top of previous results (e.g., **Figs. 4, S6, and S15**). We summarized these findings into a working model in **Fig. 6i**. Meanwhile, we added a section entitled "*Mechanistic insights on oxaliplatin resistance and PLK4 dependency*" in **Results** part to explicitly discuss this point.

New Fig. 4d

d, Immunoblot analysis of indicated proteins for multiple oxaliplatin-sensitive or -resistant HCT116 cell lines in the absence or presence of 5 μ M oxaliplatin for 48 h.

New Supplementary Fig. S6b

b, Cell cycle analysis by propidium iodide (PI) staining of indicated cell lines in the absence or presence of 2.5 μ M oxaliplatin for 48 h. OXA, oxaliplatin. Mean \pm SD with $n = 3$. Unpaired t test, * $p < 0.05$ and *** $p < 0.001$.

New Fig. 6a-i

a, Cell cycle analysis upon *PLK4* knockout in oxaliplatin-sensitive and -resistant cells. Mean \pm SD with $n = 3$. Unpaired t test, $*p < 0.05$, $**p < 0.01$ and $***p < 0.001$. **b**, Cell cycle analysis by BrdU and PI staining upon CFI-400945 treatment in oxaliplatin-sensitive and -resistant cells. Mean \pm SD with $n = 3$. Unpaired t test, $**p < 0.01$ and $***p < 0.001$. **c**, *PLK4* knockout results in spindle collapse, folding and slippage in oxaliplatin-resistant cells. Scale bar, 5 μm . **d**, Quantification of spindle collapse for (c). Mean \pm SD with $n = 3$, 100-200 cells were analyzed per independent experiment. Unpaired t test, $*p < 0.05$ and $**p < 0.01$. **e-f**, Distribution of cell populations with indicated centrosome number according to γ -tubulin staining as shown in **Fig. S14c** before or after *PLK4* knockout (**e**) or CFI-400945 treatment (**f**) for oxaliplatin-sensitive and -resistant cells synchronized at G2/M phase. Mean \pm SD with $n = 3$, 100-200 cells were analyzed per independent experiment. Unpaired t test, $*p < 0.05$ and $**p < 0.01$. **g-h**, Quantification of cells with $>4N$ DNA content (polyploidy) upon *PLK4* knockout (**g**) or CFI-400945 treatment (**h**) in oxaliplatin-sensitive and -resistant cells. Mean \pm SD with $n = 3$. Unpaired t test, $*p < 0.05$ and $**p < 0.01$. **i**, Schematic model illustrating the molecular mechanisms about oxaliplatin resistance in colorectal cancer and the *PLK4* dependency of oxaliplatin-resistant cells.

New Supplementary Fig. 14a-e

a, Treatment with CFI-400945 results in spindle collapse, folding and slippage in oxaliplatin-resistant cells. Scale bar, 5 μ m. **b**, Quantification of spindle collapse for **(a)**. Mean \pm SD with $n = 3$, 100-200 cells were analyzed per independent experiment. Unpaired t test, $*p < 0.05$ and $**p < 0.01$. **c**, Representative images for centrosome quantification by γ -tubulin staining. **d-e**, Distribution of cell populations with zero centrosome number before or after *PLK4* knockout **(d)** or CFI- 400945 treatment **(e)** for oxaliplatin-sensitive and -resistant cells. Mean \pm SD with $n = 3$, 100-200 cells were analyzed per independent experiment. Unpaired t test, $*p < 0.05$ and $**p < 0.01$.

Minor comments:

12- Line 65 - I do not consider cisplatin or oxaliplatin to be alkylating agents, like the classic molecules such as chlorambucil or iphosphamide, since the platinum drugs do not attach an alkyl group to DNA. These agents are better described as DNA cross-linkers (or perhaps “alkylating-like”).

Response: We have changed the statement as the reviewer suggested. The corrected text was as follows: “chemotherapy drugs can be grouped into different categories including alkylating or alkylating-like agents (inducing DNA damage, e.g., cisplatin and oxaliplatin),...”.

References

1. Consortium, I.T.P.-C.A.o.W.G. Pan-cancer analysis of whole genomes. *Nature* **578**, 82-93 (2020).
2. Sabarinathan, R. *et al.* The whole-genome panorama of cancer drivers. *bioRxiv*, 190330 (2017).
3. Tang, D. *et al.* ERK activation mediates cell cycle arrest and apoptosis after DNA damage independently of p53. *J Biol Chem* **277**, 12710-12717 (2002).
4. Singh, S., Upadhyay, A.K., Ajay, A.K. & Bhat, M.K. p53 regulates ERK activation in carboplatin induced apoptosis in cervical carcinoma: a novel target of p53 in apoptosis. *FEBS Lett* **581**, 289-295 (2007).
5. Cheng, X. *et al.* Modeling CRISPR-Cas13d on-target and off-target effects using machine learning approaches. *Nat Commun* **14**, 752 (2023).
6. Fei, T. *et al.* Deciphering essential cistromes using genome-wide CRISPR screens. *Proc Natl Acad Sci U S A* **116**, 25186-25195 (2019).
7. Fei, T. *et al.* Genome-wide CRISPR screen identifies HNRNPL as a prostate cancer dependency regulating RNA splicing. *Proc Natl Acad Sci U S A* **114**, E5207-E5215 (2017).
8. Shu, S. *et al.* Synthetic Lethal and Resistance Interactions with BET Bromodomain Inhibitors in Triple-Negative Breast Cancer. *Mol Cell* **78**, 1096-1113 e1098 (2020).
9. Li, Z. *et al.* Intrinsic targeting of host RNA by Cas13 constrains its utility. *Nat Biomed Eng* (2023).
10. Mason, J.M. *et al.* Functional characterization of CFI-400945, a Polo-like kinase 4 inhibitor, as a potential anticancer agent. *Cancer Cell* **26**, 163-176 (2014).
11. Meitinger, F. *et al.* TRIM37 controls cancer-specific vulnerability to PLK4 inhibition. *Nature* **585**, 440-446 (2020).
12. Yeow, Z.Y. *et al.* Targeting TRIM37-driven centrosome dysfunction in 17q23-amplified breast cancer. *Nature* **585**, 447-452 (2020).
13. Press, M.F. *et al.* Role for polo-like kinase 4 in mediation of cytokinesis. *Proc Natl Acad Sci U S A* **116**, 11309-11318 (2019).
14. Rosario, C.O. *et al.* Plk4 is required for cytokinesis and maintenance of chromosomal stability. *Proc Natl Acad Sci U S A* **107**, 6888-6893 (2010).

Reviewers' Comments:

Reviewer #1:

Remarks to the Author:

In the revised manuscript submitted by Zhong et al., the authors have adequately addressed most of the concerns the peer reviewers expressed. The additional supplementary figures were clarifying or confirmatory in a way that supports the paper's major conclusions. The datasets generated while conducting this study should be a valuable resource. The manuscript's findings on the increased dependence of p53-mutant/null cells on PLK4 under stress conditions build upon the findings of Nakamura et al., *Nature Comms* 2013. The authors should cite and discuss this paper, as it offers potential explanations for their observations and would enhance understanding of the relationships outlined in panel Fig 6i.

Nakamura T, Saito H, Takekawa M. SAPK pathways and p53 cooperatively regulate PLK4 activity and centrosome integrity under stress. *Nat Commun.* 2013;4:1775. doi: 10.1038/ncomms2752. PMID: 23653187.

Reviewer #2:

Remarks to the Author:

I have assessed the revised manuscript and believe that the authors have appropriately addressed all of the comments I raised. The authors have provided significant amount of new data particularly with regard to the PLK4 mechanism, a number of additional relevant controls (as requested), and new data to strengthen and support their *in vitro* screens. More detailed methodology has also been provided. I have no additional comments/concerns and believe that the manuscript is of suitable quality for publication.

Reviewer #3:

Remarks to the Author:

Re-review of Zhong et al., "CRISPR screens reveal convergent targeting strategies against evolutionarily distinct chemoresistance in cancer"

In this revised manuscript, the authors have largely dealt with issues of reproducibility (thank you for Reviewer only Figure 3), and now have performed a single replicate experiment of their second round CRISPR screen with reasonably correlation coefficients for the MaGECK RRA coefficients. and transparency of the methods. They also did a limited amount of comparison of their 'hits' with patient data, though there are limited amounts of reliable well-documented patient datasets out there, and I am quite satisfied with what the authors have done. They also better addressed issues with PLK4 Knock-down effects on G2/M accumulation, but I think the comparisons they are showing in Figure 6a and b are incorrect (see below). This will not change the conclusions, but the correct comparisons are more appropriate for examining Plk4 effects. The data on the effects of PLK4 on reducing centrosome number in unsynchronized cells yet an increase in centrosome number in PLK4 knock-down cells arrested at G2/M fits with the authors' model of failed cytokinesis or issues related to the function of Plk4 on the PCM during G2.

There are a few minor remaining issues the authors should address.

1. Lines 322-328 – here the authors are describing loss of G1/S phase regulators, but the papers they cite describing ERK activity (Tang et al 2002) deals with G2/M progression, not G1/S progression. The correct reference showing that DNA damage regulates G1/S cell cycle arrest through ERK activation is Tentner et al, "Combined experimental and computational analysis of DNA damage signaling reveals context-dependent roles for Erk in apoptosis and G1/S arrest after genotoxic stress" *Molecular Systems Biology* 8:568 (2012), and this reference should be cited.
2. The correct statistical comparisons for G2/M arrest in Figure 6a are between the AAVS1 sgRNA and the PLK4 sgRNA for each of the respective cell lines, not between the different cell lines. That

is, they should compare sgAAVS1-Vector with sgPLK4-Vector, and compare sgAAVS1-(delta)SLC43A2OR-1 with sgPLK4-(delta)SLC43A2OR-1, etc. The same set of comparisons should be done with panel b for the CFI-400945 treatment. Notably, in the sensitive cells, there is very little increase in G2/M arrest upon PLK4 depletion or inhibition, but this is much more pronounced in the resistant cells.

3. The caption to Figure 7 (line 1560) – “d.e.” should read “e.f.” and the reference to panel ‘f’ (line 1562) should refer to panel g instead.

4. The paper would be improved if it was quickly edited by a native English speaker. There are some complex double-negative arguments related to depletion ‘hits’ in the screen and up- and down-regulated gene sets that are a bit challenging to understand, and several points where improved syntax would make the paper easier to read.

Point-by-Point Responses to Reviewers

We would like again to thank all the reviewers for their insightful and constructive comments. We have now addressed these concerns of the reviewers and carefully revised the manuscript accordingly. The changes are marked red in the manuscript, and point-by-point responses to the reviewers' comments are provided below.

Reviewers' comments:

Reviewer #1 (Remarks to the Author):

In the revised manuscript submitted by Zhong et al., the authors have adequately addressed most of the concerns the peer reviewers expressed. The additional supplementary figures were clarifying or confirmatory in a way that supports the paper's major conclusions. The datasets generated while conducting this study should be a valuable resource. The manuscript's findings on the increased dependence of p53-mutant/null cells on PLK4 under stress conditions build upon the findings of Nakamura et al., Nature Comms 2013. The authors should cite and discuss this paper, as it offers potential explanations for their observations and would enhance understanding of the relationships outlined in panel Fig 6i.

Nakamura T, Saito H, Takekawa M. SAPK pathways and p53 cooperatively regulate PLK4 activity and centrosome integrity under stress. Nat Commun. 2013;4:1775. doi: 10.1038/ncomms2752. PMID: 23653187.

Response: We thank the reviewer for these comments and have cited this paper in the revised manuscript (Ref. #64).

Reviewer #2 (Remarks to the Author):

I have assessed the revised manuscript and believe that the authors have appropriately addressed all of the comments I raised. The authors have provided significant amount of new data particularly with regard to the PLK4 mechanism, a number of additional relevant controls (as requested), and new data to strengthen and support their in vitro screens. More detailed methodology has also been provided. I have no additional comments/concerns and believe that the manuscript is of suitable quality for publication.

Response: We thank the reviewer for these comments.

Reviewer #3 - Polo-like kinases, synergy (Remarks to the Author):

Re-review of Zhong et al., "CRISPR screens reveal convergent targeting strategies against evolutionarily distinct chemoresistance in cancer"

In this revised manuscript, the authors have largely dealt with issues of reproducibility (thank you for Reviewer only Figure 3), and now have performed a single replicate experiment of their second round CRISPR screen with reasonable correlation coefficients for the MaGECK RRA coefficients. and transparency of the methods. They also did a limited amount of comparison of their 'hits' with patient data, though there are limited amounts of reliable well-documented patient datasets out there, and I am quite satisfied with what the authors have done. They also better addressed issues with PLK4 Knock-down effects on G2/M accumulation, but I think the comparisons they are showing in Figure 6a and b are incorrect (see below). This will not change the conclusions, but the correct comparisons are more appropriate for examining Plk4 effects. The data on the effects of PLK4 on reducing centrosome number in unsynchronized cells yet an increase in centrosome number in PLK4 knock-down cells arrested at G2/M fits with the authors' model of failed cytokinesis or issues related to the function of Plk4 on the PCM during G2.

Response: We thank the reviewer for these positive comments.

There are a few minor remaining issues the authors should address.

1. Lines 322-328 – here the authors are describing loss of G1/S phase regulators, but the papers they cite describing ERK activity (Tang et al 2002) deals with G2/M progression, not G1/S progression. The correct reference showing that DNA damage regulates G1/S cell cycle arrest through ERK activation is Tentner et al, “Combined experimental and computational analysis of DNA damage signaling reveals context-dependent roles for Erk in apoptosis and G1/S arrest after genotoxic stress” *Molecular Systems Biology* 8:568 (2012), and this reference should be cited.

Response: We have now added the reference as the reviewer suggested (Ref. #46).

2. The correct statistical comparisons for G2/M arrest in Figure 6a are between the AAVS1 sgRNA and the PLK4 sgRNA for each of the respective cell lines, not between the different cell lines. That is, they should compare sgAAVS1-Vector with sgPLK4-Vector, and compare sgAAVS1-(delta)SLC43A2OR-1 with sgPLK4-(delta)SLC43A2OR-1, etc. The same set of comparisons should be done with panel b for the CFI-400945 treatment. Notably, in the sensitive cells, there is very little increase in G2/M arrest upon PLK4 depletion or inhibition, but this is much more pronounced in the resistant cells.

Response: In the **New Figs. 6a,6b**, we have now performed the comparison as the reviewer pointed out.

New Figs. 6a,b

a, Cell cycle analysis upon *PLK4* knockout in oxaliplatin-sensitive and -resistant cells. Mean \pm SD with $n = 3$ biological replicates. Unpaired two-sided t test, $**p < 0.01$ and $***p < 0.001$. **b**, Cell cycle analysis by BrdU and PI staining upon CFI-400945 treatment in oxaliplatin-sensitive and -resistant cells. Mean \pm SD with $n = 3$ biological replicates. Unpaired two-sided t test, $***p < 0.001$.

3. The caption to Figure 7 (line 1560) – “d.e.” should read “e.f.” and the reference to panel ‘f’ (line 1562) should refer to panel g instead.

Response: We have corrected this typo in the revised manuscript.

4. The paper would be improved if it was quickly edited by a native English speaker. There are some complex double-negative arguments related to depletion ‘hits’ in the screen and up- and down-regulated gene sets that are a bit challenging to understand, and several points where improved syntax would make the paper easier to read.

Response: We have further polished the language of this manuscript.